# A Stochastic Newton Algorithm for Distributed Convex Optimization

**Brian Bullins**
Toyota Technological
Institute at Chicago
bbullins@ttic.edu

**Kumar Kshitij Patel**
Toyota Technological
Institute at Chicago
kkpatel@ttic.edu

**Ohad Shamir**
Weizmann Institute of Science
ohad.shamir@weizmann.ac.il

**Nathan Srebro**
Toyota Technological
Institute at Chicago
nati@ttic.edu

**Blake Woodworth**
Toyota Technological
Institute at Chicago
blake@ttic.edu

## Abstract

We propose and analyze a stochastic Newton algorithm for homogeneous distributed stochastic convex optimization, where each machine can calculate stochastic gradients of the same population objective, as well as stochastic *Hessian-vector products* (products of an independent unbiased estimator of the Hessian of the population objective with arbitrary vectors), with many such stochastic computations performed between rounds of communication. We show that our method can reduce the number, and frequency, of required communication rounds compared to existing methods without hurting performance, by proving convergence guarantees for quasi-self-concordant objectives (e.g., logistic regression), alongside empirical evidence.

## 1  Introduction

Stochastic optimization methods that leverage parallelism have proven immensely useful in modern optimization problems. Recent advances in machine learning have highlighted their importance as these techniques now rely on millions of parameters and increasingly large training sets.

While there are many possible ways of parallelizing optimization algorithms, we consider the intermittent communication setting (Zinkevich et al., 2010; Cotter et al., 2011; Dekel et al., 2012; Shamir et al., 2014; Woodworth et al., 2018, 2021), where $M$ parallel machines work together to optimize an objective during $R$ rounds of communication, and where during each round each machine may perform some basic operation (e.g., access the objective by invoking some oracle) $K$ times, and then communicate with all other machines. An important example of this setting is when this basic operation gives independent, unbiased stochastic estimates of the gradient, in which case this setting includes algorithms like Local SGD (Zinkevich et al., 2010; Coppola, 2015; Zhou and Cong, 2018; Stich, 2019; Woodworth et al., 2020a), Minibatch SGD (Dekel et al., 2012), Minibatch AC-SA (Ghadimi and Lan, 2012), and many others.

We are motivated by the observation of Woodworth et al. (2020a) that for *quadratic* objectives, first-order methods such as one-shot averaging (Zinkevich et al., 2010; Zhang et al., 2013)—a special case of Local SGD with a single round of communication—can optimize the objective to a very high degree of accuracy. This prompts trying to reduce the task of optimizing general convex objectives to a short sequence of quadratic problems. Indeed, this is precisely the idea behind many second-order algorithms including Newton's method (Nesterov and Nemirovskii, 1994), trust-region methods

35th Conference on Neural Information Processing Systems (NeurIPS 2021).

Table 1: Convergence guarantees for different algorithms in the intermittent communication setting. Notation is as follows: $H$: smoothness; $U$: third-order-smoothness; $\sigma$: stochastic gradient variance; $\rho$: stochastic Hessian-vector product variance; $g(x; z)$: stochastic gradient oracle; $h(x, u; z')$: stochastic Hessian-vector product oracle (see Section 2 for complete details). For the sake of clarity, we omit additional constants and logarithmic factors.

| Algorithm (Reference) | Convergence Rate | Assumption, Oracle Access |
|---|---|---|
| Local SGD (Woodworth et al., 2020a) | $\frac{HB^2}{KR} + \frac{\sigma B}{\sqrt{MKR}} + \frac{H^{1/3}\sigma^{2/3}B^{4/3}}{K^{1/3}R^{2/3}}$ | A1 $g(x; z)$ |
| FEDAC (Yuan and Ma, 2020) | $\frac{HB^2}{KR^2} + \frac{\sigma B}{\sqrt{MKR}}$ $+ \min\left\{ \frac{H^{1/3}\sigma^{2/3}B^{4/3}}{K^{1/3}R}, \frac{H^{1/2}\sigma^{1/2}B^{3/2}}{K^{1/4}R} \right\}$ | A1 $g(x; z)$ |
| Local SGD (Yuan and Ma, 2020) | $\frac{HB^2}{KR} + \frac{\sigma B}{\sqrt{MKR}} + \frac{U^{1/3}\sigma^{2/3}B^{5/3}}{K^{1/3}R^{2/3}}$ | A3 ($3^{rd}$-order Smooth) $g(x; z)$ |
| FEDAC (Yuan and Ma, 2020) | $\frac{HB^2}{KR^2} + \frac{\sigma B}{\sqrt{MKR}}$ $+ \frac{H^{1/3}\sigma^{2/3}B^{4/3}}{M^{1/3}K^{1/3}R} + \frac{U^{1/3}\sigma^{2/3}B^{5/3}}{K^{1/3}R^{4/3}}$ | A3 ($3^{rd}$-order Smooth) $g(x; z)$ |
| FEDSN **(Theorem 1)** | exp. decay $+ \frac{HB^2}{KR} + \frac{\sigma B}{\sqrt{MK}} + \frac{\rho B^2}{\sqrt{K}R}$ | A2 (QSC) $g(x; z), h(x, u; z')$ |

(Nocedal and Wright, 2006), and cubic regularization (Nesterov and Polyak, 2006), as well as methods that go beyond second-order information (Nesterov, 2019; Bullins, 2020).

Computing each Newton step requires solving, for convex $F$, a linear system of the form $\nabla^2 F(x)\Delta x = -\nabla F(x)$. Unfortunately, this may be prohibitive in a high dimensional setting, and may not even be feasible if $F$ is only accessible through a stochastic oracle in a streaming fashion, as is the case in the setting we consider. To avoid these issues, we reformulate the Newton step as the solution to a convex quadratic problem, $\min_{\Delta x} \frac{1}{2}\Delta x^\top \nabla^2 F(x)\Delta x + \nabla F(x)^\top \Delta x$, which we then solve using one-shot averaging. Conveniently, computing stochastic gradient estimates for this quadratic objective does not require computing the full Hessian matrix, as it only requires stochastic gradients and *stochastic Hessian-vector products*. This is attractive computationally since, for many problems, the cost of computing stochastic Hessian-vector products is similar to the cost of computing stochastic gradients, and both involve similar operations (Pearlmutter, 1994). Furthermore, highlighting the importance of these estimates, recent works have relied on Hessian-vector products to attain faster rates for reaching approximate stationary points in both deterministic (Agarwal et al., 2017; Carmon et al., 2018) and stochastic (Allen-Zhu, 2018; Arjevani et al., 2020) non-convex optimization.

In the context of distributed optimization, second-order methods have shown promise in the empirical risk minimization (ERM) setting, whereby estimates of $F$ are constructed by distributing the component functions of the finite-sum problem across machines. Such methods which leverage this structure have since been shown to lead to improved communication efficiency (Shamir et al., 2014; Zhang and Xiao, 2015; Reddi et al., 2016; Wang et al., 2018; Crane and Roosta, 2019; Islamov et al., 2021; Gupta et al., 2021). An important difference, however, is that these methods work in a batch setting, meaning they allow for repeated access to the same $K$ examples each round on each machine, giving a total of $MK$ samples. In contrast, we work in the stochastic (one-pass, streaming) setting, and so our model independently samples a fresh set of $MK$ examples per round, for a total of $MKR$ examples (see Appendix G.5 for an empirical comparison).

**Our results**

Our primary algorithmic contribution, which we present in Section 3 (and include a sketch in Appendix A), is the method FEDERATED-STOCHASTIC-NEWTON (FEDSN), a distributed approximate Newton method which leverages the benefits of one-shot averaging for quadratic problems. We pro-

vide in Section 3, under the condition of quasi-self-concordance (Bach, 2010), the main guarantees of our method (Theorem 1). In Section 4 we show how, for some regimes in terms of $M$, $K$, and $R$, our method may improve upon the rates of previous first-order methods, including FEDAC (Yuan and Ma, 2020). In Section 5, we compare a more practical version of our method, FEDSN-LITE (Algorithm 6) against the other methods, showing we can significantly reduce communication compared to other first-order methods.

## 2 Preliminaries

We consider the following optimization problem:

$$\min_{x \in \mathbb{R}^d} F(x), \tag{1}$$

and throughout we use $F^*$ to denote the minimum of this problem. We further use $\|\cdot\|$ to denote the standard $\ell_2$ norm, we let $\|x\|_{\mathbf{A}} := \sqrt{x^\top \mathbf{A} x}$ for a positive semidefinite matrix $\mathbf{A}$, and we let $\mathbf{I}$ denote the identity matrix of order $d$.

Next, we establish several sets of assumptions, beginning with those which are standard for smooth, stochastic, distributed convex optimization. We would note that we are working in the *homogeneous* distributed setting (i.e., each machine may access the same distribution), rather than the heterogeneous setting (Khaled et al., 2019; Karimireddy et al., 2019; Koloskova et al., 2020; Woodworth et al., 2020b; Khaled et al., 2020).

**Assumption 1** (A1).

(a) *F is convex, differentiable, and H-smooth, i.e., for all $x, y \in \mathbb{R}^d$, $F(y) \leq F(x) + \nabla F(x)^\top (y - x) + \frac{H}{2}\|y - x\|^2$.*

(b) *There is a minimizer $x^* \in \arg\min_x F(x)$ such that $\|x^*\| \leq B$.*

(c) *We are given access to a stochastic first-order oracle in the form of an estimator $g : \mathbb{R}^d \times \mathcal{Z} \mapsto \mathbb{R}^d$, and a distribution $\mathcal{D}$ on $\mathcal{Z}$ such that, for any $x \in \mathbb{R}^d$ queried by the algorithm, the oracle draws $z \sim \mathcal{D}$, and the algorithm observes an estimate $g(x; z)$ that satisfies:*

   (i) *$g(x; z)$ is an unbiased gradient estimate, i.e., $\mathbb{E}_z g(x; z) = \nabla F(x)$.*
   (ii) *$g(x; z)$ has bounded variance, i.e., $\mathbb{E}_z \|g(x; z) - \nabla F(x)\|^2 \leq \sigma^2$.*

In order to provide guarantees for Newton-type methods, we will require additional notions of smoothness. In particular, we consider $\alpha$-quasi-self-concordance (QSC) (Bach, 2010), which for convex and three-times differentiable $F$ is satisfied for $\alpha \geq 0$ when, for all $x \in \text{dom}(F)$, $v, u \in \mathbb{R}^d$,

$$|\nabla^3 F(x)[v, u, u]| \leq \alpha \|v\| \left(\nabla^2 F(x)[u, u]\right),$$

where we define

$$\nabla^k F(x)[u_1, u_2, \ldots, u_k] := \frac{\partial^k}{\partial u_1, \partial u_2, \ldots, \partial u_k}\Big|_{t_1=0, t_2=0, \ldots, t_k=0} F(x + t_1 u_1 + t_2 u_2 + \cdots + t_k u_k),$$

for $k \geq 1$, i.e., the $k^{th}$ directional derivative of $F$ at $x$ along the directions $u_1, u_2, \ldots, u_k$. Related to this is the condition of $\alpha$-self-concordance, which has proven useful for classic problems in linear optimization (Nesterov and Nemirovskii, 1994), whereby for all $x \in \text{dom}(F), u \in \mathbb{R}^d$,

$$|\nabla^3 F(x)[u, u, u]| \leq 2\alpha \left(\nabla^2 F(x)[u, u]\right)^{3/2}.$$

Though quasi-self-concordance is perhaps not as widely studied as self-concordance, recent work has brought its usefulness to light in the context of machine learning (Bach, 2010; Karimireddy et al., 2018; Carmon et al., 2020). Notably, for logistic regression, i.e., problems of the form

$$\min_x F(x) = \frac{1}{N}\sum_{i=1}^{N} \log\left(1 + e^{-b_i \langle a_i, x \rangle}\right), \tag{2}$$

we observe that $\alpha$-quasi-self-concordance holds with $\alpha \leq \max_i\{\|b_i a_i\|\}$. Interestingly, this function is *not* self-concordant, thus highlighting the importance of introducing the notion of QSC for such problems, and indeed, neither of these conditions implies the other in general.[1]

---

[1]For the other direction, note that $F(x) = -\ln(x)$ is 1-self-concordant but not quasi-self-concordant.

We now introduce further assumptions in terms of both additional oracle access and other smoothness notions. The following outlines the requirements for the stochastic Hessian-vector products, though we again stress that the practical cost of such an oracle is often on the order of that for stochastic gradients (Pearlmutter, 1994; Allen-Zhu, 2018).

**Assumption 2** (A2). *In addition to Assumption 1, we have:*

(a) *$F$ is three-times differentiable and $\alpha$-quasi-self-concordant, i.e., for all $x, v, u \in \mathbb{R}^d$, $\left|\nabla^3 F(x)[v, u, u]\right| \leq \alpha\|v\|\nabla^2 F(x)[u, u]$.*

(b) *We are given access to a stochastic Hessian-vector product oracle in the form of an estimator $h : \mathbb{R}^d \times \mathbb{R}^d \times \mathcal{Z} \mapsto \mathbb{R}^d$, and a distribution $\mathcal{D}$ on $\mathcal{Z}$ such that, for any pair $x, u \in \mathbb{R}^d$ queried by the algorithm, the oracle draws $z' \sim \mathcal{D}$, and the algorithm observes an estimate $h(x, u; z')$ that satisfies:*

  (i) *$h(x, u; z')$ is an unbiased Hessian-vector product estimate, i.e., $\mathbb{E}_{z'} h(x, u; z') = \nabla F^2(x)u$.*
  (ii) *$h(x, u; z')$ has bounded variance of the form $\mathbb{E}_{z'}\|h(x, u; z') - \nabla^2 F(x)u\|^2 \leq \rho^2\|u\|^2$.*

Meanwhile, other works (e.g., Yuan and Ma, 2020) require different control over third-order smoothness and fourth central moment. We do not require this assumption in our analysis, and include it here for comparison.

**Assumption 3** (A3). *In addition to Assumption 1, we have:*

(a) *$F$ is twice-differentiable and $U$-third-order-smooth, i.e., for all $x, y \in \mathbb{R}^d$, $F(y) \leq F(x) + \nabla F(x)^\top(y - x) + \frac{1}{2}\left\langle \nabla^2 F(x)(y - x),\, y - x\right\rangle + \frac{U}{6}\|y - x\|^3$.*

(b) *$g(x; z)$ has bounded fourth central moment, i.e., $\mathbb{E}_z\|g(x; z) - \nabla F(x)\|^4 \leq \sigma^4$.*

# 3 Main results

We begin by describing our main algorithm, FEDSN (Algorithm 1). Namely, our aim is to solve convex minimization problems $\min_x F(x)$, subject to Assumption 2.

---

**Algorithm 1** FEDERATED-STOCHASTIC-NEWTON, a.k.a., FEDSN($x_0$)

---

(Operating on objective $F(\cdot)$ with stochastic gradient $g(\cdot; \cdot)$ and Hessian-vector product $h(\cdot; \cdot, \cdot)$ oracles.)
**Input:** $x_0 \in \mathbb{R}^d$.
**Hyperparameters:** $T$: main iterations; and $\bar{\xi}$: local stability (see Table 4).
**Output:** Approximate solution to $\min_x F(x)$            ▷ See Theorem 1
    **for** $t = 0, 1, \ldots, T - 1$ **do**

$$\Delta\tilde{x}_t = \text{CONSTRAINED-QUADRATIC-SOLVER}(x_t) \qquad \triangleright \text{Approx. } \min_{u:\|u\|\leq\frac{1}{2}\bar{r}} \begin{aligned}&\frac{\bar{\xi}}{2}u^\top\nabla^2 F(x_t)u\\&+\nabla F(x_t)^\top u\end{aligned}$$

        Update: $x_{t+1} = x_t + \Delta\tilde{x}_t$
    **Return:** $x_T$

---

We will rely throughout the paper on several hyperparameter settings and parameter functions, which we collect in Tables 3 and 4. Recall that $M$ is the amount of parallel workers, $R$ is the number of rounds of communication, $K$ is the number of basic operations performed between rounds of communication, and $H$, $B$, $\sigma$, $\alpha$, and $\rho$ are as defined in Assumptions 1 and 2.

Among our assumptions, we note in particular the condition of quasi-self-concordance, under which several works have provided efficient optimization methods. For example, Bach (2010) analyzes Newton's method under QSC conditions, in a manner analogous to that of standard self-concordance analyses, to establish its behavior in the region of quadratic $(\log\log(1/\epsilon))$ convergence. More recently, both Karimireddy et al. (2018) and Carmon et al. (2020) have presented methods which rely instead on a trust-region approach, whereby, for a given iterate $x_t$, each iteration amounts to approximately solving a *constrained* subproblem of the form $\min_{\Delta x:\|\Delta x\|\leq c} \frac{\xi}{2}\Delta x^\top\nabla^2 F(x_t)\Delta x + \nabla F(x_t)^\top\Delta x$, for some $\xi \geq 1$ and problem-dependent radius $c > 0$. This stands in constrast to the

| Hyperparameter Setting | Description |
| --- | --- |
| $T := \left\lfloor \frac{R}{4\zeta} \log^2\left(\left(\frac{R}{\zeta}\right)\right) \right\rfloor$ 
 (for $\zeta = 4096 + 4(80 + 32\log K + 24\log(1 + 2\alpha B))^2$) | Main iterations |
| $\beta := 0$ | Momentum |
| $\bar{r} := \min\left\{\frac{32B}{T}\log(TK), \frac{1}{5\alpha}\right\}$ | Trust-region radius |
| $\bar{\xi} := \exp(\alpha\bar{r})$ | Local stability |
| $\bar{\lambda} := \max\left\{\frac{2eH}{K-2}, \frac{2\rho}{\sqrt{K}}, \frac{32eH\log(51200)}{K}, \frac{4\rho\sqrt{2\log(51200)}}{\sqrt{K}}, \frac{320\sqrt{2}\rho}{\sqrt{MK}}, \frac{320\sigma}{\bar{r}\sqrt{MK}}, \frac{8eH}{K-16}\right\}$ | Regularization bound |
| $N := \left\lceil 1 + \frac{5}{2}\log\frac{H(B+5T\bar{r})}{3\bar{\lambda}\bar{r}} \right\rceil$ | Binary search iterations |
| $C := \left\lceil 8\log\left(\lceil\log_2 N\rceil\left(4 + \frac{eH}{\bar{\lambda}} + \frac{80H(B+5T\bar{r})}{\bar{\lambda}\bar{r}}\right)\right)\right\rceil$ | Reg. quadratic repetitions |

Table 2: Hyperparameters $T$, $\beta$, $\bar{r}$, $\bar{\xi}$, $\bar{\lambda}$, $N$, and $C$, as used by FEDSN and its subroutines.

| Parameter Function | Description |
| --- | --- |
| $\eta_k(\lambda) := \begin{cases} \eta_\lambda & K \le \frac{2}{\lambda}\max\left\{\bar{\xi}H + \lambda, \frac{\rho^2}{\lambda}\right\} \text{ or } k < \frac{K}{2} \\ \frac{4}{\lambda\left(\frac{8}{\lambda}\max\left\{\bar{\xi}H+\lambda, \frac{\rho^2}{\lambda}\right\}+k-\frac{K}{2}\right)} & K > \frac{2}{\lambda}\max\left\{\bar{\xi}H + \lambda, \frac{\rho^2}{\lambda}\right\} \text{ and } k \ge \frac{K}{2} \end{cases}$ | Reg. quad. stepsizes |
| $w_k(\lambda) := \begin{cases} (1 - \lambda\eta_\lambda + \eta_\lambda^2\rho^2)^{-k-1} & K \le \frac{2}{\lambda}\max\left\{\bar{\xi}H + \lambda, \frac{\rho^2}{\lambda}\right\} \\ 0 & K > \frac{2}{\lambda}\max\left\{\bar{\xi}H + \lambda, \frac{\rho^2}{\lambda}\right\} \text{ and } k < \frac{K}{2} \\ \frac{8}{\lambda}\max\left\{\bar{\xi}H, \frac{\rho^2}{\lambda}\right\} \\ \quad + k - \frac{K}{2} - 1 & K > \frac{2}{\lambda}\max\left\{\bar{\xi}H + \lambda, \frac{\rho^2}{\lambda}\right\} \text{ and } k \ge \frac{K}{2} \end{cases}$ | Reg. quad. weights |

Table 3: Parameter functions $\eta_k(\lambda)$ and $w_k(\lambda)$, as used by FEDSN and its subroutines, where $\bar{\bar{\xi}}$ is as defined in Table 4, and where $\eta_\lambda$ denotes $\eta(\lambda) := \frac{1}{2}\min\left\{\frac{1}{\bar{\xi}H + \lambda}, \frac{\lambda}{\rho^2}\right\}$.

*unconstrained* minimization problem $\min_{\Delta x} \frac{\xi}{2}\Delta x^\top \nabla^2 F(x)\Delta x + \nabla F(x)^\top \Delta x$, which, as we may recall, forms the basis of the standard (damped) Newton method. Carmon et al. (2020) further use their trust-region subroutine to approximately implement a certain $\ell_2$-ball minimization oracle, which they combine with an acceleration scheme (Monteiro and Svaiter, 2013).

These results show, at a high level, that as long as the radius of the constrained quadratic (trust-region) subproblem is not too large, it is possible to make sufficient progress on the *global* problem by approximately solving the quadratic subproblem. Our method proceeds in a similar fashion: each iteration of Algorithm 1 provides an approximate solution to a constrained quadratic problem. To begin, we follow Karimireddy et al. (2018) in defining $\delta(r)$-local (Hessian) stability.

**Definition 1.** *Let $\delta : \mathbb{R}^+ \mapsto \mathbb{R}^+$. We say that a twice-differentiable and convex function $F$ is $\delta(r)$-locally stable if, for any $r > 0$ and any $x, y \in \mathbb{R}^d$ ($x \ne y$) such that $\|x - y\| \le r$ and $\|x - y\|_{\nabla^2 F(x)} > 0$, we have $\|x - y\|_{\nabla^2 F(y)} \le \delta(r)\|x - y\|_{\nabla^2 F(x)}$.*

As the next lemma shows, quasi-self-concordance is sufficient to provide this type of local stability.

**Lemma 1** (Theorem I (Karimireddy et al., 2018)). *If $F$ is $\alpha$-quasi-self-concordant, then $F$ is $\delta(r) = \exp(\alpha r)$-locally stable.*

The advantage of local stability is that it ensures that approximate solutions to locally-defined constrained quadratic problems can guarantee progress on the global objective, and we state this more formally in the following lemma. Note that this lemma is similar to (Theorem IV Karimireddy et al., 2018), though we allow for an additive error in the subproblem solves in addition to the multiplicative error, and its proof can be found in Appendix B.

**Lemma 2.** *Let $F$ satisfy Assumption 2 and be $\delta(r)$-locally stable for $\delta : \mathbb{R}^+ \mapsto \mathbb{R}^+$, let $x_0 \in \mathbb{R}^d$ be as input to FEDSN (Algorithm 1), let $c > 0$, let $\theta \in [0, 1)$, and define $Q_t(\Delta x) := \frac{\delta(5c)}{2}\Delta x^\top \nabla^2 F(x_t)\Delta x + \nabla F(x_t)^\top \Delta x$, where $x_t$ is the $t^{th}$ iterate of Algorithm 1. Furthermore, suppose we are given that in each iteration of Algorithm 1, $\|\Delta \tilde{x}_t\| \leq 5c$ and*

$$\mathbb{E}Q_t(\Delta \tilde{x}_t) - \min_{\Delta x: \|\Delta x\| \leq \frac{1}{2}c} Q_t(\Delta x) \leq \theta\left( Q_t(0) - \min_{\Delta x: \|\Delta x\| \leq \frac{1}{2}c} Q_t(\Delta x) \right) + \epsilon,$$

*for $\epsilon > 0$. Then for each $T \geq 0$, Algorithm 1 guarantees*

$$\mathbb{E}F(x_T) - F^* \leq \mathbb{E}[F(x_0) - F^*]\exp\left(-\frac{Tc(1-\theta)}{2B\delta(\frac{1}{2}c)\delta(5c)}\right) + \frac{2B\delta(\frac{1}{2}c)d(5c)\epsilon}{c(1-\theta)} .$$

We have now seen how to turn approximate solutions of constrained quadratic problems into an approximate minimizer of the overall objective. We next need to ensure that the output of our method CONSTRAINED-QUADRATIC-SOLVER (Algorithm 2) meets the conditions of Lemma 2. As previously discussed, Woodworth et al. (2020a) showed that first-order methods can very accurately optimize *unconstrained* quadratic objectives using a single round of communication; however, here we need to optimize a quadratic problem subject to a norm constraint. Our constrained quadratic solver is thus based on the following idea: the minimizer of the *constrained* problem $\min_{x:\|x\| \leq c} Q(x)$ is the same as the minimizer of the *unconstrained* problem $\min_x Q(x) + \frac{\lambda^*}{2}\|x\|^2$ for some problem-dependent regularization parameter $\lambda^*$. While the algorithm does not know what $\lambda^*$ should be a priori, we show that it can be found with sufficient confidence using binary search. Lemma 3, proven in Appendix C, provides the relevant guarantees.

**Lemma 3.** *Let $F$ satisfy Assumption 2, let $x$ be as input to CONSTRAINED-QUADRATIC-SOLVER (Algorithm 2), let $\bar{\xi}$ be as in Table 4, define*

$$Q(u) := \frac{\bar{\xi}}{2}u^\top \nabla^2 F(x)u + \nabla F(x)^\top u, \qquad Q_\lambda(u) := \frac{1}{2}u^\top(\bar{\xi}\nabla^2 F(x) + \lambda\mathbf{I})u + \nabla F(x)^\top u,$$

$$u_\lambda^* := \arg\min_u Q_\lambda(u), \qquad r^*(\lambda) := \|u_\lambda^*\|,$$

*and let $\lambda_r$ denote, for any $r > 0$, the value such that $r^*(\lambda_r) = r$. Let $\hat{u}$ be the output of Algorithm 2 for hyperparameters $\bar{r}$, $\bar{\xi}$, $\bar{\lambda}$, $N$ and $C$ as in Table 4, and suppose the output $\tilde{u}_\lambda$ of REGULARIZED-QUADRATIC-SOLVER$(x, \lambda)$ satisfies for all $\lambda \geq \bar{\lambda}$ that*

$$\mathbb{E}Q_\lambda(\tilde{u}_\lambda) - \min_u Q_\lambda(u) \leq \epsilon(\lambda) := \frac{\lambda(r^*(\lambda)^2 + \bar{r}^2)}{800} .$$

*Then $\|\hat{u}\| \leq 5\bar{r}$ and*

$$\mathbb{E}Q(\hat{u}) - \min_{u:\|u\| \leq \frac{1}{2}\bar{r}} Q(u) \leq \frac{3}{4}\left( Q(0) - \min_{u:\|u\| \leq \frac{1}{2}\bar{r}} Q(u) \right) + \epsilon(\lambda_{4\bar{r}}) + \frac{\bar{\lambda}\bar{r}^2}{4} .$$

We now show that using one-shot averaging (Zinkevich et al., 2010; Zhang et al., 2012) with $M$ machines—i.e., averaging the results of $M$ independent runs of SGD—suffices to solve each quadratic problem to the desired accuracy. The following lemma, which we prove in Appendix D, establishes that REGULARIZED-QUADRATIC-SOLVER (Algorithm 3) supplies Algorithm 2 with an output $\hat{u}$ that satisfies the conditions of Lemma 3.

---

**Algorithm 2** CONSTRAINED-QUADRATIC-SOLVER($x$)

---

(Operating on objective $F(\cdot)$ with stochastic gradient $g(\cdot;\cdot)$ and Hessian-vector product $h(\cdot;\cdot,\cdot)$ oracles.)

**Input:** $x \in \mathbb{R}^d$.

**Hyperparameters:** $\bar{r}$: trust-region radius; $\bar{\xi}$: local stability; $\bar{\lambda}$: regularization bound; $N$: binary search iterations; and $C$: reg. quadratic repetitions (see Table 4).

**Output:** Approximate solution to $\min_{u:\|u\|\le\frac{1}{2}\bar{r}} \frac{\bar{\xi}}{2} u^\top \nabla^2 F(x) u + \nabla F(x)^\top u$      ▷ See Lemma 3

$\quad \Lambda_1 = \left\{ \bar{\lambda}\left(\frac{3}{2}\right)^{n-1} : n = 1, \ldots, N \right\}$

$\quad i \leftarrow 1$

$\quad$ **while** $\Lambda_i \ne \varnothing$ **do**

$\quad\quad \lambda^{(i)} = \text{Median}(\Lambda_i)$

$\quad\quad$ **for** $c = 1, \ldots, C$ **do**

$\quad\quad\quad \tilde{u}^{(i,c)} = \text{REGULARIZED-QUADRATIC-SOLVER}(x, \lambda^{(i)})$

$\quad\quad$ **if** $\left| \left\{ \tilde{u}^{(i,c)} : \left\| \tilde{u}^{(i,c)} \right\| \in \left[\frac{3}{2}\bar{r}, \frac{7}{2}\bar{r}\right] \right\} \right| > \frac{C}{2}$ **then**

$\quad\quad\quad \tilde{u} = \text{REGULARIZED-QUADRATIC-SOLVER}(x, \lambda^{(i)})$

$\quad\quad\quad$ **Return:** $\hat{u} = \min\left\{1, \frac{5\bar{r}}{\|\tilde{u}\|}\right\}\tilde{u}$

$\quad\quad$ **else if** $\left| \left\{ \tilde{u}^{(i,c)} : \left\| \tilde{u}^{(i,c)} \right\| \le \frac{5}{2}\bar{r} \right\} \right| > \frac{C}{2}$ **then**

$\quad\quad\quad \Lambda_{i+1} = \left\{ \lambda' \in \Lambda_i : \lambda' < \lambda^{(i)} \right\}$

$\quad\quad$ **else if** $\left| \left\{ \tilde{u}^{(i,c)} : \left\| \tilde{u}^{(i,c)} \right\| > \frac{5}{2}\bar{r} \right\} \right| > \frac{C}{2}$ **then**

$\quad\quad\quad \Lambda_{i+1} = \left\{ \lambda' \in \Lambda_i : \lambda' > \lambda^{(i)} \right\}$

$\quad\quad$ **else**

$\quad\quad\quad$ **Return:** $\hat{u} = 0$

$\quad\quad i \leftarrow i + 1$

$\quad \tilde{u} = \text{REGULARIZED-QUADRATIC-SOLVER}(x, \bar{\lambda})$

$\quad$ **Return:** $\hat{u} = \min\left\{1, \frac{5\bar{r}}{\|\tilde{u}\|}\right\}\tilde{u}$

---

**Lemma 4.** *Let $F$ satisfy Assumption 2, let $x \in \mathbb{R}^d$, $\lambda \in \mathbb{R}^+$ be as input to* REGULARIZED-QUADRATIC-SOLVER *(Algorithm 3), let $Q_\lambda(u) = \frac{1}{2} u^\top (\bar{\xi}\nabla^2 F(x) + \lambda\mathbf{I})u + \nabla F(x)^\top u$, let $Q_\lambda^* := \min_u Q_\lambda(u)$, let $u^* := \arg\min_u Q_\lambda(u)$, and let stochastic first-order and stochastic Hessian-vector product oracles for $F$, as defined in Assumptions 1 and 2, respectively, be available for each call to* REGULARIZED-QUADRATIC-GRADIENT-ACCESS *(Algorithm 4), for either Case 1* (Different-Samples) *or Case 2* (Same-Sample)*. Let $\hat{u}$, as output by Algorithm 3, be a weighted average of the iterates of $M$ independent runs of SGD with stepsizes $\eta_0(\lambda), \ldots, \eta_{K-1}(\lambda)$, i.e., $\hat{u} = \frac{1}{M\sum_{k=0}^{K-1} w_k} \sum_{m=1}^{M} \sum_{k=0}^{K-1} w_k u_k^m$. Then, for both Cases 1 and 2,*

$$
\mathbb{E}Q_\lambda(\hat{u}) - Q_\lambda^* \le
\begin{cases}
2\max\left\{\bar{\xi}H + \lambda, \frac{\rho^2}{\lambda}\right\}\|u^*\|^2 \min\left\{\frac{1}{K}, \exp\left(-\frac{K+1}{4}\min\left\{\frac{\lambda}{\bar{\xi}H+\lambda}, \frac{\lambda^2}{\rho^2}\right\}\right)\right\} \\
\qquad\qquad + \frac{2(\sigma^2 + \rho^2\|u^*\|^2)}{\lambda M K} \quad \text{if } K \le \frac{2}{\lambda}\max\left\{\bar{\xi}H + \lambda, \frac{\rho^2}{\lambda}\right\} \\[2ex]
96\lambda\|u^*\|^2 \exp\left(-\frac{K}{8}\min\left\{\frac{\lambda}{\bar{\xi}H+\lambda}, \frac{\lambda^2}{\rho^2}\right\}\right) + \frac{96(\sigma^2 + \rho^2\|u^*\|^2)}{\lambda M K} \\
\qquad\qquad\qquad \text{if } K > \frac{2}{\lambda}\max\left\{\bar{\xi}H + \lambda, \frac{\rho^2}{\lambda}\right\}.
\end{cases}
$$

Our analysis for Algorithm 3 is based on ideas similar to those of Woodworth et al. (2020a), whereby the algorithm may access the stochastic oracles via REGULARIZED-QUADRATIC-GRADIENT-ACCESS (Algorithm 4). However, additional care must be taken to account for the fact that Algorithm 4 supplies stochastic gradient estimates of the *quadratic subproblems* $Q_\lambda(u)$ as per the oracles models described in Assumptions 1 and 2. Thus, the estimates—based in part on stochastic Hessian-vector products—have variance that scales with the norm of the respective iterates of Algorithm 3 (see Assumption 2(b.ii)).

---
**Algorithm 3** REGULARIZED-QUADRATIC-SOLVER$(x, \lambda)$
---
(Operating on objective $F(\cdot)$ with stochastic gradient $g(\cdot; \cdot)$ and Hessian-vector product $h(\cdot; \cdot, \cdot)$ oracles.)

**Input:** $x \in \mathbb{R}^d$, $\lambda \in \mathbb{R}^+$.

**Hyperparameters:** $\beta$: momentum; $\bar{\xi}$: local stability; and parameter functions $\eta_k(\lambda)$, $w_k(\lambda)$ (see Tables 3 and 4).

**Output:** Approximate solution to $\min_u Q_\lambda(u) = \frac{1}{2}u^\top(\bar{\xi}\nabla^2 F(x) + \lambda \mathbf{I})u + \nabla F(x)^\top u$      ▷ See Lemma 4

     Initialize: $u_0^1, \ldots, u_0^M = 0$               ▷ Initial iterates on each machine

     **for** Each machine $m = 1, \ldots, M$ in parallel **do**

         **for** $k = 0, \ldots, K - 1$ **do**

             $\gamma(u_k^m; z_k^m, z_k^{'m}) = $ REGULARIZED-QUADRATIC-GRADIENT-ACCESS$(x, u_k^m, \lambda)$

             $u_{k+1}^m = u_k^m - \eta_k(\lambda)\gamma(u_k^m; z_k^m, z_k^{'m}) + \mathbb{1}_{\{k>0\}}\beta(u_k^m - u_{k-1}^m)$ [2]

     **Return:** $\tilde{u} = \frac{1}{M\sum_{k=1}^K w_k(\lambda)}\sum_{m=1}^M\sum_{k=1}^K w_k(\lambda)u_k^m$
---

We also note two possible cases for the oracle access: Case 1 (`Different-Samples`) in Algorithm 4 requires both a call to a stochastic first-order oracle (which draws $z \sim \mathcal{D}$) and a call to a stochastic Hessian-vector product oracle (which draws a different $z' \sim \mathcal{D}$); while Case 2 (`Same-Sample`) allows both stochastic estimators to be observed for the same random sample $z \sim \mathcal{D}$. These cases differ by only a small constant factor in the final convergence rate, and we base our practical method on this single sample model. We refer the reader to Appendix G.4 for discussion of these settings.

---
**Algorithm 4** REGULARIZED-QUADRATIC-GRADIENT-ACCESS$(x, u, \lambda)$
---
(Operating on objective $F(\cdot)$ with stochastic gradient $g(\cdot; \cdot)$ and Hessian-vector product $h(\cdot; \cdot, \cdot)$ oracles.)

**Input:** $x, u \in \mathbb{R}^d$, $\lambda \in \mathbb{R}^+$.

**Hyperparameters:** $\bar{\xi}$: local stability (see Table 4).

**Output:** $\gamma(u; z, z')$ s.t. $\mathbb{E}_{z,z'}[\gamma(u; z, z')] = \nabla Q_\lambda(u)$ and $\mathbb{E}_{z,z'}\|\gamma(u; z, z') - \nabla Q_\lambda(u)\|^2 \leq \sigma^2 + \rho^2\|u\|^2$

     **Case 1:** `Different-Samples` ($z, z'$ drawn independently for each stochastic oracle)

         • Query the stochastic first-order oracle at $x$ (as in Assumption 1(c)), so that the oracle draws $z \sim \mathcal{D}$, and observe $g(x; z)$

         • Query the stochastic Hessian-vector product oracle at $x$ and $u$ (as in Assumption 2(b)), so that the oracle draws $z' \sim \mathcal{D}$, and observe $h(x, u; z')$

     **Case 2:** `Same-Sample` (Same $z' = z$ used for both stochastic oracles)

         • Query the stochastic first-order oracle at $x$ (as in Assumption 1(c)), so that the oracle draws $z \sim \mathcal{D}$, and observe $g(x; z)$

         • Query the stochastic Hessian-vector product oracle at $x$ and $u$ (as in Assumption 2(b)) for $z' = z$, and observe $h(x, u; z')$

     $\gamma(u; z, z') \coloneqq \bar{\xi}h(x, u; z') + \lambda u + g(x; z)$

     **Return:** $\gamma(u; z, z')$
---

Finally, having analyzed Algorithms 1, 2, 3, and 4, we may put them all together to provide our main theoretical result, whose proof can be found in Appendix E.

**Theorem 1.** *Let $F$ satisfy Assumption 2. Then, for $K \geq 175$ and $R \geq \tilde{\Omega}(1)$, and for hyperparameters $T$, $\beta$, $\bar{r}$, $\bar{\xi}$, $\bar{\lambda}$, $N$, $C$ and parameter functions $\eta_k(\lambda)$, $w_k(\lambda)$ as in Tables 3 and 4, the output of FEDSN (Algorithm 1) with initial point $x_0 \in \mathbb{R}^d$, using Algorithms 2, 3, and 4 (for both Cases 1 and 2)*

---

[2]We add heavy-ball/Polyak momentum in this step with momentum parameter $\beta$. Our theoretical results do not require any momentum, and thus FEDSN is analyzed for $\beta = 0$. For our experiments we compare the algorithms both with and without momentum, i.e., $\beta = 0$ or optimally tuned $\beta \in \{0.1, 0.3, 0.5, 0.7, 0.9\}$.

*satisfies*

$$\mathbb{E}[F(x_T)] - F^* \leq HB^2\left(\exp\left(-\frac{R}{\tilde{O}(\alpha B)}\right) + \exp\left(-\frac{K}{O(1)}\right)\right) + \tilde{O}\left(\frac{\sigma B}{\sqrt{MK}} + \frac{HB^2}{KR} + \frac{\rho B^2}{\sqrt{K}R}\right),$$

*where $\tilde{\Omega}$, $\tilde{O}$ hide terms logarithmic in $R$, $K$, and $\alpha B$.*

## 4  Comparison with related methods and lower bounds

In this section, we compare our algorithm's guarantees with those of FEDAC, and we include additional comparisons in Appendix F. A difficulty in making this comparison is determining the "typical" relative scale of the parameters $H$, $\sigma$, $U$, $\alpha$, and $\rho$. Drawing inspiration from training generalized linear models, we consider a natural scaling of the parameters that arises when the objective has the form $F(x) = \mathbb{E}_z \ell(\langle x, z \rangle)$, where $|\ell'|$, $|\ell''|$, and $|\ell'''|$ are $O(1)$, and where $\|z\| \leq D$; this holds, e.g., for logistic regression problems (see (2)). In this case, upper bounds on the derivatives of $F$ will generally scale with $\|z\|$. So if we assume that $\|z\| \leq D$ for some $D$, then the derivatives of $F$ would scale as $\|\nabla F(x)\| \lesssim D$, $\|\nabla^2 F(x)\|_{\mathrm{op}} \lesssim D^2$, and $\|\nabla^3 F(x)\|_{\mathrm{op}} \lesssim D^3$, where $\|\cdot\|_{\mathrm{op}}$ denotes the operator norm. Thus, we will take $H = D^2$, $\sigma = D$, $U = D^3$, $\alpha = D$, and $\rho = D^2$. These parameters could have different relationships, but we focus on this regime for simplicity.

In addition to working within this natural scaling, we consider the case where we have access to sufficient machines (i.e., $M \gtrsim \frac{KR^3}{D^2B^2}$) and for $K$ large enough. We explore various regimes w.r.t. both the number of rounds of communication $R$ and the "size" of the problem $DB$. Thus, ignoring constants and terms logarithmic in $R$, $K$, and $\alpha B$, our upper bound from Theorem 1 reduces to

$$\mathbb{E}F(\hat{x}) - F^* \lesssim D^2B^2 \exp\left(-\frac{R}{DB}\right) + D^2B^2 \exp(-K) + \frac{D^2B^2}{KR^{3/2}} + \frac{D^2B^2}{KR} + \frac{D^2B^2}{\sqrt{K}R} \approx \frac{D^2B^2}{\sqrt{K}R}.$$

**Comparison with FEDAC**

The previous best known first-order distributed method under third-order smoothness assumptions is FEDAC (Yuan and Ma, 2020), an accelerated variant of Local SGD, which achieves a guarantee of

$$\mathbb{E}F(\hat{x}) - F^* \leq \tilde{O}\left(\frac{HB^2}{KR^2} + \frac{\sigma B}{\sqrt{MKR}} + \frac{H^{1/3}\sigma^{2/3}B^{4/3}}{M^{1/3}K^{1/3}R} + \frac{U^{1/3}\sigma^{2/3}B^{5/3}}{K^{1/3}R^{4/3}}\right).$$

For the setting as outlined above, this bound reduces to

$$\mathbb{E}F(\hat{x}) - F^* \lesssim \frac{D^2B^2}{KR^2} + \frac{D^{5/3}B^{5/3}}{K^{1/3}R^{4/3}}.$$

In the case where $DB$ is not too large ($DB \lesssim K^2R^2$), the dominant term for FEDAC is $\frac{D^{5/3}B^{5/3}}{K^{1/3}R^{4/3}}$, and so we see that our algorithm improves upon FEDAC as long as $R \lesssim \frac{\sqrt{K}}{DB}$, whereas for $R \gtrsim \frac{\sqrt{K}}{DB}$, FEDAC provides better guarantees than FEDSN.

**Comparison with first-order lower bounds**

Woodworth et al. (2021) provide lower bounds under other smoothness conditions, including quasi-self-concordance, which are relevant to the current work. Roughly speaking, they show that under Assumption 2(a), no first-order intermittent communication algorithm can guarantee suboptimality less than (ignoring constant and $\log M$ factors)

$$\mathbb{E}F(\hat{x}) - F^* \geq \frac{HB^2}{K^2R^2} + \frac{\sigma B}{\sqrt{MKR}} + \min\left\{\frac{HB^2}{R^2}, \frac{\alpha\sigma B^2}{\sqrt{K}R^2}, \frac{\sigma B}{\sqrt{K}R}\right\}.$$

In the same parameter regime as above, the lower bound reduces to

$$\mathbb{E}F(\hat{x}) - F^* \gtrsim \frac{D^2B^2}{K^2R^2} + \min\left\{\frac{D^2B^2}{\sqrt{K}R^2}, \frac{DB}{\sqrt{K}R}\right\}.$$

Comparing this lower bound with our guarantee in Theorem 1, we see that, when $DB = O(1)$ and the number of rounds of communication is small (e.g., $R = O(\log K)$), our approximate Newton method can (ignoring $\log K$ factors) achieve an upper bound of $\mathbb{E}F(\hat{x}) - F^* \lesssim 1/\sqrt{K}$. Therefore, in this important regime, FEDSN matches the lower bound under Assumption 2(a), albeit using a stronger oracle. No prior work has matched this lower bound, and so we do not know whether such an oracle is necessary in order to achieve it, or if perhaps the stronger oracle allows for breaking it.

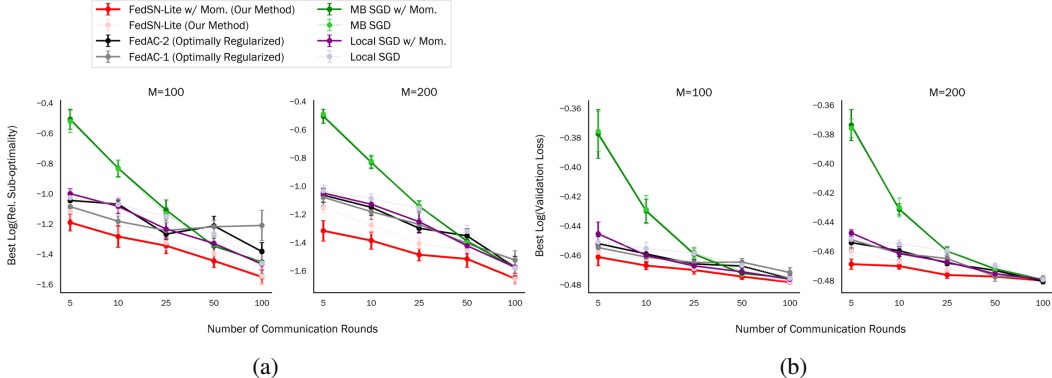

Figure 1: Empirical comparison of FEDSN-LITE (Algorithm 6) to other methods (see Appendix G.1) on the LIBSVM a9a (Chang and Lin, 2011; Dua and Graff, 2017) dataset for minimizing: (a) in-sample, and (b) out-of-sample unregularized logistic regression loss using $M \in \{100, 200\}$ machines. We vary the frequency of communication (horizontal axis of each plot), while keeping the total number of steps on each machine (theoretical parallel runtime) fixed at $KR = 100$. Thus, every point in the sub-plot is a separately tuned instance of an algorithm, where each algorithm besides FEDAC solves an unregularized ERM problem and reports (a) the best relative sub-optimality w.r.t. the optimal minimizer and (b) the best validation loss on a held-out dataset. All results are averaged over multiple runs (see Appendix G.3 for full details).

## 5   Experiments

In Appendix G.1 we present a more practical variant of FEDSN called FEDSN-LITE (Algorithm 6), which does away with the search over the regularization parameter as in Algorithm 2. We compare FEDSN-LITE against the two variants of FEDAC (Algorithm 7, Yuan and Ma (2020)), Minibatch SGD (Algorithm 9, Dekel et al. (2012)), and Local SGD (Algorithm 8, Zinkevich et al. (2010)). We also study the effect of adding Polyak's momentum, which we denote by $\beta$, to these algorithms (see Appendix G.1). FEDAC is mainly presented and analyzed for strongly convex functions by Yuan and Ma (2020). In fact, they assume the knowledge of the strong convexity constant to tune FEDAC, which is typically hard to know unless the function is explicitly regularized. To handle general convex functions, Yuan and Ma (2020) build some *internal regularization* into FEDAC (see Appendix E.2 in their paper). However, their hyperparameter recommendations in this setting also depend on unknowns such as the smoothness of the function and the variance of the stochastic gradients. This poses a difficulty in comparing FEDAC to the other algorithms, which do not require the knowledge of these unknowns.

To overcome this we take the more carefully optimized version of FEDAC for strongly convex functions and tune its internal regularization and learning rate. This emulates the setting where the objective is assumed to be just convex but FEDAC sees a strongly convex function instead. In our experiments in Figure 6, we notice that FEDSN-LITE is either competitive with or outperforms the other baselines. This is especially true for the sparse communication settings, which are of most practical interest. A more comprehensive set of experiments can be found in Appendix G.2 along with full implementational details in Appendix G.3.[3]

## 6   Conclusion

In this work, we have shown how to more efficiently optimize convex quasi-self-concordant objectives by leveraging parallel methods for quadratic problems. Our method can, in some parameter regimes, improve upon existing stochastic methods while maintaining a similar computational cost, and we have further seen how our method may provide empirical improvements in the low communication regime. It remains open whether the same guarantees we achieve here can also be achieved using only independent stochastic gradients (a single stochastic gradient on each sample), or whether in the distributed stochastic setting access to Hessian-vector products is strictly more powerful than access to only independent stochastic gradients.

---

[3]Code is availabe at https://github.com/kishinmh/Inexact-Newton.

**Acknowledgements.** This research was partially supported by NSF-BSF award 1718970. BW is supported by a Google Research PhD Fellowship. We also thank all the anonymous reviewers for their time and suggestions.

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
