**for** $t = 0, 1, \ldots, T-1$ **do**

      $\Lambda = \{\lambda_1, \ldots, \lambda_N\}$                              ▷ Set of regularization parameters

      **while** $\Lambda \neq \varnothing$ **do**

         $\lambda = \mathrm{Median}(\Lambda)$

         Define: $Q_\lambda(u) = \frac{1}{2} u^\top (\bar{\xi} \nabla^2 F(x) + \lambda \mathbf{I}) u + \nabla F(x)^\top u$ ▷ Regularized quadratic subproblem

         Run SGD on $Q_\lambda(u)$ independently on each machine (starting with $u_0 = 0$) for $K$ steps, then communicate the final iterate $u_K^m$ from each machine $m$, and average them to obtain $\tilde{u} = \frac{1}{M} \sum_{m=1}^{M} u_K^m$.

         **if** Sufficiently good $\lambda$ is found **then**

            $\Delta \tilde{x}_t = \tilde{u}$

            **break**

         **else**

            Reduce the size of $\Lambda$ by half.

      Update: $x_{t+1} = x_t + \Delta \tilde{x}_t$

   **Return:** $x_T$

---

# B  Analysis of Algorithm 1

We will use the following notation in our analysis:

$$Q_t^\sigma(\Delta x) = \frac{\sigma}{2} \Delta x^\top \nabla^2 F(x_t) \Delta x + \nabla F(x_t)^\top \Delta x. \tag{3}$$

**Lemma 5** (Lemma 5 (Karimireddy et al., 2018))**.** *Let $F$ be $\delta(r)$-locally stable for a given $\delta : \mathbb{R}^+ \mapsto \mathbb{R}^+$, let $\|x^*\| \leq B$, let $r > 0$, let $\gamma = r/B$ and $x_\gamma^* = (1-\gamma)x_t + \gamma x^*$, and let $x_{t+1} = x_t + \Delta \tilde{x}_t$ for $\|\Delta \tilde{x}_t\| \leq r$. Then*

$$F(x_{t+1}) - F(x_t) \leq Q_t^{\delta(r)}(\Delta \tilde{x}_t)$$
$$F(x_\gamma^*) - F(x_t) \geq Q_t^{1/\delta(r)}(x_\gamma^* - x_t).$$

**Lemma 6** (Lemma 6 (Karimireddy et al., 2018))**.** *For any convex domain $\mathcal{Q}$ and constants $a \cdot b \geq 1$,*

$$\min_{\Delta x \in \mathcal{Q}} Q^a(\Delta x) \leq \frac{1}{ab} \min_{\Delta x \in \mathcal{Q}} Q^{1/b}(\Delta x).$$

We will now prove Lemma 2 from Section 3.

**Lemma 2.** *Let $F$ satisfy Assumption 2 and be $\delta(r)$-locally stable for $\delta : \mathbb{R}^+ \mapsto \mathbb{R}^+$, let $x_0 \in \mathbb{R}^d$ be as input to FEDSN (Algorithm 1), let $c > 0$, let $\theta \in [0, 1)$, and define $Q_t(\Delta x) := \frac{\delta(5c)}{2} \Delta x^\top \nabla^2 F(x_t) \Delta x + \nabla F(x_t)^\top \Delta x$, where $x_t$ is the $t^{th}$ iterate of Algorithm 1. Furthermore, suppose we are given that in each iteration of Algorithm 1, $\|\Delta \tilde{x}_t\| \leq 5c$ and*

$$\mathbb{E} Q_t(\Delta \tilde{x}_t) - \min_{\Delta x: \|\Delta x\| \leq \frac{1}{2}c} Q_t(\Delta x) \leq \theta \left( Q_t(0) - \min_{\Delta x: \|\Delta x\| \leq \frac{1}{2}c} Q_t(\Delta x) \right) + \epsilon,$$

*for $\epsilon > 0$. Then for each $T \geq 0$, Algorithm 1 guarantees*

$$\mathbb{E}F(x_T) - F^* \leq \mathbb{E}[F(x_0) - F^*] \exp\left(-\frac{Tc(1-\theta)}{2B\delta(\frac{1}{2}c)\delta(5c)}\right) + \frac{2B\delta(\frac{1}{2}c)d(5c)\epsilon}{c(1-\theta)} .$$

*Proof.* To begin, we are given by the assumptions of the theorem statement that Algorithm 1 chooses the update $\Delta\tilde{x}_t$ such that $\|\Delta\tilde{x}_t\| \leq 5c$ and

$$\mathbb{E}Q_t^{\delta(5c)}(\Delta\tilde{x}_t) - \min_{\|\Delta x\| \leq \frac{1}{2}c} Q_t^{\delta(5c)}(\Delta x) \leq \epsilon + \theta\left(Q_t^{\delta(5c)}(0) - \min_{\|\Delta x\| \leq \frac{1}{2}c} Q_t^{\delta(5c)}(\Delta x)\right). \quad (4)$$

Therefore,

$$\mathbb{E}[F(x_{t+1}) - F(x_t)] \leq \mathbb{E}Q_t^{\delta(5c)}(\Delta\tilde{x}_t) \tag{5}$$

$$\leq \epsilon + (1-\theta) \min_{\Delta x \,:\, \|\Delta x\| \leq \frac{1}{2}c} Q_t^{\delta(5c)}(\Delta x) \tag{6}$$

$$\leq \epsilon + \frac{1-\theta}{\delta(5c)\delta(\frac{1}{2}c)} \min_{\Delta x \,:\, \|\Delta x\| \leq \frac{1}{2}c} Q_t^{1/\delta(\frac{1}{2}c)}(\Delta x) \tag{7}$$

$$\leq \epsilon + \frac{1-\theta}{\delta(5c)\delta(\frac{1}{2}c)} Q_t^{1/\delta(\frac{1}{2}c)}\left(\left(1 - \frac{c}{2B}\right)x_t + \frac{c}{2B}x^* - x_t\right) \tag{8}$$

$$\leq \epsilon + \frac{1-\theta}{\delta(5c)\delta(\frac{1}{2}c)}\left[F\left(\left(1 - \frac{c}{2B}\right)x_t + \frac{c}{2B}x^*\right) - F(x_t)\right] \tag{9}$$

$$\leq \epsilon + \frac{1-\theta}{\delta(5c)\delta(\frac{1}{2}c)}\left[\left(1 - \frac{c}{2B}\right)F(x_t) + \frac{c}{2B}F^* - F(x_t)\right] \tag{10}$$

$$= \epsilon - \frac{c(1-\theta)}{2B\delta(5c)\delta(\frac{1}{2}c)}[F(x_t) - F^*]. \tag{11}$$

Here we used Lemma 5 for the first inequality, Lemma 6 for the third, Lemma 5 again for the fifth, and the convexity of $F$ for the sixth.

Rearranging, and unravelling the recursion, we conclude that

$$\mathbb{E}[F(x_T) - F^*] \leq \left(1 - \frac{c(1-\theta)}{2B\delta(5c)\delta(\frac{1}{2}c)}\right)^T \mathbb{E}[F(x_0) - F^*] + \epsilon \sum_{t=0}^{T}\left(1 - \frac{c(1-\theta)}{2B\delta(5c)\delta(\frac{1}{2}c)}\right)^t \tag{12}$$

$$\leq \mathbb{E}[F(x_0) - F^*] \exp\left(-\frac{Tc(1-\theta)}{2B\delta(5c)\delta(\frac{1}{2}c)}\right) + \frac{2B\delta(5c)\delta(\frac{1}{2}c)\epsilon}{c(1-\theta)} . \tag{13}$$

This completes the proof. $\qquad\square$

## C  Proof of Lemma 3

Before we analyze Algorithm 2, we recall some key definitions. For a given $x \in \mathbb{R}^d$, we let

$$Q(u) = \frac{\bar{\xi}}{2}u^\top \nabla^2 F(x)u + \nabla F(x)^\top u, \tag{14}$$

and for a regularization penalty $\lambda$, we also define

$$Q_\lambda(u) = \frac{1}{2}u^\top(\bar{\xi}\nabla^2 F(x) + \lambda\mathbf{I})u + \nabla F(x)^\top u. \tag{15}$$

We use $u_\lambda^*$ to denote the (unique) minimizer of $Q_\lambda(u)$, and we use $r^*(\lambda) = \|u_\lambda^*\|$ to denote the norm of the minimizer. For $0 \leq r \leq r^*(0)$, we also use $\lambda_r$ to denote the value of $\lambda$ such that $r^*(\lambda_r) = r$ (Lemma 7 below shows that $\lambda_r$ is unique).

**Lemma 7** (Lemmas 35 and 36 (Carmon et al., 2020)). *For any $r$, there exists a unique $\lambda_r \geq 0$ such that*

$$u_r^* = \arg\min_{u:\|u\| \leq r} Q(u) = \arg\min_{u \in \mathbb{R}^d} Q_{\lambda_r}(u) = -(\nabla^2 F(x) + \lambda_r\mathbf{I})^{-1}\nabla F(x).$$

*Also, $\lambda_r$ is decreasing in $r$, and if $\lambda_r \neq 0$ then $\|u_r^*\| = r$.*

**Lemma 8.** *For* $r \leq r^*(0)$, $\lambda_r \in \left[0, \frac{\|b\|}{r}\right]$.

*Proof.* By Lemma 7, $\lambda \geq 0$. If $\lambda > 0$, then since $\nabla^2 F(x) \succeq 0$ ( by convexity of $F$), we have that

$$r = \|u_\lambda^*\| = \|(\nabla^2 F(x) + \lambda \mathbf{I})^{-1} \nabla F(x)\| \leq \frac{\|\nabla F(x)\|}{\lambda}. \tag{16}$$

Rearranging completes the proof. $\square$

**Lemma 9.** *For any* $r \geq 0$ *let* $\lambda \geq \gamma \geq 0$ *such that* $r^*(\lambda) = r$ *and* $r^*(\gamma) = 2r$. *Then*

$$\mathbb{E}\big[Q_\gamma(u) - Q_\gamma(u_\gamma^*)\big] \leq \epsilon \implies \mathbb{E}[Q(u) - Q(u_\lambda^*)] \leq \epsilon.$$

*Furthermore, for any* $y$ *and* $\gamma$,

$$\mathbb{E}\big[Q_\gamma(u) - Q_\gamma(u_\gamma^*)\big] \leq \epsilon \implies \mathbb{E}[Q(u) - Q(y)] \leq \epsilon + \frac{\gamma \|y\|^2}{2}.$$

*Proof.* For any $u$,

$$\mathbb{E}[Q_\gamma(u) - Q(u_\lambda^*)] = \mathbb{E}\big[Q_\gamma(u) - Q_\gamma(u_\gamma^*)\big] + Q_\gamma(u_\gamma^*) - Q_\gamma(u_\lambda^*) + \frac{\gamma}{2}\|u_\lambda^*\|^2 \tag{17}$$

$$\leq \epsilon + Q_\gamma(u_\gamma^*) - Q_\gamma(u_\lambda^*) + \frac{\gamma}{2}\|u_\lambda^*\|^2. \tag{18}$$

By the $\gamma$-strong convexity of $Q_\gamma$, the fact that either $\gamma = 0$ or $\|u_\lambda^*\| = r$ and $\|u_\gamma^*\| = 2r$, and the reverse triangle inequality,

$$Q_\gamma(u_\gamma^*) - Q_\gamma(u_\lambda^*) + \frac{\gamma}{2}\|u_\lambda^*\|^2 \leq -\frac{\gamma}{2}\big\|u_\gamma^* - u_\lambda^*\big\|^2 + \frac{\gamma r^2}{2} \leq -\frac{\gamma(2r-r)^2}{2} + \frac{\gamma r^2}{2} = 0. \tag{19}$$

Combining (19) with (18) completes the first part of the proof. For the second part, we observe that

$$\mathbb{E}\left[Q(u) + \frac{\gamma}{2}\|u\|^2 - Q(y)\right] = \mathbb{E}\big[Q_\gamma(u) - Q_\gamma(u_\gamma^*)\big] + Q_\gamma(u_\gamma^*) - Q_\gamma(y) + \frac{\gamma}{2}\|y\|^2 \leq \epsilon + \frac{\gamma\|y\|^2}{2}.$$
$$\tag{20}$$

Above, we used that $Q_\lambda(u_\lambda^*) = \min_u Q_\lambda(u) \leq Q_\lambda(y)$. $\square$

**Lemma 10.** *Let* $\lambda \geq \gamma \geq 0$. *Then ,*

$$\frac{\lambda}{\gamma} \geq \frac{\|u_\gamma^*\|}{\|u_\lambda^*\|}.$$

*Proof.* By the definition of $u_\gamma^* = \arg\min_u Q_\gamma(u)$ and $u_\lambda^* = \arg\min_u Q_\lambda(u)$ and the $\gamma$- and $\lambda$-strong convexity of $Q_\gamma$ and $Q_\lambda$, respectively,

$$\frac{\lambda - \gamma}{2}\left(\big\|u_\gamma^*\big\|^2 - \big\|u_\lambda^*\big\|^2\right) = Q_\gamma(u_\lambda^*) - Q_\gamma(u_\gamma^*) + Q_\lambda(u_\gamma^*) - Q_\lambda(u_\lambda^*) \geq \frac{\lambda + \gamma}{2}\big\|u_\lambda^* - u_\gamma^*\big\|^2. \tag{21}$$

Therefore, by rearranging and applying the reverse triangle inequality

$$\frac{\lambda - \gamma}{\lambda + \gamma} \geq \frac{\big\|u_\gamma^* - u_\lambda^*\big\|^2}{\big\|u_\gamma^*\big\|^2 - \big\|u_\lambda^*\big\|^2} \geq \frac{\big(\big\|u_\gamma^*\big\| - \big\|u_\lambda^*\big\|\big)^2}{\big\|u_\gamma^*\big\|^2 - \big\|u_\lambda^*\big\|^2} = \frac{\big\|u_\gamma^*\big\| - \big\|u_\lambda^*\big\|}{\big\|u_\gamma^*\big\| + \big\|u_\lambda^*\big\|}. \tag{22}$$

By Lemma 7, since $\lambda \geq \gamma$, $\big\|u_\gamma^*\big\| \geq \|u_\lambda^*\|$. Therefore, rearranging this inequality completes the proof. $\square$

**Lemma 11.** *Let* $X \sim Binomial(C, p)$ *with* $p \geq \frac{3}{4}$. *Then,*

$$\mathbb{P}\left(X > \frac{C}{2}\right) \geq 1 - \exp\left(-\frac{C}{8}\right).$$

*Proof.* This is a simple application of Hoeffding's inequality:

$$\mathbb{P}\left(\frac{1}{C}X \leq \frac{1}{2} \leq p - \frac{1}{4}\right) \leq \exp\left(-\frac{C}{8}\right). \tag{23}$$

$\square$

**Lemma 12.** *Let $x \in \mathbb{R}^d$ be as input to Algorithm 2, let $Q_{\lambda^{(i)}}(u) = \frac{1}{2}u^\top\big(\delta(5\bar{r})\nabla^2 F(x) + \lambda^{(i)}\big)u + \nabla F(x)^\top u$, let $Q^*_{\lambda^{(i)}} := \min_u Q_{\lambda^{(i)}}(u)$, and for each $i$, let $\tilde{u}^{(i,1)}, \ldots, \tilde{u}^{(i,C)}$ be computed as in Algorithm 2 such that*

$$\mathbb{E}Q_{\lambda^{(i)}}(\tilde{u}^{(i,c)}) - Q^*_{\lambda^{(i)}} \le \epsilon(\lambda^{(i)}) \le \frac{\lambda^{(i)}\big(r^*(\lambda^{(i)})^2 + \bar{r}^2\big)}{800} \; .$$

*Then,*

$$r^*(\lambda^{(i)}) \in [2\bar{r},\, 3\bar{r}] \implies \mathbb{P}\left(\sum_{c=1}^C \mathbb{1}\left\{\left\|\tilde{u}^{(i,c)}\right\| \in \left[\frac{3}{2}\bar{r},\, \frac{7}{2}\bar{r}\right]\right\} > \frac{C}{2}\right) \ge 1 - \exp\left(-\frac{C}{8}\right)$$

$$r^*(\lambda^{(i)}) \notin [\bar{r},\, 4\bar{r}] \implies \mathbb{P}\left(\sum_{c=1}^C \mathbb{1}\left\{\left\|\tilde{u}^{(i,c)}\right\| \in \left[\frac{3}{2}\bar{r},\, \frac{7}{2}\bar{r}\right]\right\} \le \frac{C}{2}\right) \le \exp\left(-\frac{C}{8}\right)$$

$$r^*(\lambda^{(i)}) < 2\bar{r} \implies \mathbb{P}\left(\sum_{c=1}^C \mathbb{1}\left\{\left\|\tilde{u}^{(i,c)}\right\| \le \frac{5}{2}\bar{r}\right\} > \frac{C}{2}\right) \ge 1 - \exp\left(-\frac{C}{8}\right)$$

$$r^*(\lambda^{(i)}) > 3\bar{r} \implies \mathbb{P}\left(\sum_{c=1}^C \mathbb{1}\left\{\left\|\tilde{u}^{(i,c)}\right\| > \frac{5}{2}\bar{r}\right\} > \frac{C}{2}\right) \ge 1 - \exp\left(-\frac{C}{8}\right) \; .$$

*Proof.* By the $\lambda^{(i)}$-strong convexity of $Q_{\lambda^{(i)}}$, for each $c$,

$$\mathbb{E}\left[\frac{\lambda^{(i)}}{2}\|\tilde{u}^{(i,c)} - u^*_{\lambda^{(i)}}\|^2\right] \le \mathbb{E}Q_{\lambda^{(i)}}(\tilde{u}^{(i,c)}) - Q^*_{\lambda^{(i)}} \le \epsilon(\lambda^{(i)}). \tag{24}$$

Therefore, by Markov's inequality, for each $c$

$$\mathbb{P}\left(\|\tilde{u}^{(i,c)} - u^*_{\lambda^{(i)}}\|^2 \ge \frac{1}{100}r^*(\lambda^{(i)})^2 + \frac{1}{100}\bar{r}^2\right) \le \frac{200\epsilon(\lambda^{(i)})}{\lambda^{(i)}\big(r^*(\lambda^{(i)})^2 + \bar{r}^2\big)} \le \frac{1}{4} \; . \tag{25}$$

Furthermore, by the reverse triangle inequality,

$$\left|\left\|\tilde{u}^{(i,c)}\right\| - r^*(\lambda^{(i)})\right| \le \|\tilde{u}^{(i,c)} - u^*_{\lambda^{(i)}}\| < \sqrt{\frac{1}{100}r^*(\lambda^{(i)})^2 + \frac{1}{100}\bar{r}^2} \le \frac{r^*(\lambda^{(i)}) + \bar{r}}{10} \; . \tag{26}$$

Therefore, for each $c$

$$\mathbb{P}\left(\frac{9}{10}r^*(\lambda^{(i)}) - \frac{1}{10}\bar{r} \le \|\tilde{u}^{(i,c)}\| \le \frac{11}{10}r^*(\lambda^{(i)}) + \frac{1}{10}\bar{r}\right) \ge \frac{3}{4} \; . \tag{27}$$

We now consider several cases:

If $r^*(\lambda^{(i)}) \in [2\bar{r},\, 3\bar{r}]$, then

$$\mathbb{P}\left(\|\tilde{u}^{(i,c)}\| \in \left[\frac{3}{2}\bar{r},\, \frac{7}{2}\bar{r}\right]\right) \tag{28}$$

$$\ge \mathbb{P}\left(\frac{17}{10}\bar{r} \le \frac{9}{10}r^*(\lambda^{(i)}) - \frac{1}{10}\bar{r} \le \|\tilde{u}^{(i,c)}\| \le \frac{11}{10}r^*(\lambda^{(i)}) + \frac{1}{10}\bar{r} \le \frac{34}{10}\bar{r}\right) \ge \frac{3}{4}. \tag{29}$$

Therefore, by Lemma 11

$$r^*(\lambda^{(i)}) \in [2\bar{r},\, 3\bar{r}] \implies \mathbb{P}\left(\sum_{c=1}^C \mathbb{1}\left\{\left\|\tilde{u}^{(i,c)}\right\| \in \left[\frac{3}{2}\bar{r},\, \frac{7}{2}\bar{r}\right]\right\} > \frac{C}{2}\right) \ge 1 - \exp\left(-\frac{C}{8}\right). \tag{30}$$

If $r^*(\lambda^{(i)}) > 4\bar{r}$, then

$$\mathbb{P}\left(\|\tilde{u}^{(i,c)}\| \notin \left[\frac{3}{2}\bar{r},\, \frac{7}{2}\bar{r}\right]\right) \ge \mathbb{P}\left(\frac{7}{2}\bar{r} \le \frac{9}{10}r^*(\lambda^{(i)}) - \frac{1}{10}\bar{r} \le \|\tilde{u}^{(i,c)}\|\right) \ge \frac{3}{4}. \tag{31}$$

Similarly, if $r^*(\lambda^{(i)}) < \bar{r}$, then

$$\mathbb{P}\left(\|\tilde{u}^{(i,c)}\| \notin \left[\frac{3}{2}\bar{r},\, \frac{7}{2}\bar{r}\right]\right) \geq \mathbb{P}\left(\|\tilde{u}^{(i,c)}\| \leq \frac{11}{10}r^*(\lambda^{(i)}) + \frac{1}{10}\bar{r} \leq \frac{12}{10}\bar{r}\right) \geq \frac{3}{4}. \tag{32}$$

Therefore, by Lemma 11,

$$r^*(\lambda^{(i)}) \notin [\bar{r},\, 4\bar{r}] \implies \mathbb{P}\left(\sum_{c=1}^{C} \mathbb{1}\left\{\left\|\tilde{u}^{(i,c)}\right\| \in \left[\frac{3}{2}\bar{r},\, \frac{7}{2}\bar{r}\right]\right\} \leq \frac{C}{2}\right) \leq \exp\left(-\frac{C}{8}\right). \tag{33}$$

If $r^*(\lambda^{(i)}) < 2\bar{r}$, then

$$\mathbb{P}\left(\|\tilde{u}^{(i,c)}\| \leq \frac{5}{2}\bar{r}\right) \geq \mathbb{P}\left(\|\tilde{u}^{(i,c)}\| \leq \frac{11}{10}r^*(\lambda^{(i)}) + \frac{1}{10}\bar{r} \leq \frac{23}{10}\bar{r}\right) \geq \frac{3}{4}. \tag{34}$$

Therefore, by Lemma 11

$$r^*(\lambda^{(i)}) < 2\bar{r} \implies \mathbb{P}\left(\sum_{c=1}^{C} \mathbb{1}\left\{\left\|\tilde{u}^{(i,c)}\right\| \leq \frac{5}{2}\bar{r}\right\} > \frac{C}{2}\right) \geq 1 - \exp\left(-\frac{C}{8}\right). \tag{35}$$

Finally, if $r^*(\lambda^{(i)}) > 3\bar{r}$, then

$$\mathbb{P}\left(\|\tilde{u}^{(i,c)}\| > \frac{5}{2}\bar{r}\right) \geq \mathbb{P}\left(\frac{26}{10}\bar{r} \leq \frac{9}{10}r^*(\lambda^{(i)}) - \frac{1}{10}\bar{r} \leq \|\tilde{u}^{(i,c)}\|\right) \geq \frac{3}{4}. \tag{36}$$

Therefore, by Lemma 11

$$r^*(\lambda^{(i)}) > 3\bar{r} \implies \mathbb{P}\left(\sum_{c=1}^{C} \mathbb{1}\left\{\left\|\tilde{u}^{(i,c)}\right\| > \frac{5}{2}\bar{r}\right\} > \frac{C}{2}\right) \geq 1 - \exp\left(-\frac{C}{8}\right). \tag{37}$$

This completes the proof. $\qquad\square$

**Lemma 13.** *Let $N \geq 1 + \frac{5}{2}\log\frac{\|\nabla F(x)\|}{3\bar{r}\lambda}$. Then, either $\bar{\lambda} \geq \lambda_{3\bar{r}}$ or $\exists \lambda \in \Lambda_1$ with $r^*(\lambda) \in [2\bar{r}, 3\bar{r}]$.*

*Proof.* In the case that $\bar{\lambda} \leq \lambda_{2\bar{r}}$, by Lemma 8, with $N \geq 1 + \frac{5}{2}\log\frac{\|\nabla F(x)\|}{3\bar{r}\lambda}$,

$$\bar{\lambda} \leq \lambda_{3\bar{r}} \leq \frac{\|\nabla F(x)\|}{3\bar{r}} \leq \bar{\lambda}c^{N-1}. \tag{38}$$

So, $\lambda_{3\bar{r}}$ is between the largest and smallest elements in $\Lambda_1$. It follows that for some $1 \leq n \leq N$,

$$\bar{\lambda}c^{n-1} \leq \lambda_{3\bar{r}} \leq \bar{\lambda}c^n. \tag{39}$$

Therefore, by Lemma 10,

$$\frac{3}{2} = \frac{\bar{\lambda}\left(\frac{3}{2}\right)^n}{\bar{\lambda}\left(\frac{3}{2}\right)^{n-1}} \geq \frac{\bar{\lambda}\left(\frac{3}{2}\right)^n}{\lambda_{3\bar{r}}} \geq \frac{3\bar{r}}{r^*\left(\bar{\lambda}\left(\frac{3}{2}\right)^n\right)} \implies r^*\left(\bar{\lambda}\left(\frac{3}{2}\right)^n\right) \geq 2\bar{r} \tag{40}$$

Finally, by Lemma 7, $r^*\left(\bar{\lambda}\left(\frac{3}{2}\right)^n\right) \leq 3\bar{r}$, so $r^*\left(\bar{\lambda}\left(\frac{3}{2}\right)^n\right) \in [2\bar{r}, 3\bar{r}]$ as claimed. $\qquad\square$

**Lemma 14.** *Let $\lambda$ satisfy $r^*(\lambda) \leq 4\bar{r}$ and let $\tilde{u}$ be chosen so that*

$$\mathbb{E}Q_\lambda(\tilde{u}) - Q_\lambda^* \leq \epsilon(\lambda) := \frac{\lambda(r^*(\lambda)^2 + \bar{r}^2)}{800}.$$

*Then if $r^*(\lambda) \geq \bar{r}$,*

$$\mathbb{E}Q\left(\min\left\{1,\, \frac{5\bar{r}}{\|\tilde{u}\|}\right\}\tilde{u}\right) - \min_{u:\|u\|\leq\frac{1}{2}\bar{r}} \leq \frac{1}{2}\left(Q(0) - \min_{u:\|u\|\leq\frac{1}{2}\bar{r}} Q(u)\right) + \epsilon(\lambda).$$

*Otherwise,*

$$\mathbb{E}Q\left(\min\left\{1,\, \frac{5\bar{r}}{\|\tilde{u}\|}\right\}\tilde{u}\right) - \min_{u:\|u\|\leq\frac{1}{2}\bar{r}} \leq \frac{1}{2}\left(Q(0) - \min_{u:\|u\|\leq\frac{1}{2}\bar{r}} Q(u)\right) + \epsilon(\lambda) + \frac{\lambda\bar{r}^2}{8}.$$

*Proof.* First,

$$\mathbb{E}Q\left(\min\left\{1, \frac{5\bar{r}}{\|\tilde{u}\|}\right\}\tilde{u}\right) - \min_{u:\|u\|\leq\frac{1}{2}\bar{r}} Q(u)$$

$$= \mathbb{E}Q\left(\left(1 - \min\left\{1, \frac{5\bar{r}}{\|\tilde{u}\|}\right\}\right)0 + \min\left\{1, \frac{5\bar{r}}{\|\tilde{u}\|}\right\}\tilde{u}\right) - \min_{u:\|u\|\leq\frac{1}{2}\bar{r}} Q(u) \tag{41}$$

$$\leq \mathbb{E}\left[\left(1 - \min\left\{1, \frac{5\bar{r}}{\|\tilde{u}\|}\right\}\right)\left(Q(0) - \min_{u:\|u\|\leq\frac{1}{2}\bar{r}} Q(u)\right)\right. \tag{42}$$

$$\left. + \min\left\{1, \frac{5\bar{r}}{\|\tilde{u}\|}\right\}\left(Q(\tilde{u}) - \min_{u:\|u\|\leq\frac{1}{2}\bar{r}} Q(u)\right)\right] \tag{43}$$

$$\leq \mathbb{E}\left[\frac{\|\tilde{u}\|}{\|\tilde{u}\| + 5\bar{r}}\right]\left(Q(0) - \min_{u:\|u\|\leq\frac{1}{2}\bar{r}} Q(u)\right) + \mathbb{E}Q(\tilde{u}) - \min_{u:\|u\|\leq\frac{1}{2}\bar{r}} Q(u) \tag{44}$$

$$\leq \frac{\mathbb{E}\|\tilde{u}\|}{\mathbb{E}\|\tilde{u}\| + 5\bar{r}}\left(Q(0) - \min_{u:\|u\|\leq\frac{1}{2}\bar{r}} Q(u)\right) + \mathbb{E}Q(\tilde{u}) - \min_{u:\|u\|\leq\frac{1}{2}\bar{r}} Q(u). \tag{45}$$

For the first inequality we used the convexity of $Q$, and for the final inequality we used Jensen's inequality on the concave function $x \mapsto \frac{x}{x+5\bar{r}}$. Next, we bound

$$\mathbb{E}\|\tilde{u}\| \leq r^*(\lambda) + \mathbb{E}\|\tilde{u} - u_\lambda^*\| \tag{46}$$

$$\leq 4\bar{r} + \sqrt{\mathbb{E}\|\tilde{u} - u_\lambda^*\|^2} \tag{47}$$

$$\leq 4\bar{r} + \sqrt{\frac{2}{\lambda}\mathbb{E}Q_\lambda(\tilde{u}) - Q_\lambda^*} \tag{48}$$

$$\leq 4\bar{r} + \sqrt{\frac{2\epsilon(\lambda)}{\lambda}} \tag{49}$$

$$\leq 4\bar{r} + \sqrt{\frac{r^*(\lambda)^2 + \bar{r}^2}{400}} \tag{50}$$

$$\leq 4\bar{r} + \sqrt{\frac{17\bar{r}^2}{400}} \tag{51}$$

$$< 5\bar{r}. \tag{52}$$

Therefore,

$$\mathbb{E}Q\left(\min\left\{1, \frac{5\bar{r}}{\|\tilde{u}\|}\right\}\tilde{u}\right) - \min_{u:\|u\|\leq\frac{1}{2}\bar{r}} Q(u) \leq \frac{1}{2}\left(Q(0) - \min_{u:\|u\|\leq\frac{1}{2}\bar{r}} Q(u)\right) + \mathbb{E}Q(\tilde{u}) - \min_{u:\|u\|\leq\frac{1}{2}\bar{r}} Q(u). \tag{53}$$

If $r^*(\lambda) \geq \bar{r}$, then by the first part of Lemma 9

$$\mathbb{E}Q(\tilde{u}) - \min_{u:\|u\|\leq\frac{1}{2}\bar{r}} Q(u) \leq \mathbb{E}Q(\tilde{u}) - \min_{u:\|u\|\leq\frac{1}{2}r^*(\lambda)} Q(u) \leq \epsilon(\lambda). \tag{54}$$

Otherwise, by the second part of Lemma 9

$$\mathbb{E}Q(\tilde{u}) - \min_{u:\|u\|\leq\frac{1}{2}\bar{r}} Q(u) = \mathbb{E}Q(\tilde{u}) - Q(u_{\lambda_{\frac{1}{2}\bar{r}}}^*) \leq \epsilon(\lambda) + \frac{\lambda\|u_{\lambda_{\frac{1}{2}\bar{r}}}^*\|^2}{2} = \epsilon(\lambda) + \frac{\lambda\bar{r}^2}{8}. \tag{55}$$

$$\square$$

We are now ready to prove Lemma 3.

**Lemma 3.** *Let $F$ satisfy Assumption 2, let $x$ be as input to* CONSTRAINED-QUADRATIC-SOLVER *(Algorithm 2), let $\bar{\xi}$ be as in Table 4, define*

$$Q(u) := \frac{\bar{\xi}}{2}u^\top\nabla^2 F(x)u + \nabla F(x)^\top u, \qquad Q_\lambda(u) := \frac{1}{2}u^\top(\bar{\xi}\nabla^2 F(x) + \lambda\mathbf{I})u + \nabla F(x)^\top u,$$

$$u_\lambda^* := \arg\min_u Q_\lambda(u), \qquad r^*(\lambda) := \|u_\lambda^*\|,$$

*and let $\lambda_r$ denote, for any $r > 0$, the value such that $r^*(\lambda_r) = r$. Let $\hat{u}$ be the output of Algorithm 2 for hyperparameters $\bar{r}$, $\bar{\xi}$, $\bar{\lambda}$, $N$ and $C$ as in Table 4, and suppose the output $\tilde{u}_\lambda$ of* REGULARIZED-QUADRATIC-SOLVER$(x, \lambda)$ *satisfies for all $\lambda \geq \bar{\lambda}$ that*

$$\mathbb{E}Q_\lambda(\tilde{u}_\lambda) - \min_u Q_\lambda(u) \leq \epsilon(\lambda) := \frac{\lambda(r^*(\lambda)^2 + \bar{r}^2)}{800} \ .$$

*Then $\|\hat{u}\| \leq 5\bar{r}$ and*

$$\mathbb{E}Q(\hat{u}) - \min_{u:\|u\|\leq\frac{1}{2}\bar{r}} Q(u) \leq \frac{3}{4}\left(Q(0) - \min_{u:\|u\|\leq\frac{1}{2}\bar{r}} Q(u)\right) + \epsilon(\lambda_{4\bar{r}}) + \frac{\bar{\lambda}\bar{r}^2}{4} \ .$$

*Proof.* First, we note that $|\Lambda_1| = N$, and in each iteration either the algorithm terminates or $\Lambda_{i+1}$ is chosen such that $|\Lambda_{i+1}| \leq \frac{1}{2}|\Lambda_i|$. Therefore, the algorithm terminates after at most $\lceil\log_2 N\rceil$ iterations.

By Lemma 13, either $\bar{\lambda} \geq \lambda_{3\bar{r}}$ or there exists $\lambda \in \Lambda_1$ for some $\lambda$ such that $r^*(\lambda) \in [2\bar{r}, 3\bar{r}]$. If such a $\lambda$ exists, we denote this (not necessarily unique) value $\lambda^*$.

By Lemma 12 and the union bound, the following holds for all iteration $i = 1, 2, \ldots$ with probability at least $1 - 2\lceil\log_2 N\rceil \exp\left(-\frac{C}{8}\right)$:

$$r^*(\lambda^{(i)}) \in [2\bar{r}, 3\bar{r}] \implies \sum_{c=1}^{C} \mathbb{1}\left\{\left\|\tilde{u}^{(i,c)}\right\| \in \left[\frac{3}{2}\bar{r}, \frac{7}{2}\bar{r}\right]\right\} > \frac{C}{2} \tag{56}$$

$$r^*(\lambda^{(i)}) \notin [\bar{r}, 4\bar{r}] \implies \sum_{c=1}^{C} \mathbb{1}\left\{\left\|\tilde{u}^{(i,c)}\right\| \in \left[\frac{3}{2}\bar{r}, \frac{7}{2}\bar{r}\right]\right\} \leq \frac{C}{2} \tag{57}$$

$$r^*(\lambda^{(i)}) < 2\bar{r} \implies \sum_{c=1}^{C} \mathbb{1}\left\{\left\|\tilde{u}^{(i,c)}\right\| \leq \frac{5}{2}\bar{r}\right\} > \frac{C}{2} \tag{58}$$

$$r^*(\lambda^{(i)}) > 3\bar{r} \implies \sum_{c=1}^{C} \mathbb{1}\left\{\left\|\tilde{u}^{(i,c)}\right\| > \frac{5}{2}\bar{r}\right\} > \frac{C}{2} \ . \tag{59}$$

For most of the rest of the proof, we condition on this event, which we denote $E$.

Under $E$, if $\lambda^{(i)} = \lambda^*$, then the algorithm will terminate on Line 2, and even if $\lambda^{(i)} \neq \lambda^*$, if the algorithm terminates on Line 2, then $\lambda^{(i)} \in [\bar{r}, 4\bar{r}]$. In either case, by the first part of Lemma 14

$$\mathbb{E}Q(\hat{u}) - \min_{u:\|u\|\leq\frac{1}{2}\bar{r}} Q(u) \leq \frac{1}{2}\left(Q(0) - \min_{u:\|u\|\leq\frac{1}{2}\bar{r}} Q(u)\right) + \epsilon(\lambda^{(i)}). \tag{60}$$

Finally, since $r^*(\lambda^{(i)}) \leq 4\bar{r}$, $\lambda^{(i)} \geq \lambda_{4\bar{r}}$.

If the algorithm instead updates

$$\Lambda_{i+1} = \Lambda_i \setminus \left\{\lambda \in \Lambda_i : \lambda \geq \lambda^{(i)}\right\} \tag{61}$$

as on Line 2, then conditioned on $E$,

$$\sum_{c=1}^{C} \mathbb{1}\left\{\left\|\tilde{u}^{(i,c)}\right\| \in \left[\frac{3}{2}\bar{r}, \frac{7}{2}\bar{r}\right]\right\} \leq \frac{C}{2} < \sum_{c=1}^{C} \mathbb{1}\left\{\left\|\tilde{u}^{(i,c)}\right\| \leq \frac{5}{2}\bar{r}\right\} \tag{62}$$

implies $\lambda^{(i)} < 2\bar{r}$. By Lemma 7, since $r^*(\lambda^*) \geq 2\bar{r} > r^*(\lambda^{(i)})$, $\lambda^* < \lambda^{(i)}$ and therefore $\lambda^* \in \Lambda_{i+1}$.

If the algorithm instead updates

$$\Lambda_{i+1} = \Lambda_i \setminus \left\{\lambda \in \Lambda_i : \lambda \leq \lambda^{(i)}\right\} \tag{63}$$

as on Line 2, then conditioned on $E$

$$\sum_{c=1}^{C} \mathbb{1}\left\{\left\|\tilde{u}^{(i,c)}\right\| \in \left[\frac{3}{2}\bar{r}, \frac{7}{2}\bar{r}\right]\right\} \leq \frac{C}{2} < \sum_{c=1}^{C} \mathbb{1}\left\{\left\|\tilde{u}^{(i,c)}\right\| > \frac{5}{2}\bar{r}\right\} \tag{64}$$

implies $\lambda^{(i)} > 3\bar{r}$. By Lemma 7, since $r^*(\lambda^*) \leq 3\bar{r} < r^*(\lambda^{(i)})$, we have $\lambda^* > \lambda^{(i)}$, and therefore $\lambda^* \in \Lambda_{i+1}$.

Finally, $E$ implies that the algorithm will never reach Line 2.

Therefore, conditioned on $E$, if $\lambda^* \in \Lambda_1$, then the algorithm will never remove $\lambda^*$ from the set of $\lambda$'s under consideration, and it will eventually terminate on Line 2 by returning a point such that

$$\mathbb{E}Q(\hat{u}) - \min_{u:\|u\|\leq\frac{1}{2}\bar{r}} Q(u) \leq \frac{1}{2}\left(Q(0) - \min_{u:\|u\|\leq\frac{1}{2}\bar{r}} Q(u)\right) + \epsilon(\lambda^{(i)}). \tag{65}$$

Otherwise, if such a $\lambda^*$ does not exist and the algorithm does not terminate on Line 2, then it terminates on Line 2 using $\bar{\lambda} \geq \lambda_{3\bar{r}} \geq \lambda_{4\bar{r}}$, which implies $r^*(\bar{\lambda}) \leq 4\bar{r}$. Therefore, by the second part of Lemma 14,

$$\mathbb{E}Q(\tilde{u}) - \min_{u:\|u\|\leq\frac{1}{2}\bar{r}} Q(u) \leq \frac{1}{2}\left(Q(0) - \min_{u:\|u\|\leq\frac{1}{2}\bar{r}} Q(u)\right) + \epsilon(\bar{\lambda}) + \frac{\bar{\lambda}\bar{r}^2}{8}. \tag{66}$$

Therefore since $\epsilon(\lambda)$ is decreasing in $\lambda$, conditioned on $E$ the algorithm's output satisfies

$$\mathbb{E}Q(\tilde{u}) - \min_{u:\|u\|\leq\frac{1}{2}\bar{r}} Q(u) \leq \frac{1}{2}\left(Q(0) - \min_{u:\|u\|\leq\frac{1}{2}\bar{r}} Q(u)\right) + \epsilon(\lambda_{4\bar{r}}) + \frac{\bar{\lambda}\bar{r}^2}{8}. \tag{67}$$

We now consider the case that $E$ does not hold. In this case, the algorithm's output is guaranteed to have norm at most $5\bar{r}$. Therefore,

$$Q(\hat{u}) = \frac{\delta(5\bar{r})}{2}\hat{u}^\top \nabla^2 F(x)\hat{u} + \nabla F(x)^\top \hat{u} \leq \frac{25\delta(5\bar{r})\|\nabla^2 F(x)\|_2\bar{r}^2}{2} + 5\bar{r}\|\nabla F(x)\| \tag{68}$$

$$= Q(0) + \frac{25\delta(5\bar{r})\|\nabla^2 F(x)\|_2\bar{r}^2}{2} + 5\bar{r}\|\nabla F(x)\|. \tag{69}$$

Therefore, conditioned on $\neg E$,

$$Q(\hat{u}) - \min_{u:\|u\|\leq\frac{1}{2}\bar{r}} Q(u) \leq Q(0) - \min_{u:\|u\|\leq\frac{1}{2}\bar{r}} Q(u) + \frac{25\delta(5\bar{r})\|\nabla^2 F(x)\|_2\bar{r}^2}{2} + 5\bar{r}\|\nabla F(x)\|. \tag{70}$$

We conclude by noting that

$$\mathbb{E}Q(\hat{u}) - \min_{u:\|u\|\leq\frac{1}{2}\bar{r}} Q(u)$$

$$= \mathbb{E}[Q(\hat{u}) \mid E]\mathbb{P}(E) + \mathbb{E}[Q(\hat{u}) \mid \neg E]\mathbb{P}(\neg E) - \min_{u:\|u\|\leq\frac{1}{2}\bar{r}} Q(u) \tag{71}$$

$$\leq \frac{1+\mathbb{P}(\neg E)}{2}\left(Q(0) - \min_{u:\|u\|\leq\frac{1}{2}\bar{r}} Q(u)\right) + \epsilon(\lambda_{4\bar{r}}) + \frac{\bar{\lambda}\bar{r}^2}{8} \tag{72}$$

$$+ \left(\frac{25\delta(5\bar{r})\|\nabla^2 F(x)\|_2\bar{r}^2}{2} + 5\bar{r}\|\nabla F(x)\|\right)\mathbb{P}(\neg E) \tag{73}$$

$$\leq \frac{1+2\lceil\log_2 N\rceil\exp\left(-\frac{C}{8}\right)}{2}\left(Q(0) - \min_{u:\|u\|\leq\frac{1}{2}\bar{r}} Q(u)\right) + \epsilon(\lambda_{4\bar{r}}) + \frac{\bar{\lambda}\bar{r}^2}{8}$$

$$+ \left(\frac{25\delta(5\bar{r})\|\nabla^2 F(x)\|_2\bar{r}^2}{2} + 5\bar{r}\|\nabla F(x)\|\right) \cdot 2\lceil\log_2 N\rceil\exp\left(-\frac{C}{8}\right) \tag{74}$$

Using the fact that

$$C \geq 8\log\left(\lceil\log_2 N\rceil\left(4 + \frac{8\delta(5\bar{r})\|\nabla^2 F(x)\|_2}{\bar{\lambda}} + \frac{80\|\nabla F(x)\|}{\bar{\lambda}\bar{r}}\right)\right) \tag{75}$$

we conclude

$$\mathbb{E}Q(\hat{u}) - \min_{u:\|u\|\leq\frac{1}{2}\bar{r}} Q(u) \leq \frac{3}{4}\left(Q(0) - \min_{u:\|u\|\leq\frac{1}{2}\bar{r}} Q(u)\right) + \epsilon(\lambda_{4\bar{r}}) + \frac{\bar{\lambda}\bar{r}^2}{4}. \tag{76}$$

This completes the proof. $\qquad\square$

## D  Proof of Lemma 4

**Lemma 4.** *Let $F$ satisfy Assumption 2, let $x \in \mathbb{R}^d$, $\lambda \in \mathbb{R}^+$ be as input to* REGULARIZED-QUADRATIC-SOLVER *(Algorithm 3), let $Q_\lambda(u) = \frac{1}{2}u^\top(\bar{\xi}\nabla^2 F(x) + \lambda\mathbf{I})u + \nabla F(x)^\top u$, let $Q_\lambda^* := \min_u Q_\lambda(u)$, let $u^* := \arg\min_u Q_\lambda(u)$, and let stochastic first-order and stochastic Hessian-vector product oracles for $F$, as defined in Assumptions 1 and 2, respectively, be available for each call to* REGULARIZED-QUADRATIC-GRADIENT-ACCESS *(Algorithm 4), for either Case 1* (Different-Samples) *or Case 2* (Same-Sample)*. Let $\hat{u}$, as output by Algorithm 3, be a weighted average of the iterates of $M$ independent runs of SGD with stepsizes $\eta_0(\lambda), \ldots, \eta_{K-1}(\lambda)$, i.e., $\hat{u} = \frac{1}{M\sum_{k=0}^{K-1} w_k}\sum_{m=1}^M \sum_{k=0}^{K-1} w_k u_k^m$. Then, for both Cases 1 and 2,*

$$\mathbb{E}Q_\lambda(\hat{u}) - Q_\lambda^* \leq \begin{cases} 2\max\left\{\bar{\xi}H + \lambda, \frac{\rho^2}{\lambda}\right\}\|u^*\|^2 \min\left\{\frac{1}{K}, \exp\left(-\frac{K+1}{4}\min\left\{\frac{\lambda}{\bar{\xi}H+\lambda}, \frac{\lambda^2}{\rho^2}\right\}\right)\right\} \\ \qquad\qquad + \frac{2(\sigma^2 + \rho^2\|u^*\|^2)}{\lambda MK} \;\; \text{if } K \leq \frac{2}{\lambda}\max\left\{\bar{\xi}H + \lambda, \frac{\rho^2}{\lambda}\right\} \\[2ex] 96\lambda\|u^*\|^2 \exp\left(-\frac{K}{8}\min\left\{\frac{\lambda}{\bar{\xi}H+\lambda}, \frac{\lambda^2}{\rho^2}\right\}\right) + \frac{96(\sigma^2 + \rho^2\|u^*\|^2)}{\lambda MK} \\ \qquad\qquad\qquad\qquad\qquad \text{if } K > \frac{2}{\lambda}\max\left\{\bar{\xi}H + \lambda, \frac{\rho^2}{\lambda}\right\}. \end{cases}$$

*Proof.* We will use $\bar{u}_k = \frac{1}{M}\sum_{m=1}^M u_k^m$ to denote the average of each independent run of SGD's $k^{\text{th}}$ iterate. This quantity is never explicitly computed until the end, but we can nevertheless use it for our analysis. Likewise, we will use $\bar{\gamma}_k = \frac{1}{M}\sum_{m=1}^M \gamma(u_k^m; z_k^m, z_k^{'m})$ to denote the average of the stochastic gradients of $Q_\lambda(u)$ computed at time $k$, whereby we recall that $\gamma(u_k^m; z_k^m, z_k^{'m}) := \bar{\xi}h(x, u_k^m; z_k^{'m}) + \lambda u_k^m + g(x; z_k^m)$ as defined in Algorithm 3, along with the requisite oracle access as described in Algorithm 4.

We also have that, by Assumptions 1(c) and 2(b), for Case 1:

$$\mathbb{E}_{z_k^m, z_k^{'m}}[\gamma(u_k^m; z_k^m, z_k^{'m})] = \nabla Q_\lambda(u) \text{ and } \mathbb{E}_{z_k^m, z_k^{'m}}\left\|\gamma(u_k^m; z_k^m, z_k^{'m}) - \nabla Q_\lambda(u_k^m)\right\|^2 \leq \sigma^2 + \rho^2\|u_k^m\|^2,$$

while for Case 2 we have:

$$\mathbb{E}_{z_k^m}[\gamma(u_k^m; z_k^m, z_k^m)] = \nabla Q_\lambda(u) \text{ and } \mathbb{E}_{z_k^m}\|\gamma(u_k^m; z_k^m, z_k^m) - \nabla Q_\lambda(u_k^m)\|^2 \leq 2\sigma^2 + 2\rho^2\|u_k^m\|^2,$$

where we used the fact that $\|a+b\|^2 \leq 2\|a\|^2 + 2\|b\|^2$ for any $a, b \in \mathbb{R}^d$, and so in either case the variance term is bounded by $2\sigma^2 + 2\rho^2\|u_k^m\|^2$.

A key feature of these stochastic gradients of $Q_\lambda(u)$, which we will use frequently, is that by the linearity of $\nabla Q_\lambda(u) = (\bar{\xi}\nabla^2 F(x) + \lambda\mathbf{I})u + \nabla F(x)$,

$$\mathbb{E}\bar{\gamma}_k = \frac{1}{M}\sum_{m=1}^M \mathbb{E}\gamma(u_k^m; z_k^m, z_k^{'m}) = \frac{1}{M}\sum_{m=1}^M \mathbb{E}\nabla Q_\lambda(u_k^m) = \nabla Q_\lambda(\mathbb{E}\bar{u}_k). \tag{77}$$

We begin by expanding,

$$\mathbb{E}\|\bar{u}_{k+1} - u^*\|^2$$

$$= \mathbb{E}\|\bar{u}_k - u^*\|^2 + \eta_k^2 \mathbb{E}\|\bar{\gamma}_k\|^2 - 2\eta_k \mathbb{E}\langle \nabla Q_\lambda(\bar{u}_k), \bar{u}_k - u^* \rangle \tag{78}$$

$$= \mathbb{E}\|\bar{u}_k - u^*\|^2 + \eta_k^2 \mathbb{E}\|\nabla Q_\lambda(\bar{u}_k)\|^2 - 2\eta_k \left[ Q_\lambda(\bar{u}_k) - Q_\lambda^* + \frac{\lambda}{2}\|\bar{u}_k - u^*\|^2 \right] \tag{79}$$

$$+ \eta_k^2 \mathbb{E}\|\bar{\gamma}_k - \nabla Q_\lambda(\bar{u}_k)\|^2 \tag{80}$$

$$\leq (1 - \eta_k\lambda)\mathbb{E}\|\bar{u}_k - u^*\|^2 - 2\eta_k(1 - (\bar{\xi}H + \lambda)\eta_k)\mathbb{E}[Q_\lambda(\bar{u}_k) - Q_\lambda^*] \tag{81}$$

$$+ \eta_k^2 \left( \frac{2\sigma^2}{M} + \frac{2\rho^2}{M^2}\sum_{m=1}^{M} \mathbb{E}\|u_k^m\|^2 \right). \tag{82}$$

Since $\eta_k \leq \frac{1}{2(\bar{\xi}H + \lambda)}$ for all $k$, $(1 - (\bar{\xi}H + \lambda)\eta_k) \geq \frac{1}{2}$ so we can rearrange

$$\mathbb{E}[Q_\lambda(\bar{u}_k) - Q_\lambda^*] \leq \left( \frac{1}{\eta_k} - \lambda \right)\mathbb{E}\|\bar{u}_k - u^*\|^2 - \frac{1}{\eta_k}\mathbb{E}\|\bar{u}_{k+1} - u^*\|^2 \tag{83}$$

$$+ \eta_k \left( \frac{2\sigma^2}{M} + \frac{2\rho^2}{M^2}\sum_{m=1}^{M} \mathbb{E}\|u_k^m\|^2 \right). \tag{84}$$

From here, we note that since $u_k^1, \ldots, u_k^M$ are i.i.d.,

$$\frac{1}{M^2}\sum_{m=1}^{M} \mathbb{E}\|u_k^m\|^2 = \frac{1}{M^2}\sum_{m=1}^{M} \left[ \mathbb{E}\|u_k^m - \mathbb{E}u_k^m\|^2 + \|\mathbb{E}u_k^m\|^2 \right] \tag{85}$$

$$= \mathbb{E}\|\bar{u}_k^m - \mathbb{E}\bar{u}_k^m\|^2 + \frac{1}{M}\|\mathbb{E}\bar{u}_k\|^2 \tag{86}$$

$$\leq \mathbb{E}\|\bar{u}_k - u^*\|^2 + \frac{1}{M}\|\mathbb{E}\bar{u}_k\|^2 \tag{87}$$

$$\leq \mathbb{E}\|\bar{u}_k - u^*\|^2 + \frac{2}{M}\|\mathbb{E}\bar{u}_k - u^*\|^2 + \frac{2}{M}\|u^*\|^2. \tag{88}$$

Furthermore, for each $k$, since $\eta_{k-1} \leq \frac{1}{\bar{\xi}H + \lambda}$

$$\|\mathbb{E}\bar{u}_k - u^*\|^2 \leq \|\mathbb{E}\bar{u}_{k-1} - u^*\|^2 + \eta_{k-1}^2\|\nabla Q_\lambda(\mathbb{E}\bar{u}_{k-1})\|^2 - 2\eta_{k-1}\langle \nabla Q_\lambda(\mathbb{E}\bar{u}_{k-1}), \bar{u}_{k-1} - u^* \rangle \tag{89}$$

$$\leq \|\mathbb{E}\bar{u}_{k-1} - u^*\|^2 + 2(\bar{\xi}H + \lambda)\eta_{k-1}^2[Q_\lambda(\mathbb{E}\bar{u}_{k-1}) - Q_\lambda^*] \tag{90}$$

$$- 2\eta_{k-1}[Q_\lambda(\mathbb{E}\bar{u}_{k-1}) - Q_\lambda^*] \tag{91}$$

$$\leq \|\mathbb{E}\bar{u}_{k-1} - u^*\|^2 \leq \|\mathbb{E}\bar{u}_0 - u^*\|^2 = \|u^*\|^2. \tag{92}$$

Therefore, returning to (84),

$$\mathbb{E}[Q_\lambda(\bar{u}_k) - Q_\lambda^*] \leq \left( \frac{1}{\eta_k} - \lambda \right)\mathbb{E}\|\bar{u}_k - u^*\|^2 - \frac{1}{\eta_k}\mathbb{E}\|\bar{u}_{k+1} - u^*\|^2 + \frac{2\eta_k\sigma^2}{M} \tag{93}$$

$$+ 2\eta_k\rho^2 \left( \mathbb{E}\|\bar{u}_k - u^*\|^2 + \frac{4}{M}\|u^*\|^2 \right) \tag{94}$$

$$= \left( \frac{1}{\eta_k} - \lambda + \eta_k\rho^2 \right)\mathbb{E}\|\bar{u}_k - u^*\|^2 - \frac{1}{\eta_k}\mathbb{E}\|\bar{u}_{k+1} - u^*\|^2 \tag{95}$$

$$+ \frac{2\eta_k\left( \sigma^2 + \rho^2\|u^*\|^2 \right)}{M}. \tag{96}$$

We first consider the case $K \leq \frac{2\max\left\{ \bar{\xi}H + \lambda, \frac{\rho^2}{\lambda} \right\}}{\lambda}$, so that

$$\eta_k = \eta = \min\left\{ \frac{1}{2(\bar{\xi}H + \lambda)}, \frac{\lambda}{2\rho^2} \right\} \quad \text{and} \quad w_k = (1 - \lambda\eta + \eta^2\rho^2)^{-k-1}.$$

Then,

$$\mathbb{E}Q_\lambda\left(\frac{1}{\sum_{k=0}^K w_k}\sum_{k=0}^K w_k \bar{u}_k\right) - Q_\lambda^*$$

$$\leq \frac{1}{\eta\sum_{k=0}^K w_k}\sum_{k=0}^K\left[w_k\left(1-\eta\lambda+\eta^2\rho^2\right)\mathbb{E}\|\bar{u}_k-u^*\|^2 - w_k\mathbb{E}\|\bar{u}_{k+1}-u^*\|^2\right] \tag{97}$$

$$+\frac{2\eta\left(\sigma^2+\rho^2\|u^*\|^2\right)}{M} \tag{98}$$

$$= \frac{1}{\eta\sum_{k=0}^K w_k}\sum_{k=0}^K\left[\left(1-\eta\lambda+\eta^2\rho^2\right)^{-k}\mathbb{E}\|\bar{u}_k-u^*\|^2 - \left(1-\eta\lambda+\eta^2\rho^2\right)^{-(k+1)}\mathbb{E}\|\bar{u}_{k+1}-u^*\|^2\right] \tag{99}$$

$$+\frac{2\eta\left(\sigma^2+\rho^2\|u^*\|^2\right)}{M} \tag{100}$$

$$\leq \frac{\mathbb{E}\|\bar{u}_0-u^*\|^2}{\eta\sum_{k=0}^K(1-\eta\lambda+\eta^2\rho^2)^{-(k+1)}} + \frac{2\eta\left(\sigma^2+\rho^2\|u^*\|^2\right)}{M} \tag{101}$$

$$\leq \frac{\|u^*\|^2}{\eta\max\left\{K,\,(1-\eta\lambda+\eta^2\rho^2)^{-(K+1)}\right\}} + \frac{2\eta\left(\sigma^2+\rho^2\|u^*\|^2\right)}{M} \tag{102}$$

$$\leq 2\max\left\{\bar{\xi}H+\lambda,\,\frac{\rho^2}{\lambda}\right\}\|u^*\|^2\min\left\{\frac{1}{K},\,\exp\left(-\frac{K+1}{4}\min\left\{\frac{\lambda}{\bar{\xi}H+\lambda},\,\frac{\lambda^2}{\rho^2}\right\}\right)^{K+1}\right\}$$

$$+\frac{2(\sigma^2+\rho^2\|u^*\|^2)}{\lambda MK}\,. \tag{103}$$

For the last line, we used that $K \leq \frac{2\max\left\{\bar{\xi}H+\lambda,\,\frac{\rho^2}{\lambda}\right\}}{\lambda} = \frac{1}{\eta\lambda}$.

In the second case $\left(K > \frac{2\max\left\{\bar{\xi}H+\lambda,\,\frac{\rho^2}{\lambda}\right\}}{\lambda}\right)$, we consider the first $K/2$ iterations where

$$\eta_k = \eta = \min\left\{\frac{1}{2(\bar{\xi}H+\lambda)},\,\frac{\lambda}{2\rho^2}\right\}:$$

$$\mathbb{E}Q_\lambda(\bar{u}_{K/2-1}) - Q_\lambda^* \le \left(\frac{1}{\eta} - \lambda + \eta\rho^2\right)\mathbb{E}\big\|\bar{u}_{K/2-1} - u^*\big\|^2 - \frac{1}{\eta}\mathbb{E}\big\|\bar{u}_{K/2} - u^*\big\|^2 \tag{104}$$

$$+ \frac{2\eta\left(\sigma^2 + \rho^2\|u^*\|^2\right)}{M} \tag{105}$$

$$\implies \mathbb{E}\big\|\bar{u}_{K/2} - u^*\big\|^2 \le \left(1 - \eta\lambda + \eta^2\rho^2\right)\mathbb{E}\big\|\bar{u}_{K/2-1} - u^*\big\|^2 + \frac{2\eta^2\left(\sigma^2 + \rho^2\|u^*\|^2\right)}{M} \tag{106}$$

$$= \left(1 - \eta\lambda + \eta^2\rho^2\right)^{K/2}\mathbb{E}\|\bar{u}_0 - u^*\|^2 \tag{107}$$

$$+ \frac{2\eta^2\left(\sigma^2 + \rho^2\|u^*\|^2\right)}{M}\sum_{k=0}^{K/2-1}\left(1 - \eta\lambda + \eta^2\rho^2\right)^k \tag{108}$$

$$= \|u^*\|^2\left(1 - \eta\lambda + \eta^2\rho^2\right)^{K/2} \tag{109}$$

$$+ \frac{2\eta^2\left(\sigma^2 + \rho^2\|u^*\|^2\right)}{M}\frac{1 - \left(1 - \eta\lambda + \eta^2\rho^2\right)^{K/2}}{\eta\lambda - \eta^2\rho^2} \tag{110}$$

$$\le \|u^*\|^2\left(1 - \eta\lambda + \eta^2\rho^2\right)^{K/2} + \frac{2\eta\left(\sigma^2 + \rho^2\|u^*\|^2\right)}{M(\lambda - \eta\rho^2)} \tag{111}$$

$$\le \|u^*\|^2\left(1 - \frac{1}{2}\eta\lambda\right)^{K/2} + \frac{4\eta\left(\sigma^2 + \rho^2\|u^*\|^2\right)}{\lambda M}. \tag{112}$$

This bounds the distance of the $(K/2)^{\text{th}}$ iterate to the optimum, from which we can upper bound the suboptimality of the averaged iterate. Let $W = \sum_{k=0}^{K}w_k = \sum_{k=K/2}^{K}w_k$. Then by the convexity of $Q_\lambda(u)$,

$$\mathbb{E}Q_\lambda\left(\frac{1}{W}\sum_{k=0}^{K}w_k\bar{u}_k\right) - Q_\lambda^*$$

$$\le \frac{1}{W}\sum_{k=0}^{K}\left[w_k\left(\frac{1}{\eta_k} - \lambda + \eta_k\rho^2\right)\mathbb{E}\|\bar{u}_k - u^*\|^2 - \frac{w_k}{\eta_k}\mathbb{E}\|\bar{u}_{k+1} - u^*\|^2 + \frac{2w_k\eta_k\left(\sigma^2 + \rho^2\|u^*\|^2\right)}{M}\right] \tag{113}$$

$$\le \frac{w_{K/2}\left(\frac{1}{\eta_{K/2}} - \lambda + \eta_{K/2}\rho^2\right)\mathbb{E}\big\|\bar{u}_{K/2} - u^*\big\|^2}{W}$$

$$+ \frac{1}{W}\sum_{k=K/2+1}^{K}\left[\left(\frac{w_k}{\eta_k} - w_k\lambda + w_k\eta_k\rho^2 - \frac{w_{k-1}}{\eta_{k-1}}\right)\mathbb{E}\|\bar{u}_k - u^*\|^2 + \frac{2w_k\eta_k\left(\sigma^2 + \rho^2\|u^*\|^2\right)}{M}\right]. \tag{114}$$

With our setting of $w_k = (a + k - K/2 - 1)$ and $\eta_k = \frac{4}{\lambda(a+k-K/2)}$ with $a = \frac{8}{\lambda}\max\left\{\frac{\rho^2}{\lambda}, \bar{\xi}H + \lambda\right\}$, we have for $k > K/2$,

$$\frac{\eta_{k-1}}{\eta_k} - \eta_{k-1}\lambda + \eta_{k-1}\eta_k\rho^2 = \frac{a+k-K/2}{a+k-K/2-1} - \frac{4}{a+k-K/2-1} \tag{115}$$

$$+ \frac{16\rho^2}{\lambda^2(a+k-K/2)(a+k-K/2-1)} \tag{116}$$

$$= \frac{a+k-K/2}{a+k-K/2-1} \tag{117}$$

$$+ \frac{1}{a+k-K/2-1}\left(2 - 4 + \frac{16\rho^2}{\lambda^2(a+k-K/2)}\right) \tag{118}$$

$$\leq \frac{w_{k-1}}{w_k} + \frac{1}{a+k-K/2-1}\left(-2 + \frac{16\rho^2}{\lambda^2 a}\right) \tag{119}$$

$$\leq \frac{w_{k-1}}{w_k} . \tag{120}$$

Therefore, we have

$$\mathbb{E}Q_\lambda(\hat{x}) - Q_\lambda^*$$

$$\leq \frac{w_{K/2}\left(\frac{1}{\eta_{K/2}} - \lambda + \eta_{K/2}\rho^2\right)\mathbb{E}\left\|\bar{u}_{K/2} - u^*\right\|^2}{W} + \frac{2}{W}\sum_{k=K/2+1}^{K}\frac{w_k\eta_k\left(\sigma^2 + \rho^2\|u^*\|^2\right)}{M} \tag{121}$$

$$= \frac{(a-1)\left(\frac{\lambda a}{4} - \lambda + \frac{4\rho^2}{\lambda a}\right)\mathbb{E}\left\|\bar{u}_{K/2} - u^*\right\|^2}{W} \tag{122}$$

$$+ \frac{2(\sigma^2 + \rho^2\|u^*\|^2)}{WM}\sum_{k=K/2+1}^{K}\frac{4(a+k-K/2-1)}{\lambda(a+k-K/2)} \tag{123}$$

$$\leq \frac{\lambda a^2\mathbb{E}\left\|\bar{u}_{K/2} - u^*\right\|^2}{4W} + \frac{8K(\sigma^2 + \rho^2\|u^*\|^2)}{\lambda WM} \tag{124}$$

$$\leq \frac{\lambda a^2}{4W}\left(\|u^*\|^2\left(1 - \frac{1}{2}\eta\lambda\right)^{K/2} + \frac{4\eta\left(\sigma^2 + \rho^2\|u^*\|^2\right)}{\lambda M}\right) + \frac{8K(\sigma^2 + \rho^2\|u^*\|^2)}{\lambda WM} \tag{125}$$

$$= \frac{\lambda a^2\|u^*\|^2}{4W}\left(1 - \frac{1}{2}\eta\lambda\right)^{K/2} + \frac{(\sigma^2 + \rho^2\|u^*\|^2)}{\lambda WM}\left(8K + \eta\lambda a^2\right) \tag{126}$$

$$= \frac{\lambda a^2\|u^*\|^2}{4W}\left(1 - \min\left\{\frac{\lambda}{4(\bar{\xi}H + \lambda)}, \frac{\lambda^2}{4\rho^2}\right\}\right)^{K/2} + \frac{(8K + 4a)(\sigma^2 + \rho^2\|u^*\|^2)}{\lambda WM} , \tag{127}$$

where, for the third-to-last line we used (112), and the last line we used that $\eta = \min\left\{\frac{1}{2(\bar{\xi}H+\lambda)}, \frac{\lambda}{2\rho^2}\right\} = \frac{4}{\lambda a}$. Finally, we lower bound

$$W = \sum_{k=K/2}^{K}(a + k - K/2) = \sum_{i=0}^{K/2}a + i \geq \frac{1}{2}aK + \frac{1}{8}K^2. \tag{128}$$

Thus, since $K \geq \frac{2}{\lambda}\max\left\{\frac{\rho^2}{\lambda}, \bar{\xi}H + \lambda\right\} = \frac{a}{4}$, we conclude

$$\mathbb{E}Q_\lambda(\hat{x}) - Q_\lambda^* \leq 96\lambda\|u^*\|^2\left(1 - \min\left\{\frac{\lambda}{4(\bar{\xi}H + \lambda)}, \frac{\lambda^2}{4\rho^2}\right\}\right)^{K/2} + \frac{96(\sigma^2 + \rho^2\|u^*\|^2)}{\lambda MK} . \tag{129}$$

Combining this and (103) completes the proof. $\square$

# E Proof of Theorem 1

**Theorem 1.** *Let $F$ satisfy Assumption 2. Then, for $K \geq 175$ and $R \geq \tilde{\Omega}(1)$, and for hyperparameters $T$, $\beta$, $\bar{r}$, $\bar{\xi}$, $\bar{\lambda}$, $N$, $C$ and parameter functions $\eta_k(\lambda)$, $w_k(\lambda)$ as in Tables 3 and 4, the output of FEDSN (Algorithm 1) with initial point $x_0 \in \mathbb{R}^d$, using Algorithms 2, 3, and 4 (for both Cases 1 and 2) satisfies*

$$\mathbb{E}[F(x_T)] - F^* \leq HB^2\left(\exp\left(-\frac{R}{\tilde{O}(\alpha B)}\right) + \exp\left(-\frac{K}{O(1)}\right)\right) + \tilde{O}\left(\frac{\sigma B}{\sqrt{MK}} + \frac{HB^2}{KR} + \frac{\rho B^2}{\sqrt{K}R}\right),$$

*where $\tilde{\Omega}$, $\tilde{O}$ hide terms logarithmic in $R$, $K$, and $\alpha B$.*

*Proof.* We first recall the hyperparameters from Table 4, whose settings we will refer to throughout the course of the proof:

| Hyperparameter Setting | Description |
|---|---|
| $T := \left\lfloor \frac{R}{4\zeta}\log^2\left(\left(\frac{R}{\zeta}\right)\right)\right\rfloor$ 
 (for $\zeta = 4096 + 4(80 + 32\log K + 24\log(1 + 2\alpha B))^2$) | Main iterations |
| $\beta := 0$ | Momentum |
| $\bar{r} := \min\left\{\frac{32B}{T}\log(TK), \frac{1}{5\alpha}\right\}$ | Trust-region radius |
| $\bar{\xi} := \exp(\alpha\bar{r})$ | Local stability |
| $\bar{\lambda} := \max\left\{\frac{2eH}{K-2}, \frac{2\rho}{\sqrt{K}}, \frac{32eH\log(51200)}{K}, \frac{4\rho\sqrt{2\log(51200)}}{\sqrt{K}}, \right.$ 
 $\left. \frac{320\sqrt{2}\rho}{\sqrt{MK}}, \frac{320\sigma}{\bar{r}\sqrt{MK}}, \frac{8eH}{K-16}\right\}$ | Regularization bound |
| $N := \left\lceil 1 + \frac{5}{2}\log\frac{H(B+5T\bar{r})}{3\bar{\lambda}\bar{r}}\right\rceil$ | Binary search iterations |
| $C := \left\lceil 8\log\left(\lceil\log_2 N\rceil\left(4 + \frac{eH}{\lambda} + \frac{80H(B+5T\bar{r})}{\bar{\lambda}\bar{r}}\right)\right)\right\rceil$ | Reg. quadratic repetitions |

Table 4: Hyperparameters $T$, $\beta$, $\bar{r}$, $\bar{\xi}$, $\bar{\lambda}$, $N$, and $C$, as used by FEDSN and its subroutines.

We also note that since all of the updates $\Delta\tilde{x}_t$ have norm at most $5\bar{r}$, $\|x_t - x^*\| \leq B + 5T\bar{r}$ for all $t$, and therefore by the $H$-smoothness of $F$, $\|\nabla F(x_t)\| \leq H(B + 5T\bar{r})$ for all $t$. Furthermore, since $F$ is $H$-smooth and $\bar{r} \leq \frac{1}{5\alpha}$, $\bar{\xi}\nabla^2 F(x_t) \preceq eH\mathbf{I}$ for all $t$. Therefore, our settings of $N$ and $C$ satisfy the conditions of Lemma 3, and for each $t$,

$$\mathbb{E}Q_t(\Delta\tilde{x}_t) - \min_{\Delta x:\|\Delta x\|\leq\frac{1}{2}\bar{r}} Q_t(\Delta x) \leq \frac{3}{4}\left(Q_t(0) - \min_{\Delta x:\|\Delta x\|\leq\frac{1}{2}\bar{r}} Q_t(\Delta x)\right) + \epsilon(\lambda_{4\bar{r}}) + \frac{\bar{\lambda}\bar{r}^2}{4} \quad (130)$$

as long as the error guarantee of Algorithm 3 satisfies for all $\lambda \geq \bar{\lambda}$

$$\mathbb{E}Q_\lambda(\hat{u}) - \min_u Q_\lambda(u) \leq \epsilon(\lambda) = \frac{\lambda(r^*(\lambda)^2 + \bar{r}^2)}{800}. \quad (131)$$

By Lemma 4, since the objectives are such that $\left\|\bar{\xi}\nabla^2 F(x_t) + \lambda\mathbf{I}\right\|_2 \le eH + \lambda$ for all $t$ and $\lambda$, the output with optimally chosen stepsizes have error at most

$$\mathbb{E}Q_\lambda(\hat{u}) - \min_u Q_\lambda(u) \le \tilde{\epsilon}(\lambda) \tag{132}$$

$$:= \begin{cases} 2\max\left\{eH + \lambda, \frac{\rho^2}{\lambda}\right\}r^*(\lambda)^2\exp\left(-\frac{K+1}{4}\min\left\{\frac{\lambda}{eH+\lambda}, \frac{\lambda^2}{\rho^2}\right\}\right) + \frac{2(\sigma^2+\rho^2 r^*(\lambda)^2)}{\lambda MK} \\ \qquad\qquad\qquad\qquad\qquad\qquad\qquad\qquad\qquad\qquad K \le \frac{2}{\lambda}\max\left\{eH + \lambda, \frac{\rho^2}{\lambda}\right\} \\ 96\lambda r^*(\lambda)^2\exp\left(-\frac{K}{8}\min\left\{\frac{\lambda}{eH+\lambda}, \frac{\lambda^2}{\rho^2}\right\}\right) + \frac{96(\sigma^2+\rho^2 r^*(\lambda)^2)}{\lambda MK} \\ \qquad\qquad\qquad\qquad\qquad\qquad\qquad\qquad\qquad\qquad K > \frac{2}{\lambda}\max\left\{eH + \lambda, \frac{\rho^2}{\lambda}\right\} \end{cases} \tag{133}$$

With our choice of

$$\bar{\lambda} = \max\left\{\frac{2eH}{K-2}, \frac{2\rho}{\sqrt{K}}, \frac{32eH\log(51200)}{K}, \frac{4\rho\sqrt{2\log(51200)}}{\sqrt{K}}, \frac{320\sqrt{2}\rho}{\sqrt{MK}}, \frac{320\sigma}{\bar{r}\sqrt{MK}}, \frac{8eH}{K-16}\right\}, \tag{134}$$

we note that

$$K \ge \frac{2}{\lambda}\max\left\{eH + \lambda, \frac{\rho^2}{\lambda}\right\}, \tag{135}$$

so for $\lambda \ge \bar{\lambda}$

$$\tilde{\epsilon}(\lambda) \le 96\lambda r^*(\lambda)^2\exp\left(-\frac{K}{8}\min\left\{\frac{\lambda}{eH+\lambda}, \frac{\lambda^2}{\rho^2}\right\}\right) + \frac{96(\sigma^2+\rho^2 r^*(\lambda)^2)}{\lambda MK}. \tag{136}$$

Furthermore, $K \ge 175$ and $\lambda \ge \bar{\lambda} \ge \max\left\{\frac{32eH\log(51200)}{K}, \frac{4\rho\sqrt{2\log(51200)}}{\sqrt{K}}\right\}$ implies

$$96\exp\left(-\frac{K}{8}\min\left\{\frac{\lambda}{eH+\lambda}, \frac{\lambda^2}{\rho^2}\right\}\right) \le \frac{1}{1600}. \tag{137}$$

Likewise, $\lambda \ge \bar{\lambda} \ge \frac{320\sqrt{2}\rho}{\sqrt{MK}}$ implies

$$\frac{96\rho^2}{\lambda^2 MK} \le \frac{1}{1600}. \tag{138}$$

Finally, $\lambda \ge \bar{\lambda} \ge \frac{320\sigma}{\bar{r}\sqrt{MK}}$ implies

$$\frac{96\sigma^2}{\lambda^2 MK} \le \frac{\bar{r}^2}{800}. \tag{139}$$

Putting these together, we conclude that for $\lambda \ge \bar{\lambda}$

$$\tilde{\epsilon}(\lambda) \le \frac{\lambda(r^*(\lambda)^2 + \bar{r}^2)}{800} = \epsilon(\lambda). \tag{140}$$

Combining this with (130), we conclude that the output of Algorithm 2 satisfies

$$\mathbb{E}Q_t(\Delta\tilde{x}_t) - \min_{\Delta x:\|\Delta x\|\le\frac{1}{2}\bar{r}} Q_t(\Delta x)$$

$$\le \frac{3}{4}\left(Q_t(0) - \min_{\Delta x:\|\Delta x\|\le\frac{1}{2}\bar{r}} Q_t(\Delta x)\right)$$

$$+ 512\lambda\bar{r}^2\exp\left(-\frac{K}{8}\min\left\{\frac{\lambda}{eH+\lambda}, \frac{\lambda^2}{\rho^2}\right\}\right) + \frac{96(\sigma^2 + 16\rho^2\bar{r}^2)}{\lambda MK} + \frac{\bar{\lambda}\bar{r}^2}{4} \tag{141}$$

$$\le \frac{3}{4}\left(Q_t(0) - \min_{\Delta x:\|\Delta x\|\le\frac{1}{2}\bar{r}} Q_t(\Delta x)\right)$$

$$+ 512\lambda\bar{r}^2\exp\left(-\frac{K}{8}\min\left\{\frac{\lambda}{eH+\lambda}, \frac{\lambda^2}{\rho^2}\right\}\right) + \frac{\sigma\bar{r} + 16\rho\bar{r}^2}{2\sqrt{MK}} + \frac{\bar{\lambda}\bar{r}^2}{4}. \tag{142}$$

Now, we upper bound $\lambda \mapsto \lambda \exp\left(-\frac{K}{8}\min\left\{\frac{\lambda}{eH+\lambda}, \frac{\lambda^2}{\rho^2}\right\}\right)$ for $\lambda \in \Lambda_1$. First,

$$\lambda \exp\left(-\frac{K}{8}\min\left\{\frac{\lambda}{eH+\lambda}, \frac{\lambda^2}{\rho^2}\right\}\right) = \max\left\{\lambda \exp\left(-\frac{\lambda K}{8eH+8\lambda}\right), \lambda \exp\left(-\frac{\lambda^2 K}{8\rho^2}\right)\right\}. \quad (143)$$

Considering each term separately,

$$\frac{d}{d\lambda}\left[\lambda \exp\left(-\frac{\lambda K}{8eH+8\lambda}\right)\right] = \exp\left(-\frac{\lambda K}{8eH+8\lambda}\right)\left(1 - \frac{eHK\lambda}{8(eH+\lambda)^2}\right). \quad (144)$$

This is less than zero if $8(eH+\lambda)^2 \leq eHK\lambda$, i.e.,

$$\frac{eH}{16}\left(K - 16 - \sqrt{(K-16)^2 - 16}\right) \leq \lambda \leq \frac{eH}{16}\left(K - 16 + \sqrt{(K-16)^2 - 16}\right). \quad (145)$$

With our choice of $\bar{\lambda} \geq \frac{8eH}{K-16}$, for any $\lambda \geq \bar{\lambda}$ and $K \geq 175$,

$$\lambda \geq \frac{8eH}{K-16} \geq \frac{eH}{16}\left(K - 16 - \sqrt{(K-16)^2 - 16}\right), \quad (146)$$

so the left side of this inequality is satisfied. Thus, for $\lambda \in \Lambda_1$ such that $\lambda \leq \frac{eH}{16}\left(K - 16 + \sqrt{(K-16)^2 - 16}\right)$,

$$\lambda \exp\left(-\frac{\lambda K}{8eH+8\lambda}\right) \leq \bar{\lambda} \exp\left(-\frac{\bar{\lambda} K}{8eH+8\bar{\lambda}}\right). \quad (147)$$

Also, if $\lambda > \frac{eH}{16}\left(K - 16 + \sqrt{(K-16)^2 - 16}\right)$, then since $K \geq 175$

$$\lambda \exp\left(-\frac{\lambda K}{8eH+8\lambda}\right) \leq \lambda \exp\left(-\frac{\frac{eH(K-16)}{16}K}{8eH + \frac{eH(K-16)}{2}}\right) \leq \lambda \exp\left(-\frac{K}{10}\right). \quad (148)$$

Furthermore, for $\lambda \in \Lambda_1$,

$$\lambda \leq \bar{\lambda}\left(\frac{3}{2}\right)^{N-1} \leq \frac{3\bar{\lambda}}{2}\frac{H(B+5T\bar{r})}{3\bar{\lambda}\bar{r}} = \frac{3H(B+5T\bar{r})}{6\bar{r}}. \quad (149)$$

Therefore, for any $\lambda \in \Lambda_1$

$$\lambda \exp\left(-\frac{\lambda K}{8eH+8\lambda}\right) \leq \max\left\{\bar{\lambda} \exp\left(-\frac{\bar{\lambda} K}{8eH+8\bar{\lambda}}\right), \frac{3H(B+5T\bar{r})}{6\bar{r}}\exp\left(-\frac{K}{10}\right)\right\}. \quad (150)$$

Similarly,

$$\frac{d}{d\lambda}\left[\lambda \exp\left(-\frac{\lambda^2 K}{8\rho^2}\right)\right] = \exp\left(-\frac{\lambda^2 K}{8\rho^2}\right)\left(1 - \frac{\lambda^2 K}{4\rho^2}\right). \quad (151)$$

This is negative for all $\lambda \geq \bar{\lambda} \geq \frac{2\rho}{\sqrt{K}}$, so

$$\lambda \exp\left(-\frac{\lambda^2 K}{8\rho^2}\right) \leq \bar{\lambda} \exp\left(-\frac{\bar{\lambda}^2 K}{8\rho^2}\right). \quad (152)$$

We conclude that

$$\mathbb{E}Q_t(\Delta\tilde{x}_t) - \min_{\Delta x:\|\Delta x\|\leq\frac{1}{2}\bar{r}}Q_t(\Delta x)$$

$$\leq \frac{3}{4}\left(Q_t(0) - \min_{\Delta x:\|\Delta x\|\leq\frac{1}{2}\bar{r}}Q_t(\Delta x)\right) + \frac{\sigma\bar{r} + 16\rho\bar{r}^2}{2\sqrt{MK}} + \frac{\bar{\lambda}\bar{r}^2}{4}$$

$$+ 512\bar{r}^2\max\left\{\bar{\lambda}\exp\left(-\frac{\bar{\lambda}K}{8eH+8\bar{\lambda}}\right), \frac{3H(B+5T\bar{r})}{6\bar{r}}\exp\left(-\frac{K}{10}\right), \bar{\lambda}\exp\left(-\frac{\bar{\lambda}^2 K}{8\rho^2}\right)\right\}. \tag{153}$$

Now, because $F$ is $\alpha$-quasi-self-concordant, by Lemma 1, $F$ is $\exp(\alpha r)$-locally stable, so with our choice of $\bar{r} \leq \frac{1}{5\alpha}$, we have that $\exp(\frac{1}{2}\bar{r})\exp(5\bar{r}) \leq e^{1.1} \leq 4$. Thus, it follows from Lemma 2, for $\theta = \frac{3}{4}$, combined with the guarantee on the output of Algorithm 2 from Lemma 3, that

$$
\mathbb{E}F(x_T) - F^*
$$

$$
\leq \mathbb{E}[F(x_0) - F^*]\exp\left(-\frac{T\bar{r}}{32B}\right) + \frac{32B}{\bar{r}}\left(\frac{\sigma\bar{r} + 16\rho\bar{r}^2}{2\sqrt{MK}} + \frac{\bar{\lambda}\bar{r}^2}{4}\right)
$$

$$
+ \frac{32B}{\bar{r}} \cdot 512\bar{r}^2 \max\left\{\bar{\lambda}\exp\left(-\frac{\bar{\lambda}K}{8eH + 8\bar{\lambda}}\right), \frac{3H(B + 5T\bar{r})}{6\bar{r}}\exp\left(-\frac{K}{10}\right), \bar{\lambda}\exp\left(-\frac{\bar{\lambda}^2 K}{8\rho^2}\right)\right\}
$$
(154)

$$
\leq \mathbb{E}[F(x_0) - F^*]\exp\left(-\frac{T\bar{r}}{32B}\right) + \frac{32\sigma B + 512\rho B\bar{r}}{2\sqrt{MK}} + 8\bar{\lambda}B\bar{r}
$$

$$
+ 2^{14}\max\left\{\bar{\lambda}B\bar{r}\exp\left(-\frac{\bar{\lambda}K}{8eH + 8\bar{\lambda}}\right), \frac{3H(B^2 + 5TB\bar{r})}{6}\exp\left(-\frac{K}{10}\right), \bar{\lambda}B\bar{r}\exp\left(-\frac{\bar{\lambda}^2 K}{8\rho^2}\right)\right\}
$$
(155)

$$
\leq \mathbb{E}[F(x_0) - F^*]\exp\left(-\frac{T\bar{r}}{32B}\right) + \frac{32\sigma B + 512\rho B\bar{r}}{2\sqrt{MK}}
$$

$$
+ \bar{\lambda}B\bar{r}\left(8 + 2^{14}\max\left\{\exp\left(-\frac{\bar{\lambda}K}{8eH + 8\bar{\lambda}}\right), \frac{3H(B + 5T\bar{r})}{6\bar{\lambda}\bar{r}}\exp\left(-\frac{K}{10}\right), \exp\left(-\frac{\bar{\lambda}^2 K}{8\rho^2}\right)\right\}\right).
$$
(156)

We have, for a constant $c$,

$$
\bar{\lambda} = \max\left\{\frac{2eH}{K - 2}, \frac{2\rho}{\sqrt{K}}, \frac{32eH\log(51200)}{K}, \frac{4\rho\sqrt{2\log(51200)}}{\sqrt{K}}, \frac{320\sqrt{2}\rho}{\sqrt{MK}}, \frac{320\sigma}{\bar{r}\sqrt{MK}}, \frac{8eH}{K - 16}\right\}
$$
(157)

$$
= \max\left\{\frac{32eH\log(51200)}{K}, \frac{4\rho\sqrt{2\log(51200)}}{\sqrt{K}}, \frac{320\sqrt{2}\rho}{\sqrt{MK}}, \frac{320\sigma}{\bar{r}\sqrt{MK}}\right\}
$$
(158)

$$
= c \cdot \max\left\{\frac{H}{K}, \frac{\rho}{\sqrt{K}}, \frac{\sigma}{\bar{r}\sqrt{MK}}\right\}.
$$
(159)

So, for a constant $c'$, and using $\mathbb{E}F(x_0) - F^* \leq \frac{HB^2}{2}$,

$$
\mathbb{E}F(x_T) - F^*
$$

$$
\leq c' \cdot \left(HB^2\exp\left(-\frac{T\bar{r}}{32B}\right) + \frac{\sigma B + \rho B\bar{r}}{\sqrt{MK}}\right.
$$

$$
\left. + \bar{\lambda}B\bar{r}\left(1 + \max\left\{\exp\left(-\frac{\bar{\lambda}K}{H + \bar{\lambda}}\right), \frac{H(B + T\bar{r})}{\bar{\lambda}\bar{r}}\exp\left(-\frac{K}{10}\right), \exp\left(-\frac{\bar{\lambda}^2 K}{\rho^2}\right)\right\}\right)\right)
$$
(160)

$$
\leq c' \cdot \left(HB^2\exp\left(-\frac{T\bar{r}}{32B}\right) + \frac{\sigma B + \rho B\bar{r}}{\sqrt{MK}} + \max\left\{\bar{\lambda}B\bar{r}, H(B^2 + TB\bar{r})\exp\left(-\frac{K}{10}\right)\right\}\right)
$$
(161)

$$
= c' \cdot \left(HB^2\exp\left(-\frac{T\bar{r}}{32B}\right) + \frac{\sigma B + \rho B\bar{r}}{\sqrt{MK}}\right.
$$

$$
\left. + \max\left\{\frac{HB\bar{r}}{K}, \frac{\rho B\bar{r}}{\sqrt{K}}, \frac{\sigma B}{\sqrt{MK}}, H(B^2 + TB\bar{r})\exp\left(-\frac{K}{10}\right)\right\}\right)
$$
(162)

$$
= c' \cdot \left(HB^2\exp\left(-\frac{T\bar{r}}{32B}\right) + \frac{\sigma B}{\sqrt{MK}} + \frac{HB\bar{r}}{K} + \frac{\rho B\bar{r}}{\sqrt{K}} + H(B^2 + TB\bar{r})\exp\left(-\frac{K}{10}\right)\right).
$$
(163)

So, since $\bar{r} = \min\{\frac{32B}{T}\log(TK), \frac{1}{5\alpha}\}$, and using the fact that, for $\zeta, a, b > 0$, $e^{-\zeta \min\{a,b\}} \leq e^{-\zeta a} + e^{-\zeta b}$, we have, for a constant $c''$,

$$\mathbb{E}F(x_T) - F^* \leq c'' \cdot \left( HB^2 \exp\left(-\frac{T}{160\alpha B}\right) + \frac{HB^2}{TK} + \frac{\sigma B}{\sqrt{MK}} + \frac{HB^2 \log TK}{TK} \right.$$
$$\left. + \frac{\rho B^2 \log TK}{T\sqrt{K}} + HB^2 \log(TK) \exp\left(-\frac{K}{10}\right) \right) \tag{164}$$

$$\leq c'' \cdot \left( HB^2 \exp\left(-\frac{T}{160\alpha B}\right) + \frac{\sigma B}{\sqrt{MK}} + \frac{HB^2 \log TK}{TK} \right.$$
$$\left. + \frac{\rho B^2 \log TK}{T\sqrt{K}} + HB^2 \log(TK) \exp\left(-\frac{K}{10}\right) \right), \tag{165}$$

where the last inequality follows from the fact $\log(TK) \geq 1$, since $TK \geq 175$.

Finally, each call to Algorithm 2 requires at most $C \lceil \log N \rceil$ rounds of communication (one for each call to Algorithm 3). Therefore, we can implement up to $R/(C \lceil \log N \rceil)$ iterations of Algorithm 1 using our $R$ rounds of communication. We recall that $N$ and $C$ are set as

$$N = \left\lceil 1 + \frac{5}{2} \log \frac{H(B + 5T\bar{r})}{3\bar{\lambda}\bar{r}} \right\rceil \tag{166}$$

$$C = \left\lceil 8 \log\left( \lceil \log_2 N \rceil \left( 4 + \frac{eH}{\bar{\lambda}} + \frac{80H(B + 5T\bar{r})}{\bar{\lambda}\bar{r}} \right) \right) \right\rceil. \tag{167}$$

Therefore, we need to choose $T$ such that $T \leq \frac{R}{C \lceil \log N \rceil}$. To provide an explicit lower bound on how large $T$ can be, we therefore lower bound the right hand side. First, we have

$$N = \left\lceil 1 + \frac{5}{2} \log \frac{H(B + 5T\bar{r})}{3\bar{\lambda}\bar{r}} \right\rceil \tag{168}$$

$$\leq 2 + \frac{5}{2} \log \frac{H\left(B + 5T\frac{B}{T}\log(TK)\right)}{3\frac{2eH}{K-2}\min\{\frac{32B}{T}\log(TK), \frac{1}{5\alpha}\}} \tag{169}$$

$$\leq 2 + \frac{5}{2} \log \frac{BK\log(TK)}{e\min\{\frac{32B}{T}\log(TK), \frac{1}{5\alpha}\}} \tag{170}$$

$$\leq 2 + \frac{5}{2} \max\{\log(TK\log(TK)), \log(2\alpha BK\log(TK))\} \tag{171}$$

$$\leq 2 + 5\log(1 + 2\alpha B) + 5\log(TK). \tag{172}$$

Similarly,

$$C = \left\lceil 8 \log\left( \lceil \log_2 N \rceil \left( 4 + \frac{eH}{\bar{\lambda}} + \frac{80H(B + 5T\bar{r})}{\bar{\lambda}\bar{r}} \right) \right) \right\rceil \tag{173}$$

$$\leq 1 + 8\log\left( \lceil \log_2 N \rceil \left( 4 + \frac{eH}{\frac{2eH}{K-2}} + 240\max\{TK\log(TK), 2\alpha BK\log(TK)\} \right) \right) \tag{174}$$

$$\leq 1 + 8\log\left( \lceil \log_2 N \rceil \left( 4K + 240\max\{T^2K^2, 2\alpha BTK^2\} \right) \right) \tag{175}$$

$$\leq 1 + 8\log\left( (1 + \log N)\left( 353T^2K^2 + 693\alpha BTK^2 \right) \right) \tag{176}$$

$$\leq 1 + 8\log(693) + 16\log(TK) + 8\log(1 + \alpha B) + 8\log(1 + \log(N)) \tag{177}$$

$$\leq 54 + 16\log(TK) + 8\log(1 + \alpha B) + 16\log\log N \tag{178}$$

$$\leq 80 + 32\log(TK) + 24\log(1 + 2\alpha B). \tag{179}$$

Therefore,

$$\frac{R}{C\lceil \log N \rceil} \geq \frac{R}{(80 + 32\log(TK) + 24\log(1 + 2\alpha B))\lceil \log(2 + 5\log(1 + 2\alpha B) + 5\log(TK))\rceil} \tag{180}$$

$$\geq \frac{R}{(80 + 32\log(TK) + 24\log(1 + 2\alpha B))^2} \tag{181}$$

$$\geq \frac{R}{2048 + 2(80 + 32\log K + 24\log(1 + 2\alpha B))^2} \ . \tag{182}$$

So, it suffices to choose $T$ such that

$$T\log^2(T) \leq \frac{R}{2048 + 2(80 + 32\log K + 24\log(1 + 2\alpha B))^2} \ . \tag{183}$$

Note that equality holds for

$$T = \exp\left(2W\left(\left(\frac{R}{4096 + 4(80 + 32\log K + 24\log(1 + 2\alpha B))^2}\right)^{1/2}\right)\right),$$

where $W(\cdot)$ denotes the Lambert $W$ function. Thus, because $W(x) \geq \log(x) - \log\log(x)$ for $x \geq e$, and since we assume

$$R \geq \tilde{\Omega}(1) = e^2\left(4096 + 4(80 + 32\log K + 24\log(1 + 2\alpha B))^2\right),$$

we have that

$$\exp\left(2W\left(\left(\frac{R}{4096 + 4(80 + 32\log K + 24\log(1 + 2\alpha B))^2}\right)^{1/2}\right)\right)$$

$$\geq \exp\left(2\left(\log\left(\frac{R}{4096 + 4(80 + 32\log K + 24\log(1 + 2\alpha B))^2}\right)^{1/2}\right.\right.$$

$$\left.\left.- \log\log\left(\frac{R}{4096 + 2(80 + 32\log K + 24\log(1 + 2\alpha B))^2}\right)^{1/2}\right)\right)$$

$$= \frac{1}{4}\left(\frac{R}{4096 + 4(80 + 32\log K + 24\log(1 + 2\alpha B))^2}\right)$$

$$\cdot \log^2\left(\left(\frac{R}{4096 + 4(80 + 32\log K + 24\log(1 + 2\alpha B))^2}\right)\right).$$

Thus, letting

$$T = \left\lfloor \frac{1}{4}\left(\frac{R}{4096 + 4(80 + 32\log K + 24\log(1 + 2\alpha B))^2}\right)\right.$$

$$\left.\cdot \log^2\left(\left(\frac{R}{4096 + 4(80 + 32\log K + 24\log(1 + 2\alpha B))^2}\right)\right)\right\rfloor,$$

and using $\tilde{O}$ notation to hide polylogarithmic factors in $R$, $K$, and $\alpha B$, we have

$$\mathbb{E}[F(x_T)] - F^* \leq HB^2\left(\exp\left(-\frac{R}{\tilde{O}(\alpha B)}\right) + \exp\left(-\frac{K}{O(1)}\right)\right) + \tilde{O}\left(\frac{\sigma B}{\sqrt{MK}} + \frac{HB^2}{KR} + \frac{\rho B^2}{\sqrt{K}R}\right) \ . \tag{184}$$

This completes the proof. $\qquad\square$

# F   Additional Comparisons

Here we include additional comparisons, recalling that we we are working in the natural scaling of parameters that arises when the objective has the form $F(x) = \mathbb{E}_z \ell(\langle x, z \rangle)$, where $|\ell'|$, $|\ell''|$, and $|\ell'''|$ are $O(1)$, and where $\|z\| \le D$, which provides for setting the parameters as $H = D^2$, $\sigma = D$, $U = D^3$, $\alpha = D$, and $\rho = D^2$.

**Comparison with Local SGD**

Recall that, under third-order smoothness assumptions (Yuan and Ma, 2020), Local SGD converges at a rate of

$$\mathbb{E}F(\hat{x}) - F^* \le \tilde{O}\left( \frac{HB^2}{KR} + \frac{\sigma B}{\sqrt{MKR}} + \frac{U^{1/3}\sigma^{2/3}B^{5/3}}{K^{1/3}R^{2/3}} \right) .$$

For the setting as outlined above, this bound reduces to

$$\mathbb{E}F(\hat{x}) - F^* \lesssim \frac{D^2B^2}{KR} + \frac{D^2B^2}{KR^2} + \frac{D^{5/3}B^{5/3}}{K^{1/3}R^{2/3}} \approx \frac{D^2B^2}{KR} + \frac{D^{5/3}B^{5/3}}{K^{1/3}R^{2/3}} .$$

In the case where $DB$ is not too large ($DB \lesssim K^2 R$), the dominant term for Local SGD is $\frac{D^{5/3}B^{5/3}}{K^{1/3}R^{2/3}}$, and so we see that our algorithm improves upon Local SGD as long as $R \gtrsim \frac{D^{1/3}B^{1/3}}{K^{1/6}}$.

**Comparison with min-max method under Assumption 1**

Woodworth et al. (2021) identified the min-max optimal (up to logarithmic factors) stochastic first-order method in the distributed setting we consider, and under Assumption 1. Namely, a min-max optimal method can be obtained by combining Minibatch-Accelerated-SGD (Cotter et al., 2011), which enjoys a guarantee of

$$\mathbb{E}F(\hat{x}) - F^* \le O\left( \frac{HB^2}{R^2} + \frac{\sigma B}{\sqrt{MKR}} \right) ,$$

with Single-Machine Accelerated SGD, which runs $KR$ steps of an accelerated variant of SGD known as AC-SA (Lan, 2012), and enjoys a guarantee of

$$\mathbb{E}F(\hat{x}) - F^* \le O\left( \frac{HB^2}{K^2R^2} + \frac{\sigma B}{\sqrt{KR}} \right) .$$

Therefore, an algorithm which returns the output of Minibatch Accelerated SGD when $K \le \frac{\sigma^2 R^3}{H^2 B^2}$, and otherwise returns the output of Single-Machine Accelerated SGD, achieves a guarantee of

$$\mathbb{E}F(\hat{x}) - F^* \le O\left( \frac{H^2B^2}{K^2R^2} + \frac{\sigma B}{\sqrt{MKR}} + \min\left\{ \frac{HB^2}{R^2}, \frac{\sigma B}{\sqrt{KR}} \right\} \right) .$$

As shown by Woodworth et al. (2021), this matches the lower bound for stochastic distributed first-order optimization under Assumption 1, up to $O(\log^2 M)$ factors.

For the setting as outlined above, this bound reduces to

$$\mathbb{E}F(\hat{x}) - F^* \lesssim \frac{D^2B^2}{K^2R^2} + \frac{D^2B^2}{KR^2} + \min\left\{ \frac{D^2B^2}{R^2}, \frac{DB}{\sqrt{KR}} \right\} \approx \frac{D^2B^2}{K^2R^2} + \min\left\{ \frac{D^2B^2}{R^2}, \frac{DB}{\sqrt{KR}} \right\} .$$

Thus, for $DB \lesssim \frac{R^{3/2}}{\sqrt{K}}$, the dominant term is $\frac{D^2B^2}{R^2}$, in which case FEDSN is better as long as $R \lesssim \sqrt{K}$. Furthermore, for $\frac{R^{3/2}}{\sqrt{K}} \lesssim DB \lesssim K^{3/2}R^{3/2}$, the dominant term is $\frac{DB}{\sqrt{KR}}$, in which case FEDSN is better as long as $R \gtrsim D^2B^2$. In these regimes, we see that Assumption 2 allows FEDSN to improve over the best possible when relying only on Assumption 1. We also note that when $DB \gtrsim K^{3/2}R^{3/2}$, the combined algorithm described above is better than the guarantee we prove for FEDSN—we do not know whether this is a weakness of our analysis or represents a true deficiency of FEDSN in this regime.

# G  Additional Information for Experiments

## G.1  Baselines

We first present a more practical version of our algorithm FEDSN, called FEDSN-LITE. It differs from FEDSN in two major ways: first, it directly uses REGULARIZED-QUADRATIC-SOLVER (Algorithm 3) (with $\bar{\xi} = 1$) as an approximate quadratic solver without requiring a search over the regularization parameter as in Algorithm 2; and in addition, it scales the Newton update with a stochastic approximation of the Newton decrement (see Nesterov (1998); Boyd and Vandenberghe (2004)) and a constant stepsize $\nu$ (we use $\nu = 1.25$ throughout our experiments). Because of these changes, the Newton update is fairly robust to the choice of $\nu$, so we then only need to tune the learning rate of REGULARIZED-QUADRATIC-SOLVER, making it as usable as any other first-order method.

---

**Algorithm 6** FEDSN-LITE($x_0$)

---

(Operating on objective $F(\cdot)$ with stochastic gradient $g(\cdot; \cdot)$ and Hessian-vector product $h(\cdot; \cdot, \cdot)$
  oracles.)
**Input:** $x_0 \in \mathbb{R}^d$.
**Hyperparameters:** $T$: main iterations; $\nu$: Newton stepsize scale; and $\beta$: momentum.
    **for** $t = 0, 1, \dots, T - 1$ **do**
        REGULARIZED-QUADRATIC-SOLVER($x_t, 0$)
        $\nu_t := \nu \left(1 + (\Delta \tilde{x}_t^\top h(x_t, \Delta \tilde{x}_t; z))^{1/2}\right)^{-1}$        $\triangleright$ $h(x_t, \Delta \tilde{x}_t; z)$ is s.t. $\mathbb{E}_z[h(x_t, \Delta \tilde{x}_t; z)] =$
        $\nabla^2 F(x_t) \Delta \tilde{x}_t$ [4]
        Update: $x_{t+1} = x_t + \nu_t \Delta \tilde{x}_t$
    **Return:** $x_T$

---

Note that in each step of FEDSN-LITE, the subroutine REGULARIZED-QUADRATIC-SOLVER approximately solves a quadratic subproblem using a variant of one-shot averaging (we set the parameter functions as $\eta(\lambda) = \eta$ and $w(\lambda) = \frac{1}{K}$). Moreover, we implement FEDSN-LITE such that each call to the stochastic oracle from within REGULARIZED-QUADRATIC-SOLVER uses only a single sample (Case 2 in Algorithm 4), and so it is asymptotically as expensive as the gradient oracle (i.e., $O(d)$, if $d$ is the dimension of the problem). For a discussion of the computational cost of using a Hessian-vector product oracle, see Appendix G.4.

We have compared FEDSN-LITE against the two variants of FEDAC (Yuan and Ma, 2020), Minibatch SGD (Dekel et al., 2012), and Local SGD (Zinkevich et al., 2010). Two settings of hyperparameters are considered for FEDAC in Yuan and Ma (2020) for strongly convex functions:

- **FEDAC-I**: $\eta \in (0, 1/H]$, $\gamma = \max\left\{\sqrt{\frac{\eta}{\lambda K}}, \eta\right\}$, $\alpha = \frac{1}{\gamma \lambda}, \beta = \alpha + 1$;

- **FEDAC-II**: $\eta \in (0, 1/H]$, $\gamma = \max\left\{\sqrt{\frac{\eta}{\lambda K}}, \eta\right\}$, $\alpha = \frac{3}{2\gamma\lambda} - \frac{1}{2}, \beta = \frac{2\alpha^2 - 1}{\alpha - 1}$,

where $H$ is the smoothness constant as in Assumption 1, $\lambda$ is an estimate of the strong convexity, and $\eta$ is the learning rate which has to be tuned. Thus, a limitation of FEDAC is that it requires either the knowledge of $\lambda$ (say, through explicit $\lambda$-regularization), or that the algorithm adds regularization to the objective itself (this is how Yuan and Ma (2020) present FEDAC for general convex functions). In our experiments we have both of these settings, i.e., FEDAC with internal regularization $\lambda$ (c.f., figures 1a and 1b) or explicitly $\lambda$-regularized objectives (c.f., figures 4 and 5). For brevity, in Algorithm 7 we present FEDAC with five hyperparameters: $\alpha, \beta, \eta, \gamma$, and $\lambda$. When the objective is regularized we use $\lambda = 0$, whereas otherwise we tune $\lambda$, to ensure the best possible performance for FEDAC.

---

[4]We can calculate an approximation to the Newton decrement at $\Delta \tilde{x}_t$, i.e., $(\Delta \tilde{x}_t^\top \nabla^2 F(x_t) \Delta \tilde{x}_t)^{1/2}$, using a call to the Hessian-vector product oracle (see Assumption 2(b)), along with an additional dot product, namely $\Delta \tilde{x}_t^\top h(x_t, \Delta \tilde{x}_t; z)$.

---

**Algorithm 7** FEDAC$(x_0, \alpha, \beta, \eta, \gamma, \lambda)$

---

(Operating on objective $F(\cdot) + \lambda/2\|.\|^2$ as opposed to $F(.)$ with stochastic gradient oracle $g_\lambda(\cdot; \cdot)$[5].)

   **Intitialize:** $x_0^{ag,m} = x_0^m = x_0$ for all $m \in [M]$

   **for** $t = 0, 1, \ldots, T-1$ **do**

      **for** every worker $m \in [M]$ **in parallel do**

         $x_t^{md,m} \leftarrow \beta^{-1} x_t^m + (1 - \beta^{-1}) x_t^{ag,m}$

         $g_t^m \leftarrow g_\lambda(x_t^{md,m}, z_t^m)$        $\triangleright$ Query the stochastic first-order oracle at $x_t^{md,m}$, for $z_t^m \sim \mathcal{D}$

         $v_{t+1}^{ag,m} \leftarrow x_t^{md,m} - \eta \cdot g_t^m$

         $v_{t+1}^m \leftarrow (1 - \alpha^{-1}) x_t^m + \alpha^{-1} x_t^{md,m} - \gamma g_t^m$

         **if** $t \bmod K = -1$ **then**

            $x_{t+1}^m \leftarrow \frac{1}{M} \sum_{m'=1}^{M} v_{t+1}^{m'}$

            $x_{t+1}^{ag,m} \leftarrow \frac{1}{M} \sum_{m'=1}^{M} v_{t+1}^{ag,m'}$

         **else**

            $x_{t+1}^m \leftarrow v_{t+1}^m$

            $x_{t+1}^{ag,m} \leftarrow v_{t+1}^{ag,m}$

   **Return:** $\bar{x}_T^{ag} = \frac{1}{M} \sum_{m'=1}^{M} x_T^{ag,m'}$

---

In Algorithm 8 we describe Local SGD (a.k.a. FEDAVG) (Zinkevich et al., 2010) with learning rate $\eta$ and Polyak's momentum (a.k.a. heavy ball method) parameter $\beta$. Setting $\beta = 0$ recovers the familiar algorithm as analyzed in Woodworth et al. (2020a). Finally, in Algorithm 9 we describe Minibatch SGD with fixed learning rate $\eta$ and momentum parameter $\beta$.

---

**Algorithm 8** Local SGD$(x_0, \beta, \eta)$

---

   (Operating on objective $F(\cdot)$ with stochastic gradient oracle $g(\cdot; \cdot)$.)

   **Intitialize:** $x_{0,0}^m \leftarrow x_0$ for all $m \in [M]$

   **for** $r = 0, \ldots, R-1$ **do**

      **for** every worker $m \in [M]$ **in parallel do**

         **for** $k = 0, \ldots, K-1$ **do**

            $g_{r,k}^m \leftarrow g(x_{r,k}^m; z_{r,k}^m)$ $\triangleright$ Query the stochastic first-order oracle at $x_{r,k}^m$ (for $z_{r,k}^m \sim \mathcal{D}$ drawn by the oracle)

            $x_{r,k+1}^m \leftarrow x_{r,k}^m - \eta g_{r,k}^m + \mathbb{1}_{k>0} \beta(x_{r,k}^m - x_{r,k-1}^m)$

    $x_{r+1} \leftarrow \frac{1}{M} \sum_{m'=1}^{M} x_{r,K}^{m'}$

   **Return:** $x_R$

---

**Algorithm 9** Minibatch SGD$(x_0, \beta, \eta)$

---

   (Operating on objective $F(\cdot)$ with stochastic gradient oracle $g(\cdot; \cdot)$.)

   **Intitialize:** $x_0^m = x_0$ for all $m \in [M]$

   **for** $r = 0, \ldots, R-1$ **do**

      **for** every worker $m \in [M]$ **in parallel do**

         **for** $k = 0, \ldots, K-1$ **do**

            $g_{r,k}^m \leftarrow g(x_r; z_{r,k}^m)$ $\triangleright$ Query the stochastic first-order oracle at $x_r$ (for $z_{r,k}^m \sim \mathcal{D}$ drawn by the oracle)

    $g_r \leftarrow \frac{1}{KM} \sum_{k'=0}^{K-1} \sum_{m'=1}^{M} g_{r,k'}^{m'}$

    $x_{r+1} \leftarrow x_r - \eta g_r + \mathbb{1}_{r>0} \beta(x_r - x_{r-1})$

   **Return:** $x_R$

---

Note that in our experiments we compare the algorithms using the same number of machines $M$, communication rounds $R$ and theoretical parallel runtime $T$ against each other, where $T = KR$.

---

[5]Note that $g_\lambda(x; z) = g(x; z) + \lambda x$, i.e., the stochastic oracle for the regularized objective can always be obtained using the oracle for the unregularized objective (see Assumption 1).)

Also note that unlike Algorithm 7, there is no internal regularization in Algorithms 8 and 9. While conducting experiments for figures 4 and 5, we assume that $F(.)$ is regularized, so that Algorithms 8 and 9 instead minimize $F(.) + \frac{\lambda}{2}\|.\|^2$ and have access to $g_\lambda(.;.)$. The reason why $\lambda$ is not presented as a hyperparameter for any algorithms beside FEDAC, is because the regularization is a part of the optimizer FEDAC itself determines its other hyper-parameters.

In addition, we may see that REGULARIZED-QUADRATIC-SOLVER with $\eta_k(\lambda) = \eta$ and $w_k(\lambda) = \frac{1}{K}$ (as in each iteration of FEDSN-LITE) is a special case of Local SGD (Algorithm 8) with $R = 1$. Thus to summarize, in our experiments, we have compared the following algorithms:

- FEDSN-LITE (which uses REGULARIZED-QUADRATIC-SOLVER) without momentum i.e., $\beta = 0$ or with optimally tuned momentum $\beta \in \{0.1, 0.3, 0.5, 0.7, 0.9\}$,
- FEDAC-1 and FEDAC-2, with either no internal regularization i.e., $\lambda = 0$ (in experiment 2 when the objective is explicitly regularized) or optimally tuned internal regularization $\lambda \in \{1e\text{-}2, 1e\text{-}3, 1e\text{-}4, 1e\text{-}5, 1e\text{-}6\}$ (in experiment 1),
- Local SGD without momentum i.e., $\beta = 0$ or with tuned momentum $\beta \in \{0.1, 0.3, 0.5, 0.7, 0.9\}$,
- Minibatch SGD without momentum i.e., $\beta = 0$ or with tuned momentum $\beta \in \{0.1, 0.3, 0.5, 0.7, 0.9\}$,

We search for the best learning rate $\eta \in \{0.0001, 0.0002, 0.0005, 0.001, 0.002, 0.005, 0.01, 0.02, 0.05, 0.1, 0.2, 0.5, 1, 2, 5, 10, 20\}$ for every configuration of each algorithm. We verify (along with Yuan and Ma (2020)) that the optimal learning rate always lies in this range.

### G.2 More extensive experiments

First, we present more comprehensive versions of the experiments presented in Section 5.

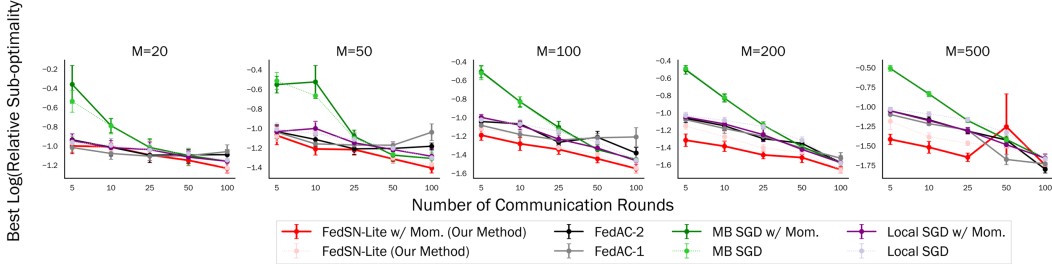

Figure 2: The same experiment as in fig. 1a but with a broader range of values of $M$. All optimization runs were repeated 30 times.

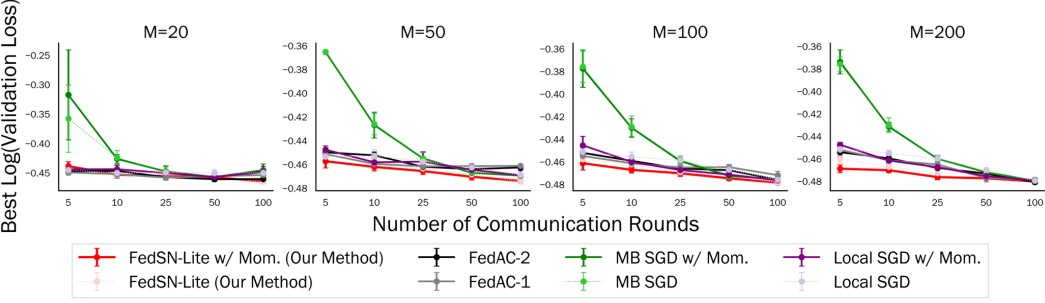

Figure 3: The same experiment as in fig. 1b but with a broader range of values of $M$. All optimization runs were repeated 20 times.

Recall, how comparison to FEDAC requires the knowledge of the strong convexity constant. To alleviate this in figures 2 and 3 we added an internal regularization parameter to FEDAC and tuned

it. Another way to deal with this is to solve a regularized empirical risk minimization problem, where FEDAC knows the level of regularization, which serves as a proxy for strong convexity. Next we consider exactly this setting in figures 4 and 5, where we provide FEDAC the regularization strength $\mu$. Unlike previous experiments, here we report regularized training loss, i.e., we train the models to optimize $F(x) + r(x) = \sum_{i \in S} f_i(x) + \frac{\mu}{2}\|x\|_2^2$, for a finite dataset $S$. We also vary the regularization strength $\mu$, to understand the algorithms' dependence on the problem's conditioning. This was precisely the experiment conducted by Yuan and Ma (2020) (c.f., Figures 3 and 5 in their paper).

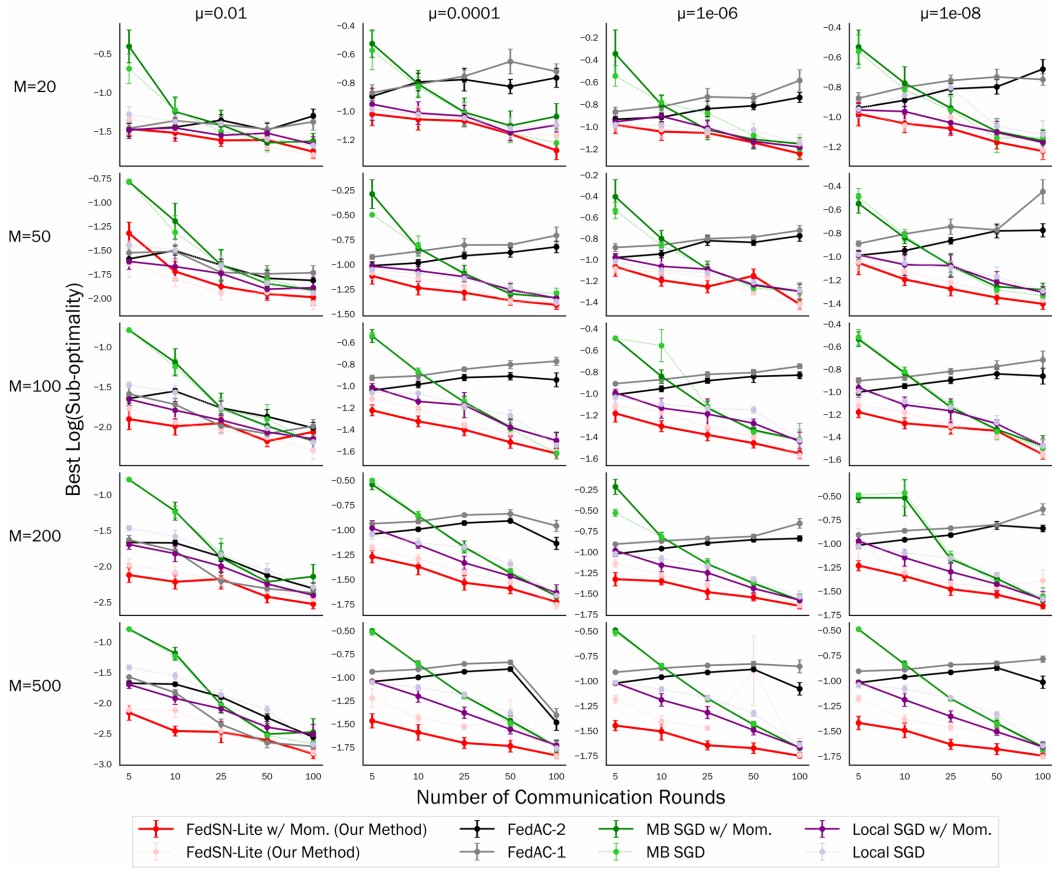

Figure 4: Empirical comparison of FEDSN-LITE (Algorithm 6) to other methods (see Appendix G.1 for complete details) on the LIBSVM a9a (Chang and Lin, 2011; Dua and Graff, 2017) dataset for minimizing $\mu$-regularized logistic regression loss using $M$ machines and $\mu$ regularization strength. We vary the frequency of communication (horizontal axis of each plot), while keeping the total number of steps on each machine (theoretical parallel runtime) fixed at $KR = 100$. Each algorithm besides FEDAC solves a regularized ERM problem, and reports the best relative sub-optimality w.r.t. the optimal minimizer. For FEDAC we use $\mu$ as the strong convexity constant to tune its hyperparameters. For the other algorithms we tune the learning rate for either $\beta = 0$, i.e., without momentum, or for all $\beta \in \{0.1, 0.3, 0.5, 0.7, 0.9\}$, to choose the optimum level of momentum. We repeat this tuning procedure for each value of $\mu, M, R$, and thus each point in the plot represents an optimal configuration of that algorithm for that setting. All optimization runs were repeated 70 times with the tuned hyperparameters.

## G.3 More implementation details

**Dataset.** Following the setup in Yuan and Ma (2020), we use the full LIBSVM a9a (Chang and Lin, 2011; Dua and Graff, 2017) dataset. It is a binary classification dataset with 32,561 points and 123 features. For generating figures 1a, 2, 4 and 5 we use the entire dataset as a training set. On the other hand, For generating figures 1b and 3 we split the dataset, using 20,000 points as the training set and the rest as the validation set.

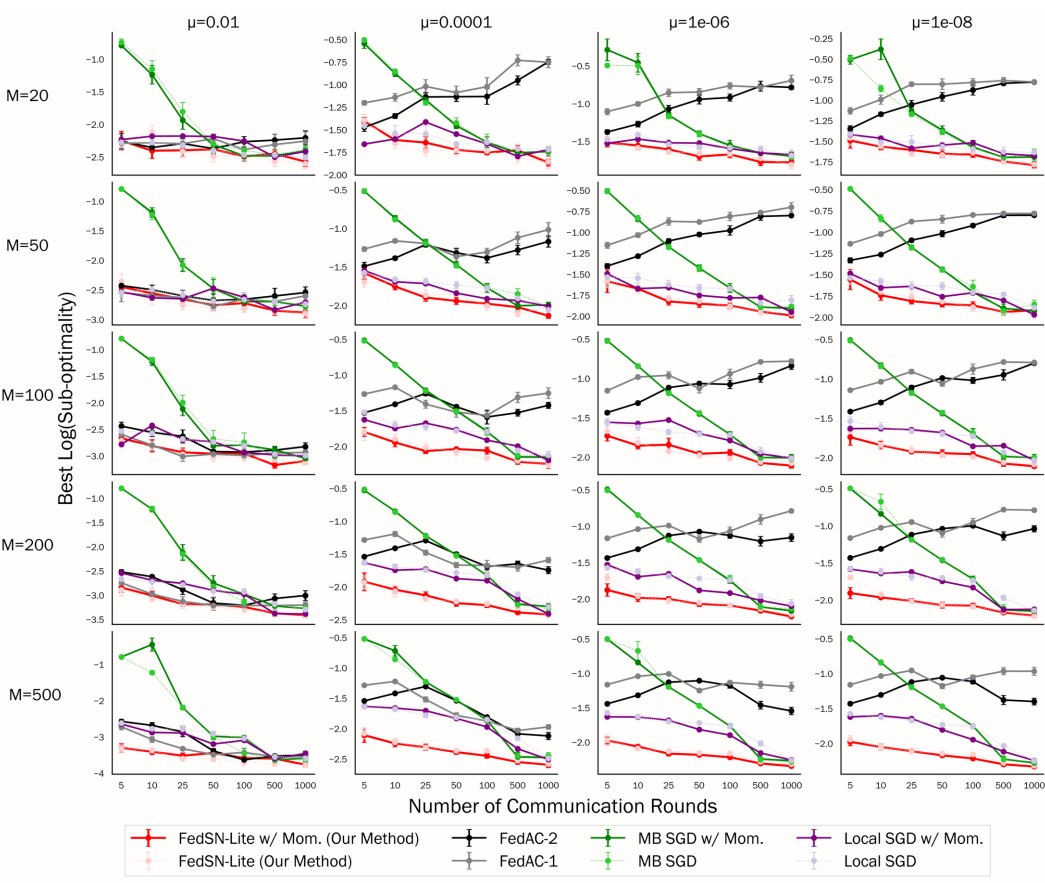

Figure 5: Same setting as figure 4 but with a longer parallel run-time of $T = 1000$.

**Figures 4 and 5.** As described in the body of the paper, each sub-plot (in Figure 4) shows the performance of the different algorithms and their variants (as discussed above) on $\mu$-regularized logistic regression. For the experiments we vary the values of $\mu$, $M$, and $R$, for fixed theoretical parallel runtime $KR$. Each of these is a different *setting*, reflecting different computation-communication-accuracy trade-offs. For each of these settings, we run an algorithm configuration (with some $\beta$ and $\lambda = 0$ for FEDAC), with different learning rates $\eta$, and record the best loss (which is the Regularized ERM-loss on the entire dataset) obtained *at any point* during optimization. Based on this run we pick the optimal learning rate for each setting for each algorithm from the set $\{0.0001, 0.0002, 0.0005, 0.001, 0.002, 0.005, 0.01, 0.02, 0.05, 0.1, 0.2, 0.5, 1, 2, 5, 10, 20\}$.

Note that this means every point in a sub-plot is an individual experiment, for which we have tuned. Since we are only concerned with optimization, we tune and report the accuracy only on the training set (with 32,561 points). Both the stochastic first-order oracle and the Hessian-vector product oracle, used by the respective algorithms, do sampling with replacement. Following Yuan and Ma (2020), we ensure that our learning rate tuning grid was both wide and fine enough for the algorithms to achieve their optimal training losses. Since we are using stochastic algorithms which sample with replacement, without any definite order on the dataset (as opposed to making a single pass), the best suboptimality is potentially different for each run of the algorithm, even when choosing the same learning rate and initialization. Thus, once we have the optimal parameters $(\eta, \beta, \lambda)$, we rerun the algorithm multiple times on the corresponding setting. The error bars represent the standard deviation of the best suboptimality obtained when using this optimal learning rate.

Note that changing the regularization strength changes the optimal error value on the RERM task. Thus, to get the optimal loss value, we run exact Newton's method separately on each of these settings (for different values of $\mu$), until the algorithm had converged up to $\approx 12$ decimal places. We also ensure that our optimal loss values look similar to Yuan and Ma (2020). The reported suboptimality

in Figure 1a is obtained after subtracting this optimal error value from the algorithms' error, followed by dividing this excess error with the optimal value.

**Figures 1a and 2.** As described in the body of the paper, each sub-plot in Figure 1a shows the performance of the different algorithms and their variants (as discussed above) on logistic regression for getting the best possible unregularized training loss. We train FEDAC on an appropriately regularized RERM problem on the training set. This is achieved through its internal regularization parameter $\lambda$ as described above. All other algorthms minimize an ERM problem on the dataset. We vary $M$ and $R$ while keeping the parallel runtime $KR$ to be fixed. Our hyperparameter tuning and repetitions are similar to the description for the experiment above, though we additionally tune the regularization strength $\lambda$ in the RERM problem for FEDAC. All the algorithms do sampling with replacement on the training set, and can make multiple passes on the dataset. The optimal training loss is again obtained using 100 iterations of the Newton method.

**Figures 1b and 3.** As described in the body of the paper, each sub-plot in Figure 1b shows the performance of the different algorithms and their variants (as discussed above) on logistic regression for getting the best possible validation loss. We train FEDAC on an appropriately regularized RERM problem on the training set. All other variants minimize an ERM problem on the training set. This is achieved through the internal regularization parameter for FEDAC as discussed above. We vary $M$ and $R$ while keeping the parallel runtime $KR$ to be fixed. Our hyperparameter tuning and repetitions are similar to the description for the experiments above, though we additionally tune the regularization strength $\lambda$ in the RERM problem for FEDAC. We use the same split of training and validation datasets across the multiple repetitions. All the algorithms do sampling without replacement on the training set, and can make at most one pass on the training dataset.

Note that in all our experiments, we are concerned with minimizing a convex function $F(x)$. In some cases we are supposedly minimizing a finite sum, i.e., $F(x) = \sum_{i \in S} f_i(x)$ where $S$ is our training dataset, for e.g., in Figure 1a. In others $F(x) = \mathbb{E}_{z \in \mathcal{D}}[f(x; z)]$, i.e., a stochastic optimization problem where we access the distribution $\mathcal{D}$ through our finite sample $S$, for e.g., in Figure 1b. To estimate the true-error on $\mathcal{D}$, we split $S$ into $S_{train}$ and $S_{val}$, then sampled without replacement from $S_{train}$ to train our models, and reported the final performance on $S_{val}$.

**Hardware details.** All the experiments were performed on a personal computer, Dell XPS 7390. The total compute time (CPU) on the machine was about 300 hours. No GPUs were used in any of the experiments in this paper.

### G.4 Computational cost of a Hessian-vector product oracle

Consider minimizing the function $F(x) = \mathbb{E}_z[f(x; z)]$ given access to various stochastic oracles. Note that FEDAC, Minibatch SGD, and Local SGD are all first-order algorithms, in that they use a first-order stochastic oracle. Thus, each time they observe a stochastic gradient (as per Assumption 1), the oracle uses a single unit of randomness (e.g., a single data point, when thinking in terms of a finite training dataset).

In contrast, our main theoretical method FEDSN proceeds via two possible options: for Case 1 as established in Algorithm 4, the algorithm queries both a stochastic gradient and a stochastic Hessian-vector product oracle at two independent samples, while in Case 2, we consider a different setting which allows the algorithm to observe, for any $x, u$, both a stochastic gradient and stochastic Hessian-vector product using the *same* random sample $z$. We may note that the final theoretical guarantee for Case 1 differs by only a small constant factor from that of Case 2, and our results as presented in Theorem 1 apply to both cases.

Thus, in order to maintain a fair comparison between these first-order methods and our practical method FEDSN-LITE, we have implemented FEDSN-LITE so that it also uses a single random sample (single data point), as outlined via Case 2 (Same-Sample) in Algorithm 4.

In addition to keeping the number of samples consistent, it is important to understand how both oracle models compare computationally. Clearly every oracle call for FEDSN-LITE is at least as expensive as a first-order stochastic oracle, as it subsumes the latter. Moreover, it is unclear a priori if the Hessian-vector product can be computed efficiently (say, in as much time as vector addition or multiplication). However, it turns out that the Hessian-vector product for logistic regression can be efficiently computed, since the Hessian matrix for a given sample is actually rank one

(i.e., an outer product of known vectors). This is generally true for loss functions which belong to the family of *generalized linear models*. To see this, note that for loss functions of the form $F(x) = \sum_{i=1}^{N} \phi(b_i x^\top a_i)$, we have by a simple calculation that

$$\nabla_x F(x) = \sum_{i=1}^{N} \phi'(b_i x^\top a_i) b_i \cdot a_i \quad \text{and} \quad \nabla_x^2 F(x) := \nabla_x(\nabla_x F(x)) = \sum_{i=1}^{N} \phi''(b_i x^\top a_i) b_i^2 \cdot a_i a_i^\top,$$

and so for any $v \in \mathbb{R}^d$,

$$\nabla_x^2 F(x) v = \sum_{i=1}^{N} \phi''(b_i x^\top a_i) b_i^2 \cdot a_i \cdot (a_i^\top v),$$

which means each summand can be calculated in $\mathcal{O}(d)$ time. When maximizing the log-likelihood in the logistic regression model with labels in $\{-1, 1\}$, we need to minimize the following function,

$$F(x) = \sum_{i=1}^{N} \left( -b_i x^\top a_i + \ln \left( 1 + \exp \left( b_i x^\top a_i \right) \right) \right),$$

which is an instance of the generalized linear models as considered above. Thus, in terms of vector operations, the Hessian-vector product oracle and the stochastic gradient oracle are *asymptotically similar* in the logistic regression model. We also note that for a general class of differentiable functions, Pearlmutter (1994) provides a technique to compute the Hessian-vector product using two passes of backpropagation (in the context of neural networks). For instance, we may consider a twice-differentiable function $F : \mathbb{R}^d \mapsto \mathbb{R}$ and note that for a vector $v \in \mathbb{R}^d$,

$$\nabla_x^2 F(x) v = \nabla_x \left( \nabla_x F(x)^\top v \right),$$

which can be obtained using two passes of backpropagation.

For the scale of our problem, it turns out that the difference in the number of vector operations is outweighed by other implementational overhead (for, e.g., loops, memory operations, etc.). In Table 5 we show the average per-step runtimes (over 250 runs) for three different algorithms for $M = 1, R = 1, K = 100000$.

| Algorithm | Avg. Runtime/Step (in $10^{-5}$ sec.) | Std. Dev. of Runtime/Step (in $10^{-5}$ sec.) |
|---|---|---|
| FEDSN-LITE | 6.67 | 1.42 |
| Local SGD | 6.39 | 1.37 |
| FEDAC | 7.48 | 1.58 |

Table 5: Comparing the wall-clock runtimes of different algorithms in our implementation. We run every repetition of each algorithm as an individual job, so that the runs are *independent*, i.e., we don't introduce extraneous biases for any one algorithm (e.g., thermal throttling affecting the runtimes for a single algorithm). We run each algorithm 1700 times so that the 95% error margins for the average runtime/step estimate are within 1% of its value. Specifically, we calculate the empirical standard deviation $\sigma$ of our average runtime/step estimate $\bar{x}$ by running it for $n = 1700$ runs. Then we report the error margin $\delta = z\sigma/\sqrt{n}$, where $z = 1.96$, is the z*-value from the standard normal distribution for a 95% confidence level. We ensure that $n = 1700$ is large enough, so that $\delta/\bar{x} \le 0.01$. Finally we round $\bar{x}$ to two significant digits, respecting the 1% error margin. We also report the standard deviation estimate of the runtime/step so that its 95% error margins are within 4% of its value.

### G.5   Comparison with GIANT

Recall that we are optimizing a convex function $F$, which is accessible only through stochastic oracles in an online fashion. To do so, we implement an inexact Newton method, where we reformulate the Newton step as the solution to a convex quadratic problem, $\min_{\Delta x} \frac{1}{2} \Delta x^\top \nabla^2 F(x) \Delta x + \nabla F(x)^\top \Delta x$. We then solve this problem using one-shot averaging, with access to stochastic gradient and Hessian-vector product oracles. If instead we were solving a batch problem, where we had a finite dataset, we could use it to determine the exact Newton step.

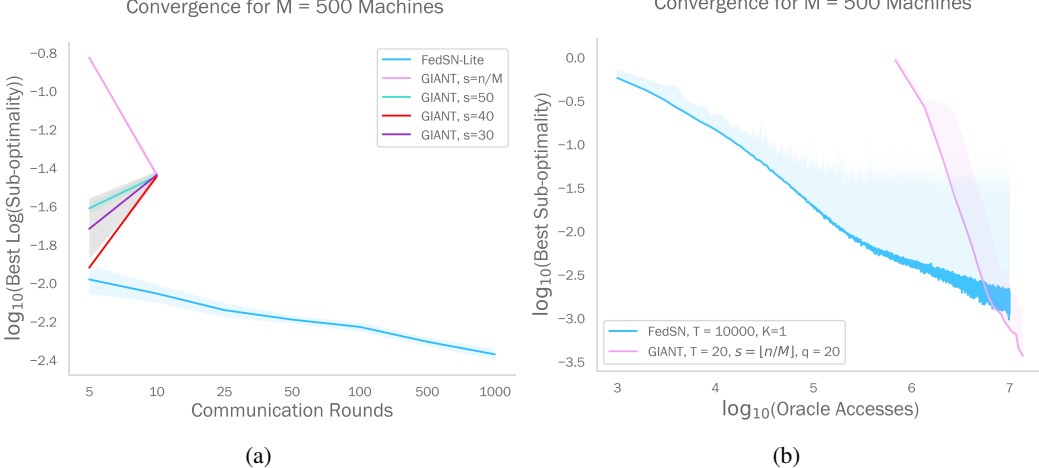

Figure 6: (a) Empirical comparison between GIANT and FEDSN-LITE, when equalizing for the number of data-accesses per communication round. We vary the frequency of communication (horizontal axis of each plot), while keeping the total number of steps on each machine (theoretical parallel runtime) fixed at $KR = 1000$ for FEDSN-LITE. We choose $s, q$ such that $n/M + sq = K$ for GIANT. For $R > 10$, GIANT could not be run in this setup, as $q$ was less than one. Note that this is the same reason why GIANT performs worse going from $K = 200$ to $K = 100$, as the value of $q$ gets closer to one. We suspect that this worsening of performance does not happen for $s = \lfloor n/M \rfloor$, as in that case the effect of increasing Newton steps is more dominant than decreasing $q$. All experiments were repeated 20 times. (b) Empirical comparison between GIANT and FEDSN-LITE when keeping the number of oracle accesses equal. For both the methods we count the stochastic oracle access as well as the Hessian-vector product oracle access as two oracle accesses. The dark lines depict the minimum error obtained across multiple runs, and the error strips denote the maximum error across those runs.

More specifically, if $F(x) = \frac{1}{n} \sum_{i \in [n]} f(x; z_i)$, where $z_i$ indexes an example in the training dataset, then we can obtain a sketch of the Hessian matrix $\nabla^2 \tilde{F}_m(x)$ on each machine, by distributing the examples across $M$ machines. Then we can solve the linear system $\nabla^2 \tilde{F}_m(x) p_m = \nabla F(x)$ where $\nabla F(x)$ could be computed exactly and broadcasted to each machine. The linear system can be effectively solved with a conjugate gradient method. Finally the directions $p_m$ can be aggregated to get an approximate Newton direction. This was precisely the idea proposed by Wang et al. (2018), in their algorithm called GIANT. In fact, there has been much work in the batch setting (Shamir et al., 2014; Zhang and Xiao, 2015; Reddi et al., 2016; Crane and Roosta, 2019; Islamov et al., 2021; Gupta et al., 2021) similar to this, but as we point out before, we work in a strictly more difficult setting. The ability to index the examples in the training set, and possibly re-use them in multiple rounds is commonly used in the batch setting, but it cannot be exploited in the online setting (except perhaps with an additional generalization error when $F(x) := \mathbb{E}_{z \sim \mathcal{D}}[f(x; z)]$).

Moreover, typically the computation per communication round for batch methods is much higher than our method. For instance if GIANT uses $s$ examples on each machine for making a sketch of the Hessian and uses $q$ steps of the conjugate gradient method, then since it still needs to compute the full gradient per communication round, the computation per round is $\propto n + Msq$, as opposed to $KM$ for FEDSN-LITE. Nevertheless, we compared GIANT to our method for a range of values of $s$ in Figure 6a. We note that in the sparse communication regime we beat GIANT. If communication is not a bottleneck, we can also compare both these methods while keeping the number of oracle accesses equal. In such a setting, as we note in Figure 6b, FEDSN-LITE has an advantage initially (when GIANT is not even applicable), but eventually GIANT's updates are more accurate, and it outperforms FEDSN-LITE. The bottom line is that these methods perform very similar updates, i.e., inexact Newton steps, but FEDSN-LITE is applicable in the online setting while GIANT is not. Moreover, FEDSN-LITE can have an advantage in the sparse communication regime.