# OpenReview forum: "A Stochastic Newton Algorithm for Distributed Convex Optimization"
_NeurIPS.cc/2021/Conference — NeurIPS 2021 Poster_

### Official Review · Reviewer_Huyz · 2021-07-04

**Rating:** 6
**Confidence:** 3

**Summary:**

In this paper, the authors introduced a stochastic Newton-type algorithm for distributed quasi-self-concordant convex optimization. The high level idea is to reduce the problem into approximately solving for a sequence of trust-region constrained quadratic subproblems. The subproblems are in turn solved by binary search on the optimal regularization parameter $\lambda^*$, followed by calling for unconstrained stochastic local SGD solver with a single round of communication. The algorithm assumes access to stochastic hessian-vector product, in addition to stochastic gradient. Numerical experiments are conducted on a more practical variant of the algorithm for the logistic objective.

**Limitations And Societal Impact:**

Yes.

**Main Review:**

The relevant literature is adequately surveyed, although I'd add that the idea of solving the inner quadratic problem using stochastic gradient method has been considered before, see e.g., Corollary 18 in Carmon et al: https://proceedings.neurips.cc/paper/2020/file/dba4c1a117472f6aca95211285d0587e-Paper.pdf. The claims do look right and the writing is pretty clear. The analysis itself is a combination of somewhat standard techniques/building blocks from the optimization literature.

The improvement seems to be quite tied to the parameter regime however, and it's a bit elusive with the somewhat non-standard oracle/assumptions/missing evidence of lower bound how applicable/relevant the algorithm really is in practice. It would be good to know how the proposed algorithm work on common objectives on real data with (potentially unknown) quantities defined in the assumptions.

====================== post author-response ========================
Thank you for the given example and I'd like to keep the positive score.

**Time Spent Reviewing:**

2.5

---

> ### Author Response · Authors · 2021-08-10
> **Response to Reviewer Huyz**
>
> We thank the reviewer for their comments, and for pointing out Corollary 18 of Carmon et al.
>
> Regarding the parameter regime, in the experiments on the LIBSVM a9a dataset, we in fact do not know the quantities defined in the assumptions. Thus, to facilitate a fair experimental setup, we tune the learning rate for each algorithm by searching over a large grid of potential stepsizes. We include more implementation details in Appendix E of the submitted version.
>
> We further acknowledge the difficulty in comparing FedSN to previous first-order methods is determining the "typical" relative scale of these parameters. Therefore, we will include a discussion of a *natural scaling* of the parameters that arises when the objective has the form $F(x) = E_{z}l(\langle x,z \rangle)$, where $\vert\vert l’\vert\vert$, $\vert\vert l''\vert\vert$, and $\vert\vert l'''\vert\vert$ are $\mathcal{O}(1)$. This holds, as one example, for logistic regression. In this case, upper bounds on the derivatives of $F$ will generally scale with the norm of the data, i.e., $\vert\vert z\vert\vert$. So if we assume that $\vert\vert z\vert\vert \leq D$ for some $D$, then the derivatives of $F$ would generally scale as $\vert\vert\nabla F(x)\vert\vert \lesssim D$,  $\vert\vert\nabla^2 F(x)\vert\vert_{op} \lesssim D^2$, and $\vert\vert\nabla^3 F(x)\vert\vert_{op} \lesssim D^3$, where $\vert\vert\cdot\vert\vert_{op}$ denotes the operator norm. It follows that, in this setting, we may take $H=D^2$, $\sigma = D$, $Q = D^3$, $\alpha = D$, and $\rho = D^2$. Though these parameters could of course have different relationships, we will provide an overview of this particular regime to help clarify the comparison.

---

### Official Review · Reviewer_sbMG · 2021-07-06

**Rating:** 7
**Confidence:** 4

**Summary:**


This paper studies a distributed Stochastic Newton method in the intermittent communication model. At each step, the Newton direction is obtained by solving the associated minimization problem (instead of directly computing the Hessian) using One Shot Averaging (one round of communication only): nodes locally run a stochastic gradient algorithm and then average their solutions at the end. This is known to work well for quadratic objective, which is the case for computing the Newton direction.

Then, under reasonable higher order smoothness assumptions (quasi self-concordance, third order smoothness and bounded fourth central moment), it is shown that obtaining approximate solutions to a constrained version of the Newton subproblem  leads to a globally convergent algorithm. Thus, the solution to the constrained Newton subproblem is replaced by the equivalent regularized subproblem, where the unknown regularization parameter is approximated by binary search. Thus, an approximate solution to the constrained subproblem is obtained by solving unconstrained subproblems only, which can be handled using one shot averaging.

Then, an appropriate choice of parameters is given, which allows to prove convergence guarantees for the whole algorithm. These guarantees are compared to the lower bound, which can be beaten in some regimes thanks to the slightly different assumptions as well as the fact that stochastic Hessian-vector oracles are considered, which is reasonable for generalized linear models.

Finally, simulations are given to show that FedSN outperforms reasonable baselines (Local and Minibatch SGD in particular) in terms of Suboptimality for a given number of communication rounds.


**Limitations And Societal Impact:**

No foreseeable societal impact.

**Main Review:**


The paper is globally clear and well-written. The relevant state of the art is adequately reviewed, and the series of (informal) theorems present a clear logical progression for the different steps needed to analyze the algorithm.

Although the idea of approximating each Newton step using One Shot Averaging to reduce communications is quite natural, several tricks are required, such as approximating the constrained optimization problem. Besides, the analysis seems sound, and is based on strong but reasonable assumptions (local stability and quasi-self concordance in particular).

If I am not mistaken, all nodes need to draw stochastic gradients for the same function. Maybe this is worth mentioning (does not allow for data heterogeneity between nodes).

It seems that FedSN is globally convergent thanks to the constrained step, and then a complex scheme is developed to turn this into an unconstrained problem. In the end, the algorithm comes down to approximately solving a series of unconstrained Newton subproblems with varying regularization, similarly to, e.g., [A]. Is there a fundamental benefit of approximating constrained subproblems instead of a sequence of regularized Newton steps? If so, I believe it would be great to highlight it. Or is it the way [A] approximates the steps (directly via sketching) that makes the analysis viable without resorting to constrained subproblems? You don't assume strong convexity (but the experiments use regularized logistic regression), could that be an explanation?

You mention that the advantage of using Hessian-vector products is due to the distributed setting, and in particular the fact that the lower bound is not quadratic, so the Hessian-vector information is not equivalent to gradient information. Yet, if I understand correctly, this is also true for generalized linear models, which is the prime example for which stochastic Hessian vector products are easy to compute (and quasi-self concordance holds). From this perspective, it seems that using this kind of second-order information is mainly beneficial when it is not that easy to compute, or have I missed something? Is it possible that some slackness in the lower bound comes from different variance that is assumed for second order information (the \rho parameter)? I believe that the "comparison with the lower bound" part is great, but could be improved even further.

[A] Marteau-Ferey, Ulysse, Francis Bach, and Alessandro Rudi. "Globally Convergent Newton Methods for Ill-conditioned Generalized Self-concordant Losses." Advances in Neural Information Processing Systems 32 (2019): 7636-7646.

**Time Spent Reviewing:**

3

---

> ### Author Response · Authors · 2021-08-10
> **Response to Reviewer sbMG**
>
> We thank the reviewer for their positive and insightful comments and suggestions.
>
> >If I am not mistaken, all nodes need to draw stochastic gradients for the same function. Maybe this is worth mentioning (does not allow for data heterogeneity between nodes).
>
> This is correct. We consider the homogeneous distributed model (as also studied by e.g. Stich (2019) [1], Dieuleveult and Patel (2019) [2], Stich and Karimireddy (2019) [3], Khaled et al. (2020) [4], Woodworth et al. (2020) [5], Yuan and Ma (2020) [6], Woodworth et al. (2021) [7]) rather than the heterogeneous distributed model, and we will mention this distinction.
>
> >Is there a fundamental benefit of approximating constrained subproblems instead of a sequence of regularized Newton steps? If so, I believe it would be great to highlight it. Or is it the way [A] approximates the steps (directly via sketching) that makes the analysis viable without resorting to constrained subproblems? You don't assume strong convexity (but the experiments use regularized logistic regression), could that be an explanation?
>
> The dependence on strong convexity is indeed an important difference, as the work of Karimireddy et al., 2018 [8] (whose analysis we build upon) is specifically interested in obtaining global linear convergence without strong convexity.  For the experiments, we have also compared all the methods on unregularized logistic regression problems for minimizing test loss in Appendix E.4. We will add further experiments on unregularized logistic regression.
>
> >You mention that the advantage of using Hessian-vector products is due to the distributed setting, and in particular the fact that the lower bound is not quadratic, so the Hessian-vector information is not equivalent to gradient information. Yet, if I understand correctly, this is also true for generalized linear models, which is the prime example for which stochastic Hessian vector products are easy to compute (and quasi-self concordance holds). From this perspective, it seems that using this kind of second-order information is mainly beneficial when it is not that easy to compute, or have I missed something?
>
> As noted by the reviewer, stochastic Hessian-vector products are easy to compute for generalized linear models. Therefore, if we understand the question correctly, while these stochastic Hessian-vector products do not help with GLMs in the sequential setting, they may still provide benefit in the distributed setting.
>
> >Is it possible that some slackness in the lower bound comes from different variance that is assumed for second order information (the \rho parameter)? I believe that the "comparison with the lower bound" part is great, but could be improved even further.
>
> Indeed, the additional $\rho$ parameter is the source of some slackness, which results due to the fact that the lower bound is only for first-order methods. We will rework our discussion of these comparisons to make them more transparent.
>
> **References**
>
> [1] Stich, Sebastian U. "Local SGD converges fast and communicates little." arXiv preprint arXiv:1805.09767 (2018).
>
> [2] Dieuleveut, Aymeric, and Kumar Kshitij Patel. "Communication trade-offs for Local-SGD with large step size." Advances in Neural Information Processing Systems 32 (2019): 13601-13612.
>
> [3] Stich, Sebastian U., and Sai Praneeth Karimireddy. "The error-feedback framework: Better rates for SGD with delayed gradients and compressed communication." arXiv preprint arXiv:1909.05350 (2019).
>
> [4] Khaled, Ahmed, Konstantin Mishchenko, and Peter Richtárik. "Tighter theory for local SGD on identical and heterogeneous data." International Conference on Artificial Intelligence and Statistics. PMLR, 2020.
>
> [5] Woodworth, Blake, et al. "Is local SGD better than minibatch SGD?." International Conference on Machine Learning. PMLR, 2020.
>
> [6] Yuan, Honglin, and Tengyu Ma. "Federated accelerated stochastic gradient descent." arXiv preprint arXiv:2006.08950 (2020).
>
> [7] Woodworth, Blake, et al. "The Min-Max Complexity of Distributed Stochastic Convex Optimization with Intermittent Communication." arXiv preprint arXiv:2102.01583 (2021).
>
> [8] Karimireddy, Sai Praneeth, Sebastian U. Stich, and Martin Jaggi. "Global linear convergence of Newton's method without strong-convexity or Lipschitz gradients." arXiv preprint arXiv:1806.00413 (2018).

---

> > ### Comment · Reviewer_sbMG · 2021-08-18
> > **Thank you**
> >
> > Thank you for the clarifications !
> >
> > About the lower bound I think I got a little confused so my question was not so clear indeed. Maybe you could explain in one short sentence (if possible) why the worst case functions you consider are different in the distributed setting. It's quite minor though.

---

> > > ### Author Response · Authors · 2021-08-18
> > > **Worst case functions for distributed setting**
> > >
> > > The reasoning, as originally presented by Woodworth et al. (2021) [1], is that while lower bounds for sequential smooth convex optimization can be obtained using quadratics, the min-max complexity of quadratics in the distributed setting depends only on the product $KR$, and so Woodworth et al. required a non-quadratic function (eqs. (4) and (5) in [1]) to establish a lower bound which distinguishes between $K$ and $R$ in the distributed smooth convex setting.
> > >
> > > [1] Woodworth, Blake, et al. "The Min-Max Complexity of Distributed Stochastic Convex Optimization with Intermittent Communication." arXiv preprint arXiv:2102.01583 (2021).

---

### Official Review · Reviewer_4Wbd · 2021-07-16

**Rating:** 6
**Confidence:** 3

**Summary:**

This work studies a distributed Newton-type algorithm for unconstrained convex optimization.


**Limitations And Societal Impact:**

Not applicable.

**Main Review:**

1. The least squares step in the Newton step can be solved using sketching based methods as well. There are published papers in the literature that focus on “distributed Newton sketch” type of algorithms where computing the exact Hessian is avoided and the Newton update is approximate. The algorithm presented in this paper seems to work very similarly to distributed Newton sketch and address the same problem. I think it would be great to compare the presented algorithm with the distributed Newton sketch approach so that the contributions of this paper can be better understood.

2. Is “sub-optimality” (y-axis of figure 1) defined somewhere in the paper?

3. I agree that comparing the performance with respect to the number of communication rounds is a good idea. But I think this, by itself, does not show the whole picture, since time complexity and required memory are important considerations as well. For instance, how does the performance look with respect to runtime?

4. Also, related to my point in 1; it has been known for a while that least-squares objectives can be optimized to high accuracy with a single round of communication, i.e. averaging sketched solutions. Authors, in fact, cite the work by Wang et al., 2018, which is one of the papers that study the concept of averaging sketched solutions. It might justify/clarify the motivation of this work to further comment on how “local-SGD for least squares” and “averaging sketched solutions” compare.


**Time Spent Reviewing:**

3.5

---

> ### Author Response · Authors · 2021-08-10
> **Response to Reviewer 4Wbd**
>
> Thanks for the review and emphasizing the similarities with Newton sketch.
>
> **Regarding the relationship with distributed Newton sketch**
>
> We do cite Wang et al. 2018, as well as Shamir et al. 2014, Zhang and Xiao 2015, Reddi et al. 2016, Crane and Roosta 2019, Islamov et al. 2021, Gupta et al. 2021.  Are these the papers you had in mind, or some other papers?
>
> There is certainly a strong relationship between the distributed Newton sketch and our method, and we should (and will) include a more comprehensive discussion. The main difference is that we work in a different, stochastic setting. The distributed Newton sketch papers we are familiar with, e.g. GIANT, Wang et al. (2018), work in a “batch” setting, where the data is fixed, but split between machines, all data is accessed each round, possibly even multiple times per round (e.g. in multiple iterations of conjugate gradient as in GIANT), and the main concern is communication between the machines. Our setting is a purely stochastic (one-pass, “streaming”) setting, where each sample from the distribution is used once, for computing a single stochastic gradient estimate or single Hessian-vector product estimate. In terms of the method, this translates to solving each local sub-problem using stochastic gradient descent on fresh and different examples each round. This is in contrast to methods (such as GIANT) which solve the subproblem involving the sketched Hessian and exact gradient using, e.g., batch conjugate gradient descent. The goal of the stochastic setting, then, is to get error that is comparable to what one could get with the same number of overall samples, i.e. with a number of samples equal to the total number of oracle accesses.  This is different from batch methods such as GIANT, where the total number of samples used is much smaller than the number of gradient or Hessian-vector-product computations.  In fact (and following your review), FedSN can indeed be viewed as a stochastic variant of distributed Newton sketch, and we agree it would be useful to also discuss it as such.
>
> This stochasticity significantly complicates the analysis. Both the fact that we are using fresh examples all the time, and that we use SGD to solve the subproblems, and so can only get probabilistic guarantees, which we then need to propagate into an analysis of the Newton-type method, are added complications. These are discussed in Section 3.
>
> In particular, our method is valid also in a purely stochastic, “streaming” setting, such as Federated learning, where typical distributed Newton sketch methods, such as GIANT, cannot be implemented as-is (since data cannot be re-accessed in different rounds, and in a “pure” setting, each data point can only be used once).
>
> When optimizing an empirical objective based on fixed data, both settings are potentially applicable.  In our stochastic view, in this case, the “population” is the training set, and the “population objective” is the (possibly regularized) training error.  This is in fact a setting we consider in the experiments  (the choice of experimenting with optimizing an empirical objective, as opposed to a true population objective on an unknown “world” distribution, which is where the method is geared, is motivating by replicating the experimental setup from Yuan and Ma (2020) in order to have a direct comparison, and also because this is a more controlled setting where it's easy to measure the population objective and its suboptimality).  In this case, both approaches (batch approaches such as GIANT, and FedSN) are possible, and a fair comparison is possible by fixing or comparing the number of single-point gradient or Hessian-vector-product computations.  But note that in the regimes we consider, the number of single-point computations per round is much lower than the size of the training set, and so GIANT is not applicable as-is, and it is not entirely clear how to compare to GIANT, as we must also make GIANT stochastic, i.e. we’ll end up with something more similar to our method.  But that said, it would be interesting to see (and we have started doing that) whether even in the settings where GIANT is experimented with and demonstrated, we can improve the runtime to get the same error using FedSN (this is not our focus and not the setting we had in mind, and to make things fair and practical we will also need to split the data across machines, but it’s still interesting to check).
>
> Another example of the difference between the stochastic and batch settings is that distributed Newton sketch makes sense also for quadratic objectives (e.g. in the form of DANE), but in the stochastic setting, FedSN has no advantage over local SGD for quadratic objectives. This emphasizes how different the situations are.
>
> To summarize, FedSN can indeed be viewed as a *stochastic* Newton sketch method, and we agree it would add to the paper to discuss it as such and highlight the differences versus batch Newton sketch methods.  But the stochastic setting is *significantly* different, with different issues, considerations and comparisons.
>
> **Regarding other questions and concerns**
>
> >Is “sub-optimality” (y-axis of figure 1) defined somewhere in the paper?
>
> While "sub-optimality" is defined in Appendix E.2, we will further clarify its meaning in the main body of the paper. We first pre-compute the optimal loss $l^\star$ for each subplot. Then for each configuration of every algorithm, i.e., every point in a subplot, we obtain the best possible loss $l$ for that configuration by tuning the learning rate. The relative sub-optimality is then calculated as $\log(\frac{l-l^\star}{l^\star})$. We will also relabel the y-axis as “log(relative sub-optimality)” to be precise.
>
> > I agree that comparing the performance with respect to the number of communication rounds is a good idea. But I think this, by itself, does not show the whole picture, since time complexity and required memory are important considerations as well. For instance, how does the performance look with respect to runtime?
>
> In our experiments, we keep the number of iterations per machine fixed, and all methods use essentially the same amount of runtime per iteration and the same amount of memory.   This is verified in the appendix.  Specifically, in Appendix E.3 we discuss the computational cost of a Hessian-vector product oracle, and show in Table 2 that the average run-time of all the algorithms is nearly the same. Similarly, looking at the algorithms’ pseudo-codes in Appendix E.1, it is clear that they all need only to store the model parameters, giving a $\mathcal{O}(d)$ memory cost.
>
> Thus, with all other factors being similar, we compare the methods with the same number of communication rounds while fixing the theoretical parallel runtime $T$. We will update the language to elaborate more on these issues, and also include a more detailed version of the experiment in Table 2 in our revised version.

---

> > ### Comment · Reviewer_4Wbd · 2021-08-25
> > **Response to the authors**
> >
> > I thank the authors for their detailed response, especially on "the connection to distributed Newton sketch". I raise my score by one point.

---

### Official Review · Reviewer_we6s · 2021-07-16

**Rating:** 6
**Confidence:** 3

**Summary:**

This paper proposes a federated stochastic Newton algorithm for optimizing quasi-self-concordant objectives,
with concrete theoretical analysis showing this algorithm can outperform other existing first-order stocahstic optimization algorithms and numerical experiments to support the analysis.

**Limitations And Societal Impact:**

The biggest limitation of this paper is how to compute the hessian. This paper avoids computing the inverse of hessian, which is a great improvement over the original Newton algorithm. However, it is not fair to say it uses less communication because the hessian is communicated, when the dimension is moderately big, the communication overhead will be significant.

**Main Review:**

* Originality

This work is novel because it approximates a Newton step by solving a local optimization problem using stochastic gradient descent and provides extensive theoretical analysis,
which are not well-explored in prior literature.

* Quality

FED-SN reaches a convergence rate that dominates other algorithms in certain regimes.
The proof is detailed and well-written, lists most assumptions and important steps.
However, the assumptions in this paper is too strong, but it is understandable since the analysis of FED-SN is challenging.

* Clarity

This paper is clearly written, well-organized and reproducible.

* Significance

The results obtained in this paper is novel, and improves upon current state-of-the art results in certain settings.

**Time Spent Reviewing:**

5

---

> ### Author Response · Authors · 2021-08-10
> **Response to Reviewer we6s**
>
> We thank the reviewer for their comments. There seems to be a significant confusion regarding our method, which we wish to correct.
>
> In our method, the Hessian is never computed, nor is it ever communicated. Each round involves communicating only $\mathcal{O}(M)$ (one per machine) vectors, of the same dimensionality as the weight vector (i.e. the same type of communication as in mini-batch or local SGD).  More specifically, we compute, on each individual machine, stochastic Hessian-vector products at every iteration. Then, at the end of each round, only the final iterates are communicated. This is computationally appealing because, as discussed in Appendix E.3, stochastic Hessian-vector products can be computed, for many problems of interest, in essentially the same time as stochastic gradients, i.e., $\mathcal{O}(d)$, and the communicated final iterates are also of size $\mathcal{O}(d)$. We admit that our initial presentation was somewhat confusing, in that the Hessian appears in our algorithm, but it was there only for conceptual purposes, i.e., to explicitly state which quadratic sub-problem we approximately solve. The Hessian is not actually computed or passed along. Instead, the quadratic problem involving it is solved using the Hessian-vector-products discussed above. We will revise the presentation to avoid this confusion and make the operations performed much clearer.

---

> > ### Comment · Reviewer_we6s · 2021-09-13
> > **Thanks for clarifying**
> >
> > Thanks for your clarification, this is a nice paper : )

---

### Decision · Program_Chairs · 2021-09-28

**Decision:**

Accept (Poster)

**Comment:**

The paper proposes a distributed Newton method that leverages a recently develop one-round communication methods for solving least-squares. The papers backs up all claims with clear theoretical arguments and numerical experiments. The reviewers have also reached a consensus that the paper should be accepted.

**Consistency Experiment:**

NeurIPS has a long history of experimentation. In 2014, NeurIPS ran an experiment in which 10% of submissions were reviewed by two independent committees to quantify the randomness in the review process. This year, we repeated a variant of this experiment to see how the quality of the review process has changed over time.  This paper was part of the experiment and was therefore assigned to two committees (consisting of reviewers, an Area Chair, and a Senior Area Chair) that reached independent decisions.  If both committees made the same recommendation, this recommendation was followed. If a single committee recommended acceptance, the paper was accepted (with the exception of a few cases in which the other committee identified what we considered a fatal flaw, e.g., an error in a key result).

This copy’s committee reached the following decision: **Accept (Poster)**

The other committee assigned to the paper recommended **Reject**.  You can find the other set of reviews, along with any follow up discussion with the authors here:
https://openreview.net/forum?id=ui0sz9Y2x9X